# Decoupled Training with Local Reinforcement Fine-Tuning in Federated Learning

Yuting Ma [1]   Lechao Cheng [2]   Xiaohua Xu [1]

## Abstract

Federated Learning (FL) with pre-trained Vision-Language Models (VLMs) has emerged as a promising paradigm for various downstream tasks. By leveraging its strong representations, recent studies improve task adaptation under insufficient local data while preserving generalization. However, these methods emphasize fully local optimization with simple parameter aggregation, which can amplify inter-client optimization inconsistency and intra-client over-specialization under heterogeneous and full-data FL settings, making it difficult to balance global task adaptation and generalization. To address these challenges, we propose FedDTL, a novel federated VLM framework that decouples the image encoder and text encoder across clients and the server. Through decoupled encoder training with server-client modality alignment, FedDTL promotes coherent global semantic update and reduces inter-client optimization inconsistency, improving global task adaptation. To further mitigate intra-client over-specialization, we introduce a two-stage local fine-tuning, where a supervised fine-tuning stage enables rapid and reliable warm-start, followed by a reinforcement learning stage that enhances generalization. Extensive experiments on multiple benchmarks, including label skew and feature shift, demonstrate that FedDTL achieves an effective balance between global task adaptation and generalization under various FL data distributions in both few-shot and full-data regimes.

[1]School of Computer Science and Technology, University of Science and Technology of China, Hefei, Anhui, China [2]School of Computer Science and Information Engineering, Hefei University of Technology, Hefei, Anhui, China. Correspondence to: Lechao Cheng <chenglc@hfut.edu.cn>, Xiaohua Xu <xiaohuaxu@ustc.edu.cn>.

*Proceedings of the 43rd International Conference on Machine Learning*, Seoul, South Korea. PMLR 306, 2026. Copyright 2026 by the author(s).

## 1. Introduction

Federated learning (McMahan et al., 2017) is an advanced distributed learning paradigm that aims to protect data privacy by collaboratively training a global model across clients without centralizing local data. However, conventional FL methods (Zhu et al., 2021; Wang et al., 2023; Ye et al., 2023; Wang et al., 2024b;c) often train models from scratch, resulting in slow convergence and high computation costs, particularly under heterogeneous data, where rapid adaptation to downstream tasks becomes challenging. In this context, pre-trained vision-language models such as CLIP (Radford et al., 2021) have been integrated into FL to accelerate downstream task adaptation by leveraging their transferable general representations. Given that CLIP contains millions of parameters, directly fine-tuning the entire model would cause high computational overhead. Therefore, most federated VLM approaches freeze the pre-trained CLIP backbone and adopt parameter-efficient fine-tuning (PEFT) techniques, such as prompt tuning (Zhou et al., 2022b), adapter tuning (Yu et al., 2023), and LoRA tuning (Zanella & Ayed, 2024) to update a few parameters and reduce computational overhead.

Building on PEFT, federated VLM methods (Guo et al., 2023; Li et al., 2024; Zhang et al., 2025) often introduce client-specific modules and aggregate lightweight prompts/adapters to improve task adaptation. However, under *Non-IID* data, clients optimize mismatched local objectives (e.g., label skew and feature shift), yielding misaligned local gradients and pronounced *inter-client optimization inconsistency* (representation-level client drift). After multiple local steps, clients drift toward different representation "coordinate systems", so parameter averaging no longer produces a coherent global semantic update, ultimately degrading global task adaptation—especially in strong *Non-IID* with longer local training trajectories and *full-data* regimes, where clients train models with all available local samples.

Apart from inter-client drift, long local training trajectories can also trigger substantial *intra-client over-specialization*. As clients continuously train local data, local optimization typically becomes stronger and longer, and lightweight PEFT parameters can become overly specialized to each client's biased label frequencies and feature statistics. This

often yields improved local fitting but compromised generalization to unseen classes or shifted domains, especially under data heterogeneity and *full-data* regimes. Recent federated VLM works partially address these issues via additional regularization or alignment constraints (e.g., auxiliary losses or shared projection layers) (Cui et al., 2024; Ghiasvand et al., 2026), or stronger knowledge-sharing tuning strategies (Khattak et al., 2023; Guo et al., 2024). However, since these approaches still rely on local-only optimization with parameter averaging as the dominant mechanism for cross-client knowledge transfer, they remain insufficient to systematically resolve *inter-client optimization inconsistency*. Moreover, most existing evaluations are conducted only in *few-shot* regimes, potentially underestimating the compounded effects of inconsistency and over-specialization in *full-data* settings, a challenging and realistic FL scenario where local training with sufficient samples can amplify the above negative effects. These limitations motivate us to explore:

*How to alleviate inter-client optimization inconsistency and intra-client over-specialization in federated VLMs for ensuring an effective balance between global task adaptation and generalization across various FL data distributions in both few-shot and full-data settings?*

To address these challenges, we propose **FedDTL**, a novel **Fed**erated learning framework with **D**ecoupled encoder training and **T**wo-stage **L**ocal fine-tuning. To mitigate inter-client optimization inconsistency, we propose a decoupled encoder training scheme inspired by the modality decoupling and alignment paradigm in CLIP, which presents a compelling analogy to server-client training and knowledge broadcasting in FL. Because the image encoder directly processes private raw images, it is trained locally to preserve data privacy. In contrast, the text encoder, which operates on category-level textual descriptions, can be optimized on the server to learn globally consistent semantic representations across heterogeneous clients. By the modality alignment of local visual representations with global textual representations, our decoupled scheme effectively reduces inter-client optimization inconsistency compared to fully local training with parameter averaging, thereby improving global task adaptation under diverse FL data distributions in both *few-shot* and *full-data* settings.

Motivated by the study (Chu et al., 2025), which reveals that supervised fine-tuning (SFT) tends to memorize training data and shows limited generalization to unseen classes or shifted domains. Notably, the issue can be amplified in long local training under heterogeneous data and *full-data* regimes. In contrast, reinforcement learning (RL) provides a complementary optimization paradigm to alleviate over-memorization and contributes to generalization enhancement. Therefore, we employ RL to refine local

fine-tuning for mitigating intra-client over-specialization instead of adding regularization or alignment constraints to SFT. Unlike prior RL-for-FL studies (Jin et al., 2024; Lan et al., 2025; Xiong et al., 2025), which mainly target system heterogeneity (e.g., straggler mitigation and latency), our RL objective targets model generalization under statistical heterogeneity. Although purely RL can encourage more conservative and generalization-aware optimization, it may be less sample-efficient for downstream task adaptation. Therefore, we propose a two-stage local fine-tuning: an initial SFT stage provides a fast and reliable optimization warm-start, while a subsequent Group Relative Policy Optimization (GRPO)-inspired RL stage refines the local update trajectory to reduce intra-client over-specialization and enhance generalization under various data distributions. Through the interplay of the decoupled encoder scheme and two-stage local fine-tuning, FedDTL can balance global task adaptation and generalization across various FL scenarios. Moreover, we adopt low-rank adaptation (LoRA) tuning to optimize encoders, reducing local computation cost. In summary, our main contributions are:

- We introduce a decoupled training scheme for federated VLMs, where the image encoder is optimized locally to preserve privacy while the centralized text encoder provides a globally consistent semantic anchor. By aligning client-side visual representations to server-side textual semantics, our framework reduces inter-client inconsistency and improves global task adaptation under various FL settings.

- We propose a two-stage local fine-tuning schedule: an SFT warm-start stage followed by a novel GRPO-inspired RL stage to suppress intra-client over-specialization and enhance generalization, leading to a more stable local fine-tuning under heterogeneous and *full-data* federated settings.

- Extensive experiments across label skew and feature shift under *few-shot* and *full-data* regimes demonstrate consistent gains of FedDTL in balancing global task adaptation and generalization over prior federated VLM baselines.

## 2. Related Works

**Federated Learning in Vision-Language Models.** Recent federated learning studies have integrated pre-trained VLMs (Radford et al., 2021; Jiang et al., 2025; Shen et al., 2025; Wang et al., 2025) with PEFT techniques (Zhou et al., 2022a; Yu et al., 2023; Zanella & Ayed, 2024; Wang et al., 2024d) to accelerate downstream task adaptation and reduce computational cost in heterogeneous data scenarios. For instance, FLoRA (Nguyen et al., 2024) applies LoRA

tuning (Zanella & Ayed, 2024) on a frozen CLIP backbone to adapt heterogeneous FL data distributions, while pFedDC (Zhang et al., 2025) employs a cross-fusion module with multi-modal prompt tuning to enhance personalization across heterogeneous clients. Although these methods improve local adaptation, their emphasis on fitting local distributions amplifies intra-client over-specialization, further compromising generalization on unseen classes or shifted domains. To alleviate this issue, subsequent works focus on reducing overfitting to balance task adaptation and generalization in federated VLMs. FedPGP (Cui et al., 2024) disentangles global and local prompts and incorporates a prompt-wise contrastive loss to suppress overfitting and maintain generalization, while pFedMMA (Ghiasvand et al., 2026) leverages shared projection layers and adapter tuning to enhance modality alignment and preserve generalization during personalization. Other methods, such as FedMaPLe (Khattak et al., 2023) and PromptFL (Guo et al., 2024), improve performance balance by employing advanced fine-tuning strategies and sharing more local knowledge. However, existing methods mainly target *few-shot* settings and rely on fully local training with simple server-side parameter aggregation, which is insufficient to address inter-client optimization inconsistency and overlooks *full-data* regimes. In contrast, our FedDTL employs a decoupled encoder training scheme with the two-stage local fine-tuning to mitigate inter-client inconsistency and intra-client over-specialization, balancing global task adaptation and generalization in *few-shot* and *full-data* settings.

**Reinforcement Learning for Federated Optimization.** With the development of reinforcement learning (RL), representative algorithms such as policy gradient (Yuan et al., 2022), PPO (Schulman et al., 2017), and GRPO (Shao et al., 2024) have been widely used for decision-making and stable model optimization with enhanced generalization. Recently, works in federated learning (Jin et al., 2024; Wang et al., 2024a; Chen et al., 2025; Lan et al., 2025; Xiong et al., 2025) have exploited RL to address system heterogeneity. For example, FedPPO (Jin et al., 2024) employs PPO to mitigate constraint heterogeneity, while AFedPG (Lan et al., 2025) adopts policy gradient to address lagged policies in asynchronous federated learning. However, these methods primarily focus on system coordination and decision making, and are difficult to reduce intra-client over-specialization and enhance generalization in federated VLMs. In parallel, recent works (Chu et al., 2025; Tan et al., 2025; Yang et al., 2025) have shown that supervised fine-tuning (SFT) tends to memorize training rules, whereas reinforcement learning can encourage generalization. Motivated by them, we exploit the complementary strengths of SFT and RL to reduce intra-client over-specialization and enhance generalization with rapid task adaptation in federated VLMs. Unlike prior works that directly apply RL algorithms, we make some

adjustments and tailor RL to the federated VLMs, ensuring stable and effective local fine-tuning in various federated learning data settings.

## 3. Method

### 3.1. Overview

As shown in Figure 1, we propose a federated VLM framework involving a central server and $K$ clients, where each client $k$ has a private dataset $\mathcal{D}_k = \{(x_i, y_i)\}_{i=1}^{N_k}$ containing $N_k$ labeled samples. To mitigate inter-client optimization inconsistency under heterogeneous data distributions, we design a **decoupled encoder training** scheme: client-side local image encoders and server-side global text encoder. Moreover, we employ LoRA tuning to train encoders. In this decoupled manner, clients and the server achieve federated broadcasting by sharing lightweight LoRA updates and high-compressed embeddings. By global textual knowledge learning and modality alignment between local visual and global textual representations, FedDTL encourages clients to converge toward a unified semantic space, reducing inter-client inconsistency and improving global task adaptation in *few-shot* and *full-data* settings across heterogeneous clients. To further alleviate intra-client over-specialization, we propose a **two-stage local fine-tuning** strategy: a SFT-based task adaptation stage to provide a fast and reliable warm-start for the subsequent RL-based generalization enhancement stage. The decoupled encoder scheme with two-stage local fine-tuning effectively balances global task adaptation and generalization in various FL scenarios.

### 3.2. Decoupled Encoder with Efficient Training

Existing federated VLM approaches often optimize the entire model on clients and perform server-side parameter aggregation to share knowledge. However, this paradigm struggles to align mismatched local objectives into a unified global solution under *full-data* and heterogeneous scenarios, remaining insufficient to resolve inter-client optimization inconsistency. Moreover, accumulated local updates tend to amplify intra-client over-specialization, ultimately degrading global task adaptation and generalization. To address challenges, we turn our attention to the modality decoupling and alignment paradigm in CLIP (Radford et al., 2021) and design a decoupled encoder training scheme: 1) **Privacy-preserving image encoding**: the image encoder $\mathcal{V}$ takes a private image $x$ along with a learnable class token as input and produces the final class token, which is then mapped to a normalized vision-language latent space to obtain an image embedding $\bar{z}_v = \mathcal{V}(x)$. Therefore, $\mathcal{V}$ needs to be trained on the client side. 2) **Global text encoding**: Text encoder $\mathcal{T}$ only requires category name to generate the text embedding $\bar{z}_{\text{text}} = \mathcal{T}([\text{classname}])$ in the same latent space, making $\mathcal{T}$ suitable for global training on the server side. Image

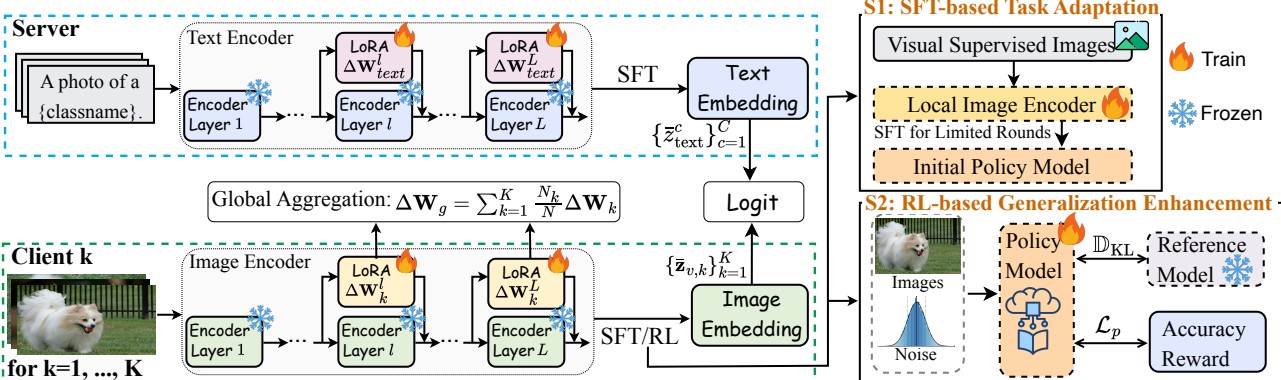

*Figure 1.* The framework of FedDTL. Each client $k$ performs a two-stage local fine-tuning with local image encoders: a supervised fine-tuning stage for rapid task adaptation followed by a reinforcement learning stage for generalization enhancement. The server trains a global text encoder via supervised fine-tuning and performs parameter aggregation. Clients and the server transmit latent embeddings and visual LoRA parameters for knowledge sharing, enabling effective global task adaptation on various federated data scenarios.

and text embeddings are aligned to compute the prediction probability of the image $x$ for each class $c$:

$$p(\hat{y} = c|x) = \frac{\exp(\text{sim}(\bar{z}_v, \bar{z}_{\text{text}}^c)/\tau)}{\sum_{j=1}^{C} \exp(\text{sim}(\bar{z}_v, \bar{z}_{\text{text}}^j)/\tau)}, \quad (1)$$

where $\tau$ represents a temperature hyperparameter and $\text{sim}(\cdot, \cdot)$ denotes cosine similarity. The embedding alignment between language and vision modality bears striking parallels to server-client knowledge sharing, making the decoupled encoder design suitable for our method. Using the global text encoder as a semantic "constraint", distributed clients can align local visual representations to a unified textual embedding space without maintaining local text encoders. This design effectively mitigates inter-client optimization inconsistency while reducing local computational overhead, thereby improving global task adaptation under various federated learning scenarios in both *few-shot* and *full-data* regimes.

To ensure efficient training under the decoupled encoder scheme, we freeze the pre-trained backbone and apply LoRA tuning to deeper encoder layers instead of fine-tuning all parameters. Specifically, LoRA tuning introduces low-rank trainable weight matrices into the last few backbone layers to capture task-specific semantics without additional model architectures, ensuring lightweight updating while preserving general representations in shallow layers. For any self-attention weight matrix $W$ in image and text encoders, we decompose it into a frozen pre-trained weight $W_0 \in \mathbb{R}^{d_1 \times d_2}$ and two low-rank learnable matrices $A \in \mathbb{R}^{r \times d_2}$ and $B \in \mathbb{R}^{d_1 \times r}$:

$$W = W_0 + \Delta W = W_0 + BA, \quad (2)$$

where the rank $r \ll \{d_1, d_2\}$. By combining the decoupled encoder scheme with LoRA tuning, our FedDTL facilitates

global task adaptation and reduces local computation burden across diverse federated settings, especially for *full-data* and heterogeneous data distributions.

### 3.3. Two-Stage Local Fine-Tuning

Beyond mitigating inter-client optimization inconsistency to improve global task adaptation by the proposed decoupled encoder scheme with LoRA tuning, another important challenge in our study is to alleviate intra-client over-specialization, which is crucial for maintaining generalization under various federated scenarios. Therefore, we propose a two-stage local fine-tuning paradigm: an initial SFT-based task adaptation stage for fast and reliable optimization warm-start, with a subsequent RL-based generalization enhancement stage to reduce over-specialization.

**Stage 1: SFT-based Task Adaptation.** During this stage, each client $k$ performs supervised fine-tuning to update the LoRA parameters of its local image encoder, ensuring rapid and efficient optimization warm-start. Specifically, the image encoder $\mathcal{V}_k$ consists of all pre-trained frozen parameters $\mathbf{W}_0$ and all LoRA trainable parameters $\Delta \mathbf{W}_k$ inserted from the $l$-th backbone layer. In each global round $t$, each client $k$ initializes all LoRA parameters $\Delta \mathbf{W}_k^t$ with the global aggregated LoRA parameters $\Delta \mathbf{W}_g^{t-1}$ and receives global text embeddings $\{\bar{z}_{\text{text}}^{c,t-1}\}_{c=1}^{C}$ from the server. Given a local image $x_i$, $\mathcal{V}_k^t$ produces a local image embedding $\bar{z}_{v,k,i}^t$. Then, client $k$ aligns all local embeddings $\bar{\mathbf{z}}_{v,k}^t = \{\bar{z}_{v,k,i}^t\}_{i=1}^{N_k}$ with global text embeddings $\{\bar{z}_{\text{text}}^{c,t-1}\}_{c=1}^{C}$ and performs $T_e$ local optimization epochs via a cross-entropy loss function:

$$\mathcal{L}_{ce} = -\frac{1}{N_k} \sum_{(x_i,y_i) \in \mathcal{D}_k} \sum_c y_i \log p(\hat{y} = c|x_i), \quad (3)$$

where the prediction probability $p$ refers to Equation (1). Consequently, the local optimization objective of the SFT

stage for client $k$ is defined as:

$$\min_{\Delta \mathbf{W}_k} \mathcal{L}_{ce}([\mathbf{W}_0, \Delta \mathbf{W}_k]; \{\bar{z}_{\text{text}}^c\}_{c=1}^C, \mathcal{D}_k). \quad (4)$$

After performing limited $M$ global communication rounds, $\mathcal{V}_k^M$ reaches a stable task-adapted state and serves as a fast and robust warm-start for the subsequent RL stage.

**Stage 2: RL-based Generalization Enhancement.** In this stage, each client $k$ initializes the local policy model $\pi_{\theta_k}$ from the task-adapted encoder $\mathcal{V}_k^M$. By parameterizing the logit output as a categorical distribution over class, the image encoder naturally induces the policy model, enabling RL-based sampling and policy updates without introducing additional layers. Inspired by Group Relative Policy Optimization (GRPO) (Shao et al., 2024), which optimizes the policy model using relative rewards among a group of sampled actions without additional value networks, we leverage group-based relative optimization to simplify local RL fine-tuning and reduce computational cost. However, directly applying GRPO to our encoder-based policy model is infeasible. Unlike LLMs that generate diverse outputs through stochastic decoding, CLIP-style image encoders produce deterministic embeddings for a given image, resulting in identical action outputs and invalid reward estimation. To address this challenge, we introduce a controlled stochasticity mechanism into the sampling process and adjust the corresponding details to adapt to RL optimization. This stage includes the sampling process, reward estimation, and policy update.

**Sampling Process.** Before processing each batch $\{x_i\}_{i=1}^{bs}$, we treat the current policy before update as a fixed old policy $\pi_{\theta_{k,\text{old}}}$ and freeze $\pi_{\theta_{k,\text{old}}}$ for sampling and reward computation. To induce stochasticity without modifying the encoder architecture, we inject unbiased Gaussian noise $\varepsilon$ into the latent embedding space before feature normalization:

$$z_v = z_v + \varepsilon, \; \varepsilon \sim \mathcal{N}(0, \sigma^2 I). \quad (5)$$

The noise scale $\sigma$ is kept small to preserve stable optimization. With noise injection, the policy model generates $G$ different actions for the same image $x_i$:

$$a_{i,j} \sim \pi_{\theta_{k,\text{old}}}(a_i|x_i, \varepsilon), \; j = 1, 2, \cdots, G. \quad (6)$$

**Reward Estimation.** To preserve classification correctness on downstream tasks, we utilize an accuracy-based reward. Specifically, if the image $x_i$ with noise $\varepsilon_{i,j}$ predicts correctly, the corresponding reward $r_{i,j}=1$, otherwise $r_{i,j}=0$. For each image $x_i$, the reward set $\{r_{i,1}, \cdots, r_{i,G}\}$ is normalized within the sampled group to compute relative advantages:

$$A_{i,j} = \frac{r_{i,j} - \text{mean}_j(r_{i,j})}{\text{std}_j(r_{i,j})}, \; j = 1, 2, \cdots, G. \quad (7)$$

**Policy Update.** Following GRPO, the optimization objective consists of a $\epsilon$-clipping policy gradient term $\mathcal{L}_p$ and a KL regularization term $\mathbb{D}_{\text{KL}}$. The current policy model $\pi_{\theta_k}$ is encouraged to produce latent embeddings that favor the correct category by:

$$\mathcal{L}_p = \min\left[\rho_{i,j} A_{i,j}, \; \text{clip}(\rho_{i,j}, 1 - \epsilon, 1 + \epsilon) A_{i,j}\right], \quad (8)$$

where $\rho_{i,j} = \frac{\pi_{\theta_k}(a_{i,j}|x_i)}{\pi_{\theta_{k,\text{old}}}(a_{i,j}|x_i, \varepsilon_{i,j})}$. Notably, noises are only injected into the sampling process, while the current policy model remains deterministic during policy update, ensuring stable training. To avoid excessive deviation from the desired embedding alignment, a KL regularization $\mathbb{D}_{\text{KL}}$ is applied between the current policy model and a reference model $\pi_{k,\text{ref}}$ through an unbiased estimator:

$$\mathbb{D}_{\text{KL}} = \frac{\pi_{k,\text{ref}}(a_{i,j}|x_i)}{\pi_{\theta_k}(a_{i,j}|x_i)} - \log\frac{\pi_{k,\text{ref}}(a_{i,j}|x_i)}{\pi_{\theta_k}(a_{i,j}|x_i)} - 1. \quad (9)$$

We construct $\pi_{k,\text{ref}}$ from a parameter combination of the final global SFT (the task-adapted model) and the latest global policy model with equal coefficient (0.5). This hybrid reference model provides a task-aware regularization signal for RL optimization, effectively preventing excessive policy drift and improving training stability. In the global round $t$ of the RL stage, each client $k$ performs $T_e$ local optimization epochs for each batch to update LoRA parameters of the current policy model $\pi_{\theta_k}$ via:

$$\mathcal{L}_{rl} = -\frac{1}{G}\sum_{j=1}^{G}\frac{1}{bs}\sum_{i=1}^{bs}\left(\mathcal{L}_p - \beta\mathbb{D}_{\text{KL}}\right), \quad (10)$$

where $\beta$ is a coefficient in the RL loss $\mathcal{L}_{rl}$. Consequently, the local optimization objective of the RL stage for client $k$ is defined as:

$$\min_{\Delta \mathbf{W}_k} \mathcal{L}_{rl}([\mathbf{W}_0, \Delta \mathbf{W}_k]; \{\bar{z}_{\text{text}}^c\}_{c=1}^C, \mathcal{D}_k, \varepsilon). \quad (11)$$

In the GRPO-inspired RL stage, each client employs policy update with an accuracy-based reward and KL regularization to alleviate intra-client over-specialization in the long local training trajectories under heterogeneous data and *full-data* regimes, thereby enhancing generalization. We offer comprehensive details for this RL stage in Algorithm 1 (Appendix Section A).

### 3.4. Global Coordination and Optimization

After local fine-tuning, each client $k$ uploads its updated local LoRA parameters $\Delta \mathbf{W}_k$ with normalized image embeddings $\bar{\mathbf{z}}_{v,k}$ (without noise) to the server for global parameter aggregation and global optimization of the server-side text encoder. Notably, clients only need to upload the visual class token embedding instead of full image feature embeddings (e.g., patch token embeddings).

**Parameter Aggregation.** To implicitly share visual knowledge and facilitate global task adaptation, we employ a weighted averaging mechanism to aggregate local visual LoRA parameters. The mechanism is defined as:

$$\Delta \mathbf{W}_g = \sum_{k=1}^{K} \frac{N_k}{N} \Delta \mathbf{W}_k, \tag{12}$$

where $N$ is the sample size of all clients. In LoRA tuning, the global image encoder $\mathcal{V}_g$ is parameterized by the frozen backbone $\mathbf{W}_0$ and aggregated LoRA updates $\Delta \mathbf{W}_g$.

**Global Optimization.** Unlike prior methods that rely solely on parameter aggregation for knowledge sharing, our framework restricts parameter averaging to visual LoRA updates and performs global training on the language branch to promote consistent global knowledge learning. Specifically, the server-side global text encoder $\mathcal{T}_g$ takes category-level textual descriptions $text_c$ (i.e., "a photo of a [classname]") as input and produces global text embeddings $\{\bar{z}_{\text{text}}^c\}_{c=1}^C$. Then, these embeddings are used to update $\mathcal{T}_g$ with uploaded image embeddings $\{\bar{\mathbf{z}}_{v,k}\}_{k=1}^K$. Since $\mathcal{T}_g$ operates solely on label descriptions and highly compressed image features, global training could mitigate privacy leakage. Moreover, to preserve the intrinsic generalization of CLIP, we adopt the paired text encoder backbone rather than introducing heterogeneous LLM-based text encoders, thereby mitigating generalization degradation. Therefore, the global objective of $\mathcal{T}_g$ is:

$$\min_{\Delta \mathbf{W}_{\text{text}}} \mathcal{L}_{ce}([\mathbf{W}_{0,\text{text}}, \Delta \mathbf{W}_{\text{text}}]; \{\bar{\mathbf{z}}_{v,k}\}_{k=1}^K, text_c), \tag{13}$$

where $\mathbf{W}_{0,\text{text}}$ and $\Delta \mathbf{W}_{\text{text}}$ represent the frozen pre-trained and trainable LoRA textual parameters. Through global text encoder optimization, our framework can better learn a coherent unified semantic space and reduce representation-level client drift, thereby improving global task adaptation, especially under heterogeneous data and *full-data* settings, while reducing local computation cost. To further reduce communication cost under *full-data* settings, clients can upload a randomly sampled subset of image embeddings for each local class. Finally, the server broadcasts the global text embeddings of all classes $\{\bar{z}_{\text{text}}^c\}_{c=1}^C$ and global visual LoRA parameters $\Delta \mathbf{W}_g$ to all clients, completing a global FL round. We provide training details in Algorithm 1.

# 4. Experiments

## 4.1. Experimental Setup

**Datasets.** To study global task adaptation and generalization in federated VLMs, we evaluate FedDTL under *few-shot* and *full-data* settings with various heterogeneous data distributions, including label skew and feature shift. For label skew, we conduct base-to-novel class generalization

experiments on nine datasets: CIFAR10 (Krizhevsky et al., 2009), CIFAR100 (Krizhevsky et al., 2009), EuroSAT (Helber et al., 2019), Tiny-ImageNet (Deng et al., 2009), Oxford-Pet (Parkhi et al., 2012), Flower102 (Nilsback & Zisserman, 2008), Caltech101 (Fei-Fei et al., 2007), Caltech256 (Griffin et al., 2007), and Food101 (Bossard et al., 2014) under IID, Dirichlet with concentration parameter $\alpha$, and Non-IID with non-overlapping classes data settings. For each dataset, classes are split into base and novel classes. Each client performs local training in local classes. We test model performance on both base classes (all local classes across all clients) and novel classes (unseen during the entire training process) to evaluate global task adaptation and generalization. For feature shift, we evaluate model performance with two multi-domain datasets: Office-Caltech10 (Gong et al., 2012) with four domains and DomainNet (Peng et al., 2019) with six domains (following prior works, we select a subset of ten labels for evaluation). In the feature shift setting, all clients share the same label space, and each client is assigned all data from a single domain. To further simulate realistic FL scenarios, we introduce the label skew on top of the feature shift, where each domain is partitioned into 3 clients with Dirichlet($\alpha = 0.1$), resulting in 12 and 18 total clients in Office-Caltech10 and DomainNet, respectively. Model performance is tested in the entire target domain to measure task adaptation and generalization on cross-domain benchmarks.

**Baselines.** We compare our FedDTL with the following baselines: 1) Zero-shot CLIP (Radford et al., 2021), serving as a reference to evaluate the intrinsic generalization of the pre-trained CLIP. 2) pFedDC (Zhang et al., 2025), which focuses on the local training trajectory under heterogeneous data settings, providing a contrast to methods that balance model performance. 3) FedPGP (Cui et al., 2024), pFedMMA (Ghiasvand et al., 2026), PromptFL (Guo et al., 2024), and FedMaPLe (replicate MaPLe (Khattak et al., 2023) in the FL setting as shown in FedPGP), which aim to balance task adaptation and generalization in *few-shot* FL settings.

**Implementation Details.** We adopt ViT-B/16 (Dosovitskiy et al., 2021) as the default backbone. Each client performs $T_e = 2(\text{SFT})/3(\text{RL})$ local epochs over $T = 20$ communication rounds with $K = 5$ clients for all datasets. We use Adam optimizer with learning rate $\eta = 1e{-}3$ and batch size $bs = 64$ to optimize global text encoder and local image encoders. LoRA tuning is implemented with rank $r = 4$, starting from the layer $l = 10$. In the RL stage, we employ Gaussian noise with $\sigma = 0.1$ and perform $G = 3$ samplings per image with a clipping threshold $\epsilon = 0.2$ and $\beta = 0.5$. Details of $M$ refer to Appendix Section D.3. In *few-shot* settings, we conduct three trials in a 16-shot configuration and report the average test accuracy across datasets. More details of the dataset and implementation are provided in

*Table 1.* The average accuracy comparison results (%) across nine benchmarks under various federated data settings. The results of each dataset and more different shot numbers are shown in Appendix Section C.1.

| Few-shot | Non-IID | | | Dirichlet(0.1) | | | Dirichlet(0.3) | | | Dirichlet(0.5) | | | IID | | |
|---|---|---|---|---|---|---|---|---|---|---|---|---|---|---|---|
| | Local | Base | Novel | Local | Base | Novel | Local | Base | Novel | Local | Base | Novel | Local | Base | Novel |
| CLIP | 80.22 | 79.62 | 81.99 | 79.44 | 79.76 | 82.12 | 79.83 | 79.76 | 82.12 | 79.77 | 79.76 | 82.12 | 79.76 | 79.76 | 82.12 |
| pFedDC | 96.28 | 37.55 | 54.58 | 87.36 | 69.96 | 69.54 | 85.84 | 80.34 | 74.97 | 88.02 | 85.77 | 75.52 | 89.66 | 89.66 | 76.70 |
| FedPGP | 82.17 | 80.66 | 78.87 | 81.43 | 79.75 | 78.48 | 83.35 | 83.02 | 79.72 | 84.40 | 84.31 | 80.42 | 85.09 | 85.09 | 79.04 |
| pFedMMA | **97.23** | 56.55 | 72.47 | 87.13 | 72.77 | 74.91 | 86.33 | 81.45 | 76.69 | 88.52 | 86.39 | 76.04 | 90.68 | 90.68 | 76.25 |
| PromptFL | 82.69 | 82.23 | 79.84 | 82.02 | 80.41 | 76.09 | 83.79 | 83.40 | 80.06 | 85.85 | 85.61 | 79.76 | 86.24 | 86.24 | 80.52 |
| FedMaPLe | 83.89 | 83.63 | 77.56 | 85.70 | 84.05 | 77.69 | 88.47 | 88.18 | 80.40 | 90.40 | 90.18 | 80.01 | 90.85 | 90.85 | 81.65 |
| FedDTL | 89.76 | **89.58** | **83.01** | **91.66** | **90.95** | **82.64** | **92.18** | **91.94** | **83.61** | **92.17** | **92.02** | **83.25** | **92.58** | **92.58** | **82.53** |

| Full-data | Non-IID | | | Dirichlet(0.1) | | | Dirichlet(0.3) | | | Dirichlet(0.5) | | | IID | | |
|---|---|---|---|---|---|---|---|---|---|---|---|---|---|---|---|
| | Local | Base | Novel | Local | Base | Novel | Local | Base | Novel | Local | Base | Novel | Local | Base | Novel |
| CLIP | 80.22 | 79.62 | **81.99** | 79.44 | 79.76 | **82.12** | 79.83 | 79.76 | **82.12** | 79.77 | 79.76 | **82.12** | 79.76 | 79.76 | **82.12** |
| pFedDC | 97.44 | 28.75 | 40.16 | 80.12 | 60.08 | 63.55 | 83.37 | 77.51 | 74.37 | 86.25 | 83.87 | 77.46 | 91.51 | 91.51 | 79.82 |
| FedPGP | 83.74 | 48.26 | 49.02 | 74.62 | 65.23 | 57.61 | 78.32 | 76.38 | 66.17 | 82.77 | 82.04 | 66.37 | 87.76 | 87.76 | 78.10 |
| pFedMMA | **98.29** | 31.54 | 45.26 | 84.07 | 62.56 | 65.56 | 84.93 | 78.34 | 71.42 | 87.31 | 84.55 | 73.32 | 92.83 | 92.83 | 73.67 |
| PromptFL | 62.92 | 62.98 | 48.48 | 77.52 | 75.82 | 57.98 | 81.62 | 81.00 | 63.06 | 85.32 | 84.93 | 66.48 | 87.29 | 87.29 | 77.39 |
| FedMaPLe | 79.77 | 80.56 | 69.41 | 90.49 | 89.27 | 70.10 | 92.28 | 91.70 | 74.39 | 92.87 | 92.57 | 76.67 | 93.97 | 93.97 | 78.77 |
| FedDTL | 91.52 | **91.64** | 77.72 | **92.69** | **92.40** | 76.59 | **93.05** | **92.93** | 79.87 | **93.70** | **93.61** | 79.23 | **94.07** | **94.07** | 79.72 |

*Table 2.* The average domain accuracy comparison results (%) in the feature shift setting ("one") and the feature shift with label skew setting ("Dir(0.1)"). Detailed per-domain results and performance of different Dirichlet distributions are provided in Appendix Section C.2.

| Method | Office-Caltech10 | | | | DomainNet | | | |
|---|---|---|---|---|---|---|---|---|
| | Few-one | Few-Dir(0.1) | Full-one | Full-Dir(0.1) | Few-one | Few-Dir(0.1) | Full-one | Full-Dir(0.1) |
| CLIP | 92.88 | 92.88 | 92.88 | 92.88 | 87.68 | 87.68 | 87.68 | 87.68 |
| pFedDC | 96.54 | 72.01 | 97.40 | 70.71 | 85.69 | 69.31 | 87.11 | 59.87 |
| FedPGP | 97.89 | 97.75 | 98.64 | 96.12 | 88.98 | 89.01 | 89.08 | 87.70 |
| pFedMMA | 98.22 | 82.25 | 98.23 | 78.56 | 88.94 | 78.67 | 86.51 | 67.24 |
| PromptFL | 98.24 | 97.03 | 98.49 | 97.87 | 88.88 | 88.79 | 89.77 | 86.99 |
| FedMaPLe | **98.44** | 98.07 | 98.02 | 96.55 | 88.99 | 88.81 | 91.94 | 90.51 |
| FedDTL | 98.33 | **98.20** | **98.65** | **98.65** | **90.88** | **91.06** | **93.38** | **93.47** |

Appendix Section B.

## 4.2. Performance Evaluation

**An effective balance between global task adaptation and generalization in the label skew setting.** Unlike prior works that separate report top-1 accuracy of local, base, and novel classes, we jointly evaluate all three metrics at the final round to ensure a reliable assessment of the balance between global task adaptation and generalization, while also reflecting model stability under various FL settings. As shown in Table 1, FedDTL consistently achieves the best performance on base classes across all data distributions, especially in Non-IID settings, demonstrating strong global task adaptation. In novel classes, FedDTL remains competitive and ranks only second to zero-shot CLIP in *full-data* settings, indicating that FedDTL effectively preserves generalization. Although pFedDC and pFedMMA achieve higher accuracy on local classes in Non-IID settings, their global performance is comparatively lower. In contrast, FedDTL achieves a better balance performance across local, base, and novel classes, demonstrating its effective balance between global task adaptation and generalization.

**Stable and robust model training.** We also observe that

several baselines suffer from poor base accuracy under strong data heterogeneity and *full-data* settings, indicating that the parameter averaging paradigm is insufficient to resolve inter-client optimization inconsistency, degrading global task adaptation. Moreover, their novel accuracies in Non-IID and Dirichlet(0.1) settings decrease significantly when moving from *few-shot* to *full-data* regimes, suggesting that intra-client over-specialization is amplified and cannot be effectively mitigated by SFT-based regularization across all FL settings. In contrast, FedDTL consistently benefits from increased local data on base accuracy while maintaining small performance fluctuations on base and novel classes across different data settings, demonstrating stable and robust training with the decoupled encoder training and two-stage local fine-tuning.

**A robust balance between global task adaptation and generalization in the feature shift setting.** We next evaluate FedDTL under feature shift settings and summarize average results in Table 2. FedDTL achieves the highest average accuracy in almost all settings. Since the average accuracy aggregates performance across both in-user and out-of-user domains, these results indicate that FedDTL effectively balances global task adaptation and generalization

*Table 3.* The average accuracy (%) of ablation study on core modules over seven datasets. Per-dataset results in Appendix Section C.3.

| Core Module | Few_IID | | | Few_Non-IID | | | Full_IID | | | Full_Non-IID | | |
|---|---|---|---|---|---|---|---|---|---|---|---|---|
| | Base | Novel | HM | Base | Novel | HM | Base | Novel | HM | Base | Novel | HM |
| FedLoRA | 89.27 | 77.44 | 82.68 | 78.32 | 78.86 | 78.56 | 91.75 | 75.80 | 82.58 | 58.11 | 70.51 | 63.12 |
| + Decoupled Encoder Training | 89.34 | 75.35 | 81.33 | 86.42 | 79.52 | 82.60 | 91.42 | 73.36 | 80.93 | 86.68 | 73.57 | 79.20 |
| + Two-Stage Local Fine-Tuning | 90.48 | **83.11** | 86.46 | 79.46 | **83.84** | 81.47 | 91.71 | **81.86** | 86.25 | 47.91 | 76.43 | 57.86 |
| FedDTL | **91.76** | 82.48 | **86.59** | **90.06** | 83.58 | **86.51** | **92.75** | 81.29 | **86.35** | **90.58** | **80.62** | **85.03** |

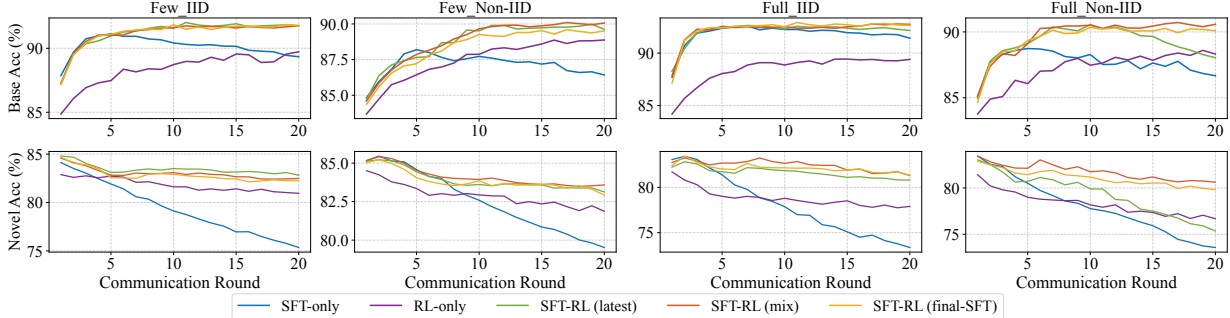

*Figure 2.* The average accuracy (%) of ablation study on local fine-tuning strategy and reference model choice over seven datasets under communication rounds. "SFT" and "RL" refer to our SFT stage and GRPO-inspired RL stage. Per-dataset results in Appendix Section C.3.

under feature-level distribution shifts, even in heterogeneous data settings.

## 4.3. Ablation Study

We perform ablation studies on seven datasets (CIFAR100, Food101, Tiny-ImageNet, OxfordPet, Flower102, Caltech101, Caltech256) under IID and Non-IID settings, and then report the average accuracy across datasets on base and novel classes with their harmonic mean (HM) to investigate how each module contributes to global task adaptation and generalization.

**Decoupled encoder training improves global task adaptation.** We first combine federated VLMs with LoRA tuning (Zanella & Ayed, 2024) as a baseline (FedLoRA). As shown in Table 3, the decoupled encoder training scheme significantly improves accuracy on base classes, particularly in Non-IID data settings (e.g., increased by 28.57% in *Full_Non-IID* setting). This improvement stems from global text encoder training and server-client modality alignment, which provides a coherent global semantic update, alleviating inter-client optimization inconsistency under heterogeneous data in both *few-shot* and *full-data* regimes.

**Two-stage local fine-tuning preserves generalization.** As shown in Table 3, two-stage local fine-tuning achieves higher novel accuracy than FedLoRA, indicating its effectiveness in alleviating intra-client over-specialization and preserving generalization across diverse FL scenarios. However, its base accuracy degrades in *Full_Non-IID* settings, suggesting that local strategy alone is insufficient to mitigate inter-client optimization inconsistency. Therefore, we

integrate the decoupled encoder training scheme with two-stage local fine-tuning, allowing the two components to complement each other to balance global task adaptation and generalization under various FL scenarios. As a result, FedDTL achieves the best HM performance.

**Different local fine-tuning strategies.** Figure 2 illustrates the impact of different local fine-tuning strategies. Under the *SFT-only* scheme, the base accuracy initially improves but degrades as training continues, while the novel accuracy consistently declines, indicating that SFT struggles to maintain stable training. In contrast, the *RL-only* variant shows continuous improvement on base classes and a slower degradation on novel classes compared to *SFT-only*. However, its base accuracy improves more slowly. Therefore, we propose a two-stage (*SFT-RL*) local fine-tuning that combines the complementary strengths of both paradigms. Specifically, the initial SFT stage serves as a fast and reliable warm-start, while the subsequent RL stage stabilizes local training and maintains generalization.

**Choice of reference model.** Our RL stage requires a reference model to stabilize policy optimization. We consider two options: 1) *latest* (the global policy parameters from the previous round), and 2) *final-SFT* (the final global SFT parameters). As shown in Figure 2, *latest* performs better in *few-shot* regimes, whereas *final-SFT* is more effective in *full-data* scenarios. Therefore, we construct a mixed reference model (*mix*) by integrating their parameters to make full use of their strengths across diverse FL settings.

**Impact of communication cost.** In *full-data* settings, uploading all image embeddings to the server inevitably increases the communication cost. Therefore, we study the

*Table 4.* The average accuracy (%) of more experimental analysis on privacy protection in uploading embeddings over nine base-to-novel datasets. Per-dataset results in Appendix Section D.1.

| Protection Method | Few_Dir(0.1) | | | Few_Non-IID | | | Full_Dir(0.1) | | | Full_Non-IID | | |
|---|---|---|---|---|---|---|---|---|---|---|---|---|
| | Base | Novel | HM | Base | Novel | HM | Base | Novel | HM | Base | Novel | HM |
| FedDTL | **90.95** | **82.64** | **86.32** | **89.58** | 83.01 | **85.97** | **92.40** | 76.59 | 82.34 | **91.64** | 77.72 | 83.12 |
| + Noise Perturbation | 89.69 | 82.17 | 85.56 | 88.77 | **83.23** | 85.11 | 91.80 | **77.86** | **83.35** | 91.44 | **79.04** | **84.15** |
| + Embedding Aggregation | 88.71 | 81.21 | 84.56 | 88.64 | 83.20 | 85.69 | 90.91 | 77.79 | 82.84 | 81.92 | 72.50 | 76.65 |

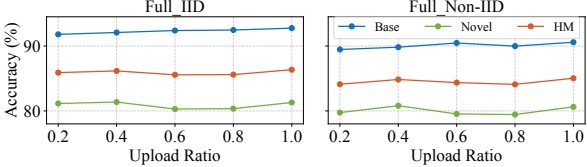

*Figure 3.* The average accuracy (%) of ablation study on different image embedding upload ratios over seven benchmarks. Per-dataset results in Appendix Section C.3.

impact of communication cost on model performance by varying the image embedding upload ratio from {0.2, 0.4, 0.6, 0.8, 1.0}. In Figure 3, different upload ratios have a small impact on model performance, indicating that FedDTL is robust to reduced communication costs, and balances global task adaptation and generalization even under partial embedding uploading.

### 4.4. More Experimental Analysis

**Privacy protection in uploading embeddings.** Since the server in our FedDTL only requires the visual class token embedding to update the global text encoder, clients upload normalized visual class token embeddings instead of full feature embeddings to the server, while keeping patch token embeddings local. Compared with patch token embeddings, which contain richer fine-grained private information, the class token embedding primarily captures high-level semantic information for classification, thereby reducing the exposure of fine-grained visual information and mitigating the risk of privacy leakage. To further enhance privacy protection under the embedding uploading, we investigate two simple privacy-preserving methods (details in Appendix Section D.1): 1) *noise perturbation*, which adds Gaussian noise to each uploaded class token embedding, and 2) *embedding aggregation*, which uploads a few aggregated embeddings for each class instead of all individual embeddings. Then, we evaluate both methods on nine base-to-novel datasets under the Dirichlet(0.1) and Non-IID settings, and report the average accuracy across datasets on base and novel classes with their harmonic mean. As shown in Table 4, both methods have competitive performance while providing additional privacy protection, especially with *noise perturbation*. Balancing privacy protection and model performance on FedDTL is an interesting direction for future research.

**More experiments.** Due to space limitations, additional experiments including the effect of different RL algorithms in our GRPO-inspired RL stage, the impact of varying client numbers, and hyperparameter ablation studies (LoRA rank $r$, LoRA starting layer $l$, RL sampling count $G$, noise scale $\sigma$ in our RL stage, and coefficient in RL loss $\beta$) are provided in Appendix Section D.

## 5. Conclusion

In this work, we propose FedDTL, a novel federated VLM framework that integrates a decoupled encoder training scheme and two-stage local fine-tuning to balance global task adaptation and generalization. Through decoupled encoder training and server-client modality alignment, FedDTL facilitates globally consistent representation learning and mitigates inter-client optimization inconsistency, thereby improving global task adaptation. Moreover, our FedDTL provides a fast and reliable warm-start via SFT-based task adaptation, upon which the subsequent GRPO-inspired RL stage alleviates intra-client over-specialization and enhances generalization. Extensive experiments across multiple datasets under diverse data distributions, including label skew and feature shift, demonstrate the effectiveness of our FedDTL in balancing global task adaptation and generalization under various FL data distributions in *few-shot* and *full-data* regimes.

## Acknowledgments

This work has been supported in part by the National Natural Science Foundation of China (Grant No. 62172383, No. 62472139, and No. 62231015), Anhui Provincial Key R&D Program (Grant No. S202103a05020098), Research Launch Project of University of Science and Technology of China (Grant No. KY0110000049), the Open Project Program of the State Key Laboratory of CAD&CG (Grant No. A2403), Zhejiang University.

## Impact Statement

This paper presents work whose goal is to advance the field of Machine Learning. There are many potential societal consequences of our work, none which we feel must be specifically highlighted here.

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

# A. Proposed Algorithm

For a better understanding of our proposed framework, we summarize the training details in Algorithm 1. The policy model $\pi_{\theta_k}^t$ shares the frozen backbone $\mathbf{W}_0$ and the LoRA parameters $\Delta \mathbf{W}_k^t$ of the local image encoder $\mathcal{V}_k^t$. For RL-based optimization, $\pi_{\theta_k}^t$ further adds a categorical probability distribution over logit, ensuring an effective sampling process and policy update with relative advantages during the RL stage. Finally, the model performance on global task adaptation and generalization is evaluated by the global image encoder $\mathcal{V}_g = [\mathbf{W}_0, \Delta \mathbf{W}_g^T]$ with the global text encoder $\mathcal{T}_g = [\mathbf{W}_{0,\text{text}}, \Delta \mathbf{W}_{\text{text}}^T]$. The details of $M$ can refer to Appendix Section D.3.

---

**Algorithm 1** FedDTL

---

**Input:** Global rounds $T$; Local epochs $T_e$; Number of clients $K$; Local datasets $\{\mathcal{D}_k\}_{k=1}^K$ with sizes $N_k$; Global image encoder $\mathcal{V}_g$; Local image encoder $\{\mathcal{V}_k\}_{k=1}^K$; Global text encoder $\mathcal{T}_g$; Gaussian noise $\varepsilon \sim \mathcal{N}(0, \sigma^2 I)$; Number of base classes $C$; Number of SFT rounds $M$.

**ServerExecute:**

> Initialization: load the frozen pre-trained visual parameter $\mathbf{W}_0$ with learnable LoRA parameters $\Delta \mathbf{W}_g^0$ to $\mathcal{V}_g$ and $\{\mathcal{V}_k\}_{k=1}^K$; load the frozen pre-trained textual parameter $\mathbf{W}_{0,\text{text}}$ with learnable LoRA parameters $\Delta \mathbf{W}_{\text{text}}^0$ to $\mathcal{T}_g$.
>
> **for** *communication round* $t = 1, \cdots, T$ **do**
>
>> Send global LoRA parameters $\Delta \mathbf{W}_g^{t-1}$ and text features $\{\bar{z}_{\text{text}}^{c,t-1}\}_{c=1}^C$ computed by $\mathcal{T}_g = [\mathbf{W}_{0,\text{text}}, \Delta \mathbf{W}_{\text{text}}^{t-1}]$ to all clients.
>>
>> **for** *client* $k = 1, \cdots, K$ **do**
>>
>>> **if** $t <= M$ **then**
>>>
>>>> $\{\Delta \mathbf{W}_k^t, \bar{\mathbf{z}}_{v,k}^t\} \leftarrow$ **LocalUpdate_SFT**$(k, \Delta \mathbf{W}_g^{t-1}, \{\bar{z}_{\text{text}}^{c,t-1}\}_{c=1}^C)$
>>>
>>> **else**
>>>
>>>> $\{\Delta \mathbf{W}_k^t, \bar{\mathbf{z}}_{v,k}^t\} \leftarrow$ **LocalUpdate_RL**$(k, \Delta \mathbf{W}_g^{t-1}, \{\bar{z}_{\text{text}}^{c,t-1}\}_{c=1}^C)$
>>
>> Aggregate global visual LoRA parameters: $\Delta \mathbf{W}_g^t \leftarrow \sum_{k=1}^K \frac{N_k}{\sum_{m=1}^K N_m} \Delta \mathbf{W}_k^t$.
>>
>> Collect $\{\bar{\mathbf{z}}_{v,k}^t\}_{k=1}^K$ as training samples to update $\Delta \mathbf{W}_{\text{text}}^t$ according to the global text objective (13).
>
> **return** $\Delta \mathbf{W}_g^T, \Delta \mathbf{W}_{text}^T$

**LocalUpdate_SFT**$(k, \Delta \mathbf{W}_g^{t-1}, \{\bar{z}_{text}^{c,t-1}\}_{c=1}^C)$:

> Initialize local LoRA parameters $\Delta \mathbf{W}_k^t \leftarrow \Delta \mathbf{W}_g^{t-1}$.
>
> **for** *SFT iteration* $e = 1, \cdots, T_e$ **do**
>
>> **foreach** *batch* $b \in \mathcal{D}_k$ **do**
>>
>>> Update $\Delta \mathbf{W}_k^t$ according to the SFT objective (4).
>
> Collect all image embeddings $\bar{\mathbf{z}}_{v,k}^t = \{\bar{z}_{v,k,i}^t\}_{i=1}^{N_k}$.
>
> **return** $\Delta \mathbf{W}_k^t, \bar{\mathbf{z}}_{v,k}^t$

**LocalUpdate_RL**$(k, \Delta \mathbf{W}_g^{t-1}, \{\bar{z}_{text}^{c,t-1}\}_{c=1}^C)$:

> Set policy model: $\pi_{\theta_k}^t \leftarrow [\mathbf{W}_0, \Delta \mathbf{W}_k^t]$ with a categorical probability distribution over logit.
>
> Initialize local LoRA parameters: $\Delta \mathbf{W}_k^t \leftarrow \Delta \mathbf{W}_g^{t-1}$.
>
> Set reference model: $\pi_{k,\text{ref}}^t \leftarrow [\mathbf{W}_0, \frac{1}{2}(\Delta \mathbf{W}_g^M + \Delta \mathbf{W}_g^{t-1})]$ with a categorical probability distribution over logit.
>
> **foreach** *batch* $b \in \mathcal{D}_k$ **do**
>
>> Set the old policy model $\pi_{\theta_{k,\text{old}}}^t \leftarrow \pi_{\theta_k}^t$.
>>
>> Sample $G$ actions $\{a_{i,j}^t\}_{j=1}^G \sim \pi_{\theta_{k,\text{old}}}^t(a_i^t | x_i, \varepsilon)$ for each image $x_i$.
>>
>> Compute advantages $A_{i,j}^t$ for each action $a_{i,j}^t$ by running Eq. (7).
>>
>> **for** *RL iteration* $e = 1, \cdots, T_e$ **do**
>>
>>> Update $\Delta \mathbf{W}_k^t$ according to the RL objective (11).
>
> Collect all image embeddings $\bar{\mathbf{z}}_{v,k}^t = \{\bar{z}_{v,k,i}^t\}_{i=1}^{N_k}$.
>
> **return** $\Delta \mathbf{W}_k^t, \bar{\mathbf{z}}_{v,k}^t$

---

# B. Experimental Setup

All experiments are performed using CLIP with a ViT-B/16 backbone and $\tau$=2.66 under NVIDIA GeForce RTX 4090 GPUs.

*Table 5.* Statistical details of datasets in experiments.

| Dataset | Class | Train | Train-base | Test | Domain |
|---|---|---|---|---|---|
| CIFAR10 (Krizhevsky et al., 2009) | 10 | 50000 | 25000 | 10000 | 1 |
| CIFAR100 (Krizhevsky et al., 2009) | 100 | 50000 | 25000 | 10000 | 1 |
| EuroSAT (Helber et al., 2019) | 10 | 21600 | 11200 | 5400 | 1 |
| Tiny-ImageNet (Deng et al., 2009) | 200 | 100000 | 50000 | 10000 | 1 |
| OxfordPet (Parkhi et al., 2012) | 37 | 3680 | 1785 | 3669 | 1 |
| Flower102 (Nilsback & Zisserman, 2008) | 102 | 7169 | 2983 | 1020 | 1 |
| Caltech101 (Fei-Fei et al., 2007) | 101 | 6907 | 4404 | 1770 | 1 |
| Caltech256 (Griffin et al., 2007) | 257 | 24385 | 11300 | 6222 | 1 |
| Food101 (Bossard et al., 2014) | 101 | 75750 | 37500 | 25250 | 1 |
| Office-Caltech10 (Gong et al., 2012) | 10 | 1969 | 1969 | 413 | 4 |
| DomainNet (Peng et al., 2019) | 10 | 15680 | 15680 | 6746 | 6 |

**Dataset Setup.** To evaluate the proposed framework, we conduct experiments on eleven visual classification benchmarks, covering generic object recognition, fine-grained recognition, remote sensing, food recognition, and cross-domain classification scenarios. CIFAR10 and CIFAR100 are coarse-grained generic image classification datasets, while OxfordPet and Flower102 are fine-grained classification benchmarks focusing on pet breeds and flower species, respectively. EuroSAT is a remote sensing dataset for land-use classification, and Food101 targets fine-grained food recognition. Caltech101 and Caltech256 are object recognition benchmarks with various categories, while Tiny-ImageNet is a large-scale generic image classification dataset. Office-Caltech10 and DomainNet are multi-domain datasets used for evaluating model performance in the feature shift settings. The first nine datasets are used for base-to-novel class generalization experiments, while the last two datasets are adopted for feature shift generalization experiments. Following the setting of prior works, we select 10 categories from the DomainNet dataset for evaluation. We adjust the data split of some datasets (Tiny-ImageNet, OxfordPet, Flower102) for better classification evaluation in *full-data* settings. For details, please refer to the dataset code files in the supplementary materials. Table 5 summarizes the number of classes, all training samples, training samples of base classes (Train-Base), all test samples, and domains for each dataset.

**Data Partition.** We consider 3 common data partition settings to simulate different FL settings: In *IID* data setting, each client receives samples uniformly drawn from the classes of the training dataset, resulting in identical label distributions across clients. In *Dirichlet($\alpha$)* data setting, each client samples local data from the classes of training samples according to a Dirichlet distribution with concentration parameter $\alpha$, leading to heterogeneous data distributions with overlapping labels among clients. A higher $\alpha$ means a more balanced data distribution. In *Non-IID* data setting, each client chooses a disjoint subset of the classes of training samples, resulting in highly heterogeneous data distributions with no overlapping labels among clients. In addition, we evaluate two different learning regimes: 1) *Few-shot* learning regime: Following the baseline settings, we adopt a 16-shot configuration, where each client has 16 labeled samples per class. 2) *Full-data* learning regime: According to the traditional federated learning paradigm, clients use all local samples for model training.

**Base-to-Novel Class Generalization.** In this task, classes of each dataset are equally split into base classes and novel classes. The base classes of training samples are distributed to all clients for training, while the novel classes are used exclusively for evaluation. In this experiment, we investigate three data partition settings in the *few-shot* and *full-data* regimes. Specifically, each client samples local data from base classes using IID, Dirichlet($\alpha$), or Non-IID data partition settings, where clients observe all base classes (IID), overlapping base classes with heterogeneous label distribution (Dirichlet), or disjoint subsets of base classes with equal category counts (Non-IID). Each client performs training on its local classes, i.e., the subset of base classes assigned to that client. During the evaluation process, we report model performance of test samples on three accuracy metrics: local classes, base classes, and novel classes (unseen in the whole training process and used to assess generalization ability). This evaluation is used to analyze the balance between global task adaptation and generalization while observing intra-client over-specialization in various FL data settings.

**Cross-Domain Feature Shift Generalization.** In this task, we consider both feature shift and feature shift with label skew scenarios. For feature shift, each client is assigned all data from a single domain, resulting in 4 clients for Office-Caltech10 and 6 clients for DomainNet. This setup simulates the feature shift across clients while keeping the label space consistent. We call this setting *one* setting. To simulate more realistic federated environments, we further introduce the label skew on top of the feature shift. Specifically, data from each domain is distributed to 3 clients, and data samples within the same domain are partitioned using IID or Dirichlet($\alpha$) distributions over labels. As a result, we obtain 12 and 18 clients for Office-Caltech10 and DomainNet datasets, respectively. We call this setting *IID* and *Dir($\alpha$)* setting. This setting captures

*Table 6.* The average accuracy comparison results (%) across nine benchmarks (CIFAR10, CIFAR100, EuroSAT, Tiny-ImageNet, OxfordPet, Flower102, Caltech101, Caltech256, Food101) under various federated data distributions in few-shot settings. Per-dataset results of 4-shot and 8-shot in Table 21, Table 22, Table 23.

| shot=4 | Non-IID | | | Dirichlet(0.1) | | | Dirichlet(0.3) | | | Dirichlet(0.5) | | | IID | | |
|---|---|---|---|---|---|---|---|---|---|---|---|---|---|---|---|
| | Local | Base | Novel | Local | Base | Novel | Local | Base | Novel | Local | Base | Novel | Local | Base | Novel |
| CLIP | 80.22 | 79.62 | 81.99 | 79.44 | 79.76 | **82.12** | 79.83 | 79.76 | 82.12 | 79.77 | 79.76 | 82.12 | 79.76 | 79.76 | 82.12 |
| pFedDC | 95.19 | 47.66 | 61.53 | 87.73 | 73.28 | 72.91 | 85.68 | 81.11 | 75.78 | 86.49 | 84.72 | 77.04 | 86.83 | 86.83 | 76.64 |
| FedPGP | 80.42 | 79.57 | 76.64 | 82.91 | 82.72 | 81.56 | 84.39 | 83.97 | 83.46 | 83.57 | 83.46 | 80.75 | 83.16 | 83.16 | 81.66 |
| pFedMMA | **95.92** | 68.46 | 77.76 | 88.14 | 79.56 | 79.57 | 85.29 | 82.72 | 80.28 | 85.29 | 83.97 | 80.19 | 85.58 | 85.58 | 79.89 |
| PromptFL | 80.99 | 80.37 | 81.63 | 81.11 | 80.37 | 78.86 | 82.36 | 82.12 | 82.08 | 82.00 | 82.05 | 80.59 | 82.98 | 82.98 | 81.40 |
| FedMaPLe | 84.97 | 84.47 | 81.58 | 85.23 | 84.67 | 80.46 | 85.39 | 85.15 | 81.92 | 87.26 | 87.20 | **83.05** | 86.69 | 86.69 | 82.76 |
| FedDTL | 88.03 | **87.85** | **82.87** | 88.73 | 87.33 | 81.60 | 89.71 | 89.25 | 84.28 | 89.79 | 89.50 | 82.45 | 89.83 | 89.83 | 84.12 |

| shot=8 | Non-IID | | | Dirichlet(0.1) | | | Dirichlet(0.3) | | | Dirichlet(0.5) | | | IID | | |
|---|---|---|---|---|---|---|---|---|---|---|---|---|---|---|---|
| | Local | Base | Novel | Local | Base | Novel | Local | Base | Novel | Local | Base | Novel | Local | Base | Novel |
| CLIP | 80.22 | 79.62 | 81.99 | 79.44 | 79.76 | 82.12 | 79.83 | 79.76 | 82.12 | 79.77 | 79.76 | 82.12 | 79.76 | 79.76 | **82.12** |
| pFedDC | 95.58 | 42.17 | 56.78 | 88.03 | 72.37 | 70.02 | 85.76 | 80.76 | 74.98 | 87.25 | 85.25 | 76.93 | 88.74 | 88.74 | 77.21 |
| FedPGP | 82.66 | 81.71 | 79.02 | 82.47 | 81.92 | 79.48 | 83.56 | 83.57 | 81.79 | 83.19 | 83.01 | 82.17 | 84.56 | 84.56 | 81.93 |
| pFedMMA | **96.51** | 63.74 | 75.24 | 87.11 | 75.62 | 77.28 | 86.28 | 82.54 | 79.13 | 86.91 | 85.12 | 79.12 | 88.29 | 88.29 | 78.65 |
| PromptFL | 82.01 | 81.60 | 80.03 | 83.44 | 82.23 | 78.88 | 84.37 | 84.07 | 80.11 | 85.90 | 85.74 | 81.06 | 85.86 | 85.86 | 79.19 |
| FedMaPLe | 84.92 | 84.62 | 80.09 | 85.52 | 84.44 | 78.16 | 86.83 | 86.68 | 81.73 | 88.77 | 88.60 | 80.87 | 89.61 | 89.61 | 81.61 |
| FedDTL | 88.48 | **88.04** | 82.83 | 90.62 | 89.71 | 82.29 | 90.88 | 90.49 | 83.79 | 90.87 | 90.70 | 83.27 | 91.94 | 91.94 | 82.05 |

| shot=16 | Non-IID | | | Dirichlet(0.1) | | | Dirichlet(0.3) | | | Dirichlet(0.5) | | | IID | | |
|---|---|---|---|---|---|---|---|---|---|---|---|---|---|---|---|
| | Local | Base | Novel | Local | Base | Novel | Local | Base | Novel | Local | Base | Novel | Local | Base | Novel |
| CLIP | 80.22 | 79.62 | 81.99 | 79.44 | 79.76 | 82.12 | 79.83 | 79.76 | 82.12 | 79.77 | 79.76 | 82.12 | 79.76 | 79.76 | 82.12 |
| pFedDC | 96.28 | 37.55 | 54.58 | 87.36 | 69.96 | 69.54 | 85.84 | 80.34 | 74.97 | 88.02 | 85.77 | 75.52 | 89.66 | 89.66 | 76.70 |
| FedPGP | 82.17 | 80.66 | 78.87 | 81.43 | 79.75 | 78.48 | 83.35 | 83.02 | 79.72 | 84.40 | 84.31 | 80.42 | 85.09 | 85.09 | 79.04 |
| pFedMMA | **97.23** | 56.55 | 72.47 | 87.13 | 72.77 | 74.91 | 86.33 | 81.45 | 76.69 | 88.52 | 86.39 | 76.04 | 90.68 | 90.68 | 76.25 |
| PromptFL | 82.69 | 82.23 | 79.84 | 82.02 | 80.41 | 76.09 | 83.79 | 83.40 | 80.06 | 85.85 | 85.61 | 79.76 | 86.24 | 86.24 | 80.52 |
| FedMaPLe | 83.89 | 83.63 | 77.56 | 85.70 | 84.05 | 77.69 | 88.47 | 88.18 | 80.40 | 90.40 | 90.18 | 80.01 | 90.85 | 90.85 | 81.65 |
| FedDTL | 89.76 | **89.58** | **83.01** | 91.66 | 90.95 | 82.64 | 92.18 | 91.94 | 83.61 | 92.17 | 92.02 | 83.25 | 92.58 | 92.58 | 82.53 |

both cross-domain feature heterogeneity and intra-domain label imbalance, posing a more complex federated learning scenario to evaluate global task adaptation and generalization.

**Ablation Study.** We conduct ablation studies on seven datasets (CIFAR100, Tiny-ImageNet, OxfordPet, Flower102, Caltech101, Caltech256, Food101). Experiments are performed on both IID and Non-IID settings, considering *few-shot* and *full-data* settings. These experiments aim to analyze the impact of the core components of our FedDTL on the balance between global task adaptation and generalization.

**More Experimental Analysis.** We conduct more experiments on base-to-novel datasets. Details of implementation and results in Appendix Section D. These experiments aim to further analyze the impact of our FedDTL on the balance between global task adaptation and generalization.

# C. Performance Evaluation and Ablation Study

## C.1. Base-to-Novel Class Generalization

**Per-Dataset Results.** The per-dataset comparison results of 16-shot and *full-data* with IID, Non-IID, Dirichlet(0.1), Dirichlet(0.3), and Dirichlet(0.5) can refer to Table 13, Table 14, Table 15, and Table 16.

**Shot Number.** Next, we study the effect of different few-shot settings by varying the shot $\in \{4, 8, 16\}$, and report the average results over nine benchmarks. As shown in Table 6, our FedDTL consistently achieves effective model performance on local, base, and novel classes under various data distributions. As the number of local samples increases, the base accuracy improves, while the novel accuracy remains stable, showing improved global task adaptation and stable generalization. These results indicate that our framework is robust under different *few-shot* settings.

**Model Performance Comparison.** We further perform new experiments to investigate the model performance on local, base, and novel classes across the communication rounds. Since zero-shot CLIP does not train the model, we do not conduct this comparison experiment. The average results across five datasets under four FL data settings are shown in Figure 4. Across all settings, the base accuracy in our FedDTL steadily improves and then stabilizes, while the accuracy degradation on novel classes is significantly slower as the communication round proceeds. This observation indicates that

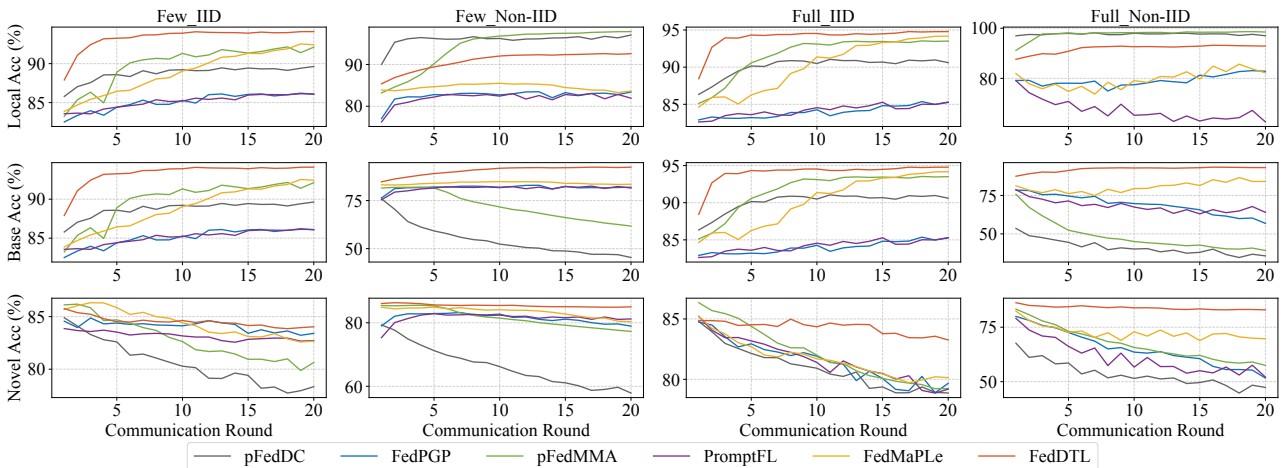

*Figure 4.* The average accuracy comparison results (%) across five datasets (OxfordPet, Flower102, CIFAR100, Caltech101, Caltech256) under communication rounds.

our decoupled encoder training and two-stage local fine-tuning can benefit the balance between global task adaptation and generalization in federated VLMs. We also observe that pFedDC achieves strong local accuracy in Non-IID settings because it focuses on local task adaptation on heterogeneous data distributions. However, as the communication round proceeds, its model performance on novel classes gradually decreases, especially in Non-IID settings. This indicates that these personalization-based methods tend to amplify intra-client over-specialization as local training is longer, and these methods are difficult to mitigate generalization degradation. Other baselines that rely on fully local SFT with regularization or alignment constraints exhibit less stable results on novel classes in *full-data* settings. Although these methods mitigate intra-client over-specialization in *few-shot* settings, the intra-client over-specialization can be amplified under *full-data* settings, degrading generalization. Moreover, these baselines, which rely on the strong local training with simple parameter aggregation, are insufficient to mitigate inter-client optimization inconsistency in heterogeneous data distributions, thereby degrading global task adaptation. We also note that pFedMMA achieves strong local accuracy in Non-IID settings. However, its accuracy on base classes performs poorly in Non-IID settings. We conjecture that this behavior is due to its limited parameter-sharing framework, which only shares parameters of projection layers. Such partial sharing may amplify inter-client optimization inconsistency as training proceeds, thereby hindering the accumulation of globally transferable knowledge. Overall, the full local SFT with simple parameter aggregation could amplify the inter-client inconsistency and intra-client over-specialization under heterogeneous data and *full-data* regimes, thereby decreasing the model performance of global task adaptation and generalization. In contrast, FedDTL addresses these challenges through the decoupled encoder training and two-stage local fine-tuning.

### C.2. Cross-Domain Feature Shift Generalization

**Per-Dataset Results.** The per-dataset comparison results of 16-shot and *full-data* with feature shift and feature shift with label skew (Dirichlet(0.1)) can refer to Table 17.

**More Performance Comparison.** We further perform new experiments on the DomainNet dataset to investigate model performance in the feature shift with label skew setting under different data distributions. We further introduce label skew (IID, Dirichlet(0.1), Dirichlet(0.3), Dirichlet(0.5)) on top of the feature shift. Data from each domain are distributed to three clients. As shown in Table 7, FedDTL performs a strong model performance in each domain and has the best average accuracy across all settings, indicating that our FedDTL can achieve effective generalization while maintaining strong global task adaptation. This observation further demonstrates the effectiveness of our decoupled encoder training and two-stage local fine-tuning on balancing global task generalization and generalization under feature-level data shift.

### C.3. Ablation Study

We perform the ablation study across seven benchmarks: OxfordPet, Flower102, CIFAR100, Caltech101, Caltech256, Tiny-ImageNet, Food101.

*Table 7.* The per-domain accuracy comparison results (%) of DomainNet dataset in feature shift with label skew setting (IID and different Dirichlet distributions) under few-shot and full-data settings.

| IID | DomainNet (Few-shot) | | | | | | DomainNet (Full-data) | | | | | |
|---|---|---|---|---|---|---|---|---|---|---|---|---|
| | Clipart | Infograph | Painting | Quickdraw | Real | Sketch | Avg. | Clipart | Infograph | Painting | Quickdraw | Real | Sketch | Avg. |
| CLIP | 96.45 | 75.33 | 96.10 | 62.93 | **97.73** | 97.56 | 87.68 | 96.45 | 75.33 | 96.10 | 62.93 | 97.73 | **97.56** | 87.68 |
| pFedDC | 94.61 | 73.10 | 92.05 | 62.71 | 95.77 | 94.10 | 85.39 | 94.84 | 73.14 | 92.98 | 63.43 | 95.68 | 94.30 | 85.73 |
| FedPGP | 96.70 | 77.23 | 96.06 | 70.03 | 97.20 | **97.78** | 89.17 | 96.64 | 78.00 | 96.08 | 70.95 | 97.27 | 97.41 | 89.39 |
| pFedMMA | 96.92 | 77.31 | 95.96 | 69.07 | 97.01 | 97.30 | 88.93 | 95.27 | 75.67 | 94.12 | 68.30 | 95.94 | 94.69 | 87.33 |
| PromptFL | 96.70 | 77.38 | 96.35 | 69.00 | 97.20 | 97.47 | 89.02 | 96.57 | 78.62 | 95.77 | 68.80 | 97.31 | 97.31 | 89.03 |
| FedMaPLe | 96.70 | 77.45 | 96.49 | 69.11 | 97.20 | 97.56 | 89.08 | **96.70** | 77.41 | 95.77 | 78.20 | 97.67 | 96.45 | 90.37 |
| FedDTL | **97.16** | **82.24** | **96.67** | **82.58** | 97.23 | 97.31 | **92.20** | 96.45 | **84.87** | **96.85** | **88.20** | **97.95** | 97.19 | **93.58** |

| Dir(0.1) | DomainNet (Few-shot) | | | | | | DomainNet (Full-data) | | | | | |
|---|---|---|---|---|---|---|---|---|---|---|---|---|
| | Clipart | Infograph | Painting | Quickdraw | Real | Sketch | Avg. | Clipart | Infograph | Painting | Quickdraw | Real | Sketch | Avg. |
| CLIP | 96.45 | 75.33 | 96.10 | 62.93 | **97.73** | 97.56 | 87.68 | 96.45 | 75.33 | 96.10 | 62.93 | 97.73 | **97.56** | 87.68 |
| pFedDC | 78.44 | 58.90 | 75.12 | 44.03 | 82.54 | 76.83 | 69.31 | 67.00 | 52.00 | 63.64 | 37.67 | 72.41 | 66.49 | 59.87 |
| FedPGP | 96.58 | 77.79 | 96.33 | 68.32 | 97.41 | **97.63** | 89.01 | 95.73 | 75.23 | 95.10 | 66.95 | 96.94 | 96.26 | 87.70 |
| pFedMMA | 88.46 | 65.74 | 87.53 | 48.33 | 93.43 | 88.54 | 78.67 | 74.07 | 56.96 | 73.85 | 41.54 | 82.21 | 74.80 | 67.24 |
| PromptFL | 96.57 | 77.01 | 96.06 | 68.64 | 97.21 | 97.23 | 88.79 | 96.07 | 77.19 | 95.66 | 59.87 | 97.29 | 95.84 | 86.99 |
| FedMaPLe | 96.66 | 77.26 | 96.24 | 68.00 | 97.38 | 97.31 | 88.81 | **96.95** | 76.21 | 94.90 | 81.27 | 97.29 | 96.45 | 90.51 |
| FedDTL | **96.70** | **80.15** | **96.42** | **78.42** | 97.53 | 97.15 | **91.06** | **96.95** | **84.10** | **96.85** | **87.60** | **98.01** | 97.31 | **93.47** |

| Dir(0.3) | DomainNet (Few-shot) | | | | | | DomainNet (Full-data) | | | | | |
|---|---|---|---|---|---|---|---|---|---|---|---|---|
| | Clipart | Infograph | Painting | Quickdraw | Real | Sketch | Avg. | Clipart | Infograph | Painting | Quickdraw | Real | Sketch | Avg. |
| CLIP | 96.45 | 75.33 | 96.10 | 62.93 | **97.73** | 97.56 | 87.68 | 96.45 | 75.33 | 96.10 | 62.93 | 97.73 | **97.56** | 87.68 |
| pFedDC | 85.40 | 64.76 | 81.53 | 51.95 | 88.83 | 83.98 | 76.08 | 80.91 | 63.87 | 77.22 | 50.90 | 83.97 | 79.14 | 72.67 |
| FedPGP | 96.37 | 77.78 | 96.20 | 67.93 | 97.25 | 97.35 | 88.81 | 96.04 | 77.64 | 95.40 | 59.90 | 97.15 | 96.23 | 87.06 |
| pFedMMA | 95.23 | 74.91 | 94.13 | 61.38 | 96.59 | 94.92 | 86.19 | 86.12 | 66.67 | 82.23 | 53.29 | 89.54 | 84.46 | 77.05 |
| PromptFL | 96.66 | 77.15 | 96.46 | 67.71 | 97.18 | 97.35 | 88.75 | 95.69 | 76.64 | 95.99 | 63.20 | 97.12 | 95.72 | 87.39 |
| FedMaPLe | 96.45 | 77.30 | 96.64 | 66.87 | 97.38 | **97.68** | 88.72 | 96.32 | 76.86 | 96.42 | 78.00 | 97.07 | 96.21 | 90.15 |
| FedDTL | **96.87** | **80.52** | **96.97** | **80.27** | 97.23 | 97.31 | **91.53** | **97.21** | **83.44** | **96.96** | **86.00** | **97.95** | 96.94 | **93.08** |

| Dir(0.5) | DomainNet (Few-shot) | | | | | | DomainNet (Full-data) | | | | | |
|---|---|---|---|---|---|---|---|---|---|---|---|---|
| | Clipart | Infograph | Painting | Quickdraw | Real | Sketch | Avg. | Clipart | Infograph | Painting | Quickdraw | Real | Sketch | Avg. |
| CLIP | 96.45 | 75.33 | 96.10 | 62.93 | **97.73** | 97.56 | 87.68 | 96.45 | 75.33 | 96.10 | 62.93 | 97.73 | **97.56** | 87.68 |
| pFedDC | 91.82 | 69.96 | 88.70 | 58.88 | 93.30 | 90.86 | 82.25 | 87.42 | 66.92 | 84.76 | 54.59 | 90.23 | 87.11 | 78.51 |
| FedPGP | 96.64 | 77.71 | **96.55** | 67.31 | 97.30 | 97.51 | 88.84 | 96.90 | 76.92 | 96.06 | 67.25 | 97.29 | 96.61 | 88.50 |
| pFedMMA | 95.65 | 75.15 | 94.21 | 64.66 | 96.60 | 95.79 | 87.01 | 91.86 | 71.22 | 89.77 | 60.28 | 94.16 | 91.20 | 83.08 |
| PromptFL | 96.62 | 76.79 | 95.84 | 68.86 | 97.23 | 97.43 | 88.80 | 96.57 | 78.73 | 95.55 | 66.93 | 97.40 | 96.82 | 88.67 |
| FedMaPLe | 96.66 | 77.19 | 96.31 | 67.69 | 97.25 | 97.39 | 88.75 | 96.45 | 78.51 | 95.99 | 78.20 | 97.23 | 96.09 | 90.41 |
| FedDTL | **97.08** | **81.62** | 96.28 | **80.76** | 97.23 | 97.02 | **91.67** | **97.08** | **83.44** | **97.29** | **87.73** | **97.79** | 97.07 | **93.40** |

*Table 8.* The average accuracy (%) of ablation study on further communication analysis over seven base-to-novel benchmarks under the full-data setting. Per-dataset results in Table 24.

| Method | IID | | | Dirichlet(0.1) | | | Non-IID | | |
|---|---|---|---|---|---|---|---|---|---|
| | Base | Novel | HM | Base | Novel | HM | Base | Novel | HM |
| FedDTL | **92.75** | **81.29** | **86.35** | **91.17** | 80.11 | 85.00 | **90.58** | **80.62** | **85.03** |
| + 16 embeddings per class | 92.47 | 81.20 | 86.17 | 90.45 | **81.60** | **85.58** | 89.52 | 80.42 | 84.52 |

**Core Module.** Table 18 shows the per-dataset model performance of different core modules across seven benchmarks under four data settings.

**Local Fine-Tuning Strategy and Reference Model Choice.** Figure 11, Figure 12, Figure 13, Figure 14 exhibit per-dataset accuracy of the ablation study on local fine-tuning strategies and reference model choices on base and novel classes over seven datasets in four different FL data settings.

**Communication Cost.** Table 19 shows the per-dataset accuracy of different image embedding upload ratios over seven benchmarks under *Full_IID* and *Full_Non-IID* data settings. The default setting is "ratio=1.0". To further reduce communication cost, we conduct an additional experiment: only 16 embeddings per class are randomly selected and uploaded to the server. The corresponding results in the *full-data* setting are in Table 8. Compared with the default setting that uploads all embeddings, the performance degradation is minimal on both base and novel classes. This observation indicates a trade-off between performance and communication in this setting. Under a fixed communication budget, exploring how to select better uploading embeddings for improving model performance is an interesting direction for future work.

*Table 9.* The average accuracy (%) of some baselines with our GRPO-inspired RL strategy across nine base-to-novel datasets under four data settings. "Few" and "Full" represent few-shot and full-data settings, while "Dir(0.1)" and "Non-IID" represent Dirichlet(0.1) and Non-IID data settings. Per-dataset results in Table 25.

| Method | Few_Dir(0.1) | | | Few_Non-IID | | | Full_Dir(0.1) | | | Full_Non-IID | | |
|---|---|---|---|---|---|---|---|---|---|---|---|---|
| | Base | Novel | HM | Base | Novel | HM | Base | Novel | HM | Base | Novel | HM |
| pFedDC | 69.96 | 69.54 | 69.75 | 37.55 | 54.58 | 44.49 | 60.08 | 63.55 | 61.77 | 28.75 | 40.16 | 33.51 |
| + our GRPO-inspired RL strategy | 76.06 ↑ | 77.68 ↑ | 76.78 ↑ | 60.46 ↑ | 74.27 ↑ | 65.67 ↑ | 69.08 ↑ | 71.55 ↑ | 70.18 ↑ | 56.74 ↑ | 62.56 ↑ | 59.31 ↑ |
| pFedMMA | 72.77 | 74.91 | 73.82 | 56.55 | 72.47 | 63.53 | 62.56 | 65.56 | 64.02 | 31.54 | 45.26 | 37.17 |
| + our GRPO-inspired RL strategy | 78.10 ↑ | 80.09 ↑ | 79.01 ↑ | 79.51 ↑ | 82.39 ↑ | 80.85 ↑ | 68.14 ↑ | 71.39 ↑ | 69.49 ↑ | 39.41 ↑ | 53.63 ↑ | 43.61 ↑ |
| FedMaPLe | 84.05 | 77.69 | 80.74 | 83.63 | 77.56 | 80.48 | 89.27 | 70.10 | 78.53 | 80.56 | 69.41 | 74.57 |
| + our GRPO-inspired RL strategy | 83.03 ↓ | 80.96 ↑ | 81.83 ↑ | 82.67 ↓ | 83.39 ↑ | 82.97 ↑ | 82.76 ↓ | 77.80 ↑ | 79.93 ↑ | 79.83 ↓ | 73.96 ↑ | 76.36 ↑ |

# D. More Experimental Analysis

We first investigate the privacy protection in uploading embeddings and the impact of our RL strategy on baselines under nine base-to-novel benchmarks, covering both *few-shot* and *full-data* settings under Dirichlet(0.1) and Non-IID data distributions. Then, we further investigate more ablation studies to analyze the impact of framework modules and hyperparameters on balancing global task adaptation and generalization. We perform these ablation studies on five benchmarks (OxfordPet, Flower102, CIFAR100, Caltech101, and Caltech256), covering both *few-shot* and *full-data* settings under IID and Non-IID data distributions.

## D.1. Privacy Protection in Uploading Embeddings

**Protection Method.** We investigate two simple privacy-preserving methods for uploading embeddings: 1) *noise perturbation*, which adds Gaussian noise $\varepsilon_{np}$ ($\sigma_{np} = 0.1$) to the uploaded class token embedding: $\bar{z}_v + \varepsilon_{np}$. Note that the noise $\varepsilon_{np}$ is different from the unbiased Gaussian noise $\varepsilon$ used in our RL stage. While $\varepsilon$ is injected into the latent embedding space for stochastic action sampling under the given image, the noise $\varepsilon_{np}$ applied to the uploaded visual class token embedding (pure embedding without injected noise $\varepsilon$) is to mitigate privacy leakage from uploading embeddings. 2) *Embedding aggregation*, which uploads a few aggregated embeddings for each class instead of all individual embeddings. All class token embeddings of the same class are randomly shuffled and partitioned into 4/8 groups. The embeddings within each group are then averaged to form a class-level aggregated embedding. We upload 4 aggregated embeddings per class for the *few-shot* setting and 8 aggregated embeddings per class for the *full-data* setting.

**Per-Dataset Results.** The per-dataset results of privacy protection in uploading embeddings across nine datasets (CIFAR10, CIFAR100, EuroSAT, Tiny-ImageNet, OxfordPet, Flower102, Caltech101, Caltech256, Food101) can refer to Table 20.

## D.2. The Impact of our RL Strategy on Baselines

Since baselines (e.g., pFedDC, pFedMMA, FedMaPLe) do not employ reinforcement learning, we conduct additional experiments to further investigate the impact of the RL strategy with these baselines on model performance. Directly porting RL to baselines is non-trivial since RL requires a stochastic policy net, while baselines use deterministic CLIP. Therefore, we apply our proposed GRPO-inspired RL strategy to these baselines. Results are shown in Table 9.

Compared with their original results, introducing our RL strategy could improve the novel accuracy, further demonstrating our effective RL strategy on generalization enhancement. Base accuracy variations suggest that the RL stage needs to be better redesigned into different FL objectives, which is an interesting direction for future work. This observation suggests that the RL module should be designed with specific optimization objectives.

## D.3. Stage Transition

*Our two-stage local training does not interleave executing SFT and RL within each global communication round. In early global rounds, clients perform SFT-based local optimization (SFT only) for fast and stable adaptation. After adaptation saturates, clients switch to our RL-based local generalization enhancement (RL only), without further SFT.* Therefore, the two-stage local fine-tuning design does not increase per-round local computation but accelerates model convergence via SFT warm-up and avoids long-term RL training, making it suitable for limited edge devices.

In practice, the number of SFT rounds $M$ is determined by a simple training stability criterion rather than a strategy to precisely find the optimal convergence point of SFT. Specifically, at communication round $t$, each client $k$ uploads its local

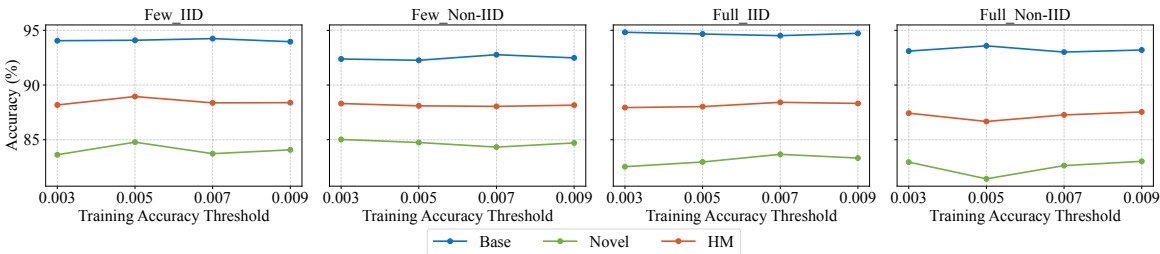

*Figure 5.* The average test accuracy (%) of different training accuracy threshold $\varepsilon_{acc}$ across five datasets (OxfordPet, Flower102, CIFAR100, Caltech101, and Caltech256) under four data settings. Per-dataset results in Table 26.

training accuracy $acc_k^t$ to the server, and the server computes the averaged accuracy $a\bar{c}c^t = \text{mean}(\{acc_1^t, \cdots, acc_K^t\})$ of all clients. The SFT stage is terminated when the fluctuation satisfies $|a\bar{c}c^t - a\bar{c}c^{t-1}| < \varepsilon_{acc}$ for $T_r$ consecutive rounds. This criterion does not aim to identify the optimal convergence point of SFT, but serves as a simple signal indicating that local task adaptation has saturated. Once the condition is met, the training process transitions to the subsequent RL stage to stabilize local fine-tuning and enhance generalization, preventing over-specialization. The simple training stability criterion is sufficient to perform stage transition and avoids additional computation overhead. Then, we investigate the effect of different training accuracy thresholds $\varepsilon_{acc}$ on model performance, and the results are shown in Figure 5.

We observe that varying $\varepsilon_{acc}$ can cause moderate changes in model performance on base and novel classes across all settings, indicating that the proposed transition criterion is not sensitive to the training accuracy threshold. As a result, the simple accuracy-based signal is sufficient for stage transition. Notably, the sample size, data distribution, and label number of different datasets are different, resulting in different levels of difficulty in model training. Therefore, a uniform training accuracy threshold is infeasible. Rather than tuning $\varepsilon_{acc}$ for each dataset, we select it according to a coarse-grained heuristic based on the dataset difficulty. We try to adopt a larger threshold (e.g., $\varepsilon_{acc} = 0.009$) in complex datasets (e.g., Food101, Tiny-ImageNet, CIFAR100, OxfordPet in the IID data setting), while a smaller threshold ($\varepsilon_{acc} = 0.001$) in CIFAR10 to ensure sufficient local adaptation without over-specialization. For other datasets, $\varepsilon_{acc} = 0.003$ is the default setting. Moreover, we set $T_r = 2$ as the default setting because SFT is a warm-start for the RL stage and is not suitable to set too many $T_r$. In addition, each client may observe only one class on CIFAR10 and EuroSAT under the Non-IID setting, we increase $T_r$ to 5 for more fully local knowledge sharing. We emphasize that this criterion is attempted as a simple solution rather than an optimal choice. Overall, these results demonstrate that our framework is robust to the training accuracy threshold in the stage transition, and the performance gains are primarily from our decoupled encoder training and two-stage local fine-tuning, rather than careful tuning of the stage transition. More sophisticated signals for stage transition may further stabilize the local fine-tuning and enhance generalization, and we leave this exploration for future work.

### D.4. RL Algorithm

We further investigate the effect of different reinforcement learning algorithms on model performance. Specifically, we replace GRPO-based RL loss with four representative algorithms: DR_GRPO (Liu et al., 2025), GMPO (Zhao et al., 2026), DAPO (Yu et al., 2025), LitePPO (Liu et al., 2026), while keeping the remaining training framework unchanged. Notably, the KL regularization term serves as a stability constraint in our controlled stochasticity mechanism. Therefore, we retain the KL regularization term $\mathbb{D}_{\text{KL}}$ and only replace the policy gradient term in the RL stage. This ensures a controlled comparison protocol, isolating the effect of different RL algorithms while keeping other components of the framework unchanged.

**DR_GRPO.** DR_GRPO removes two bias terms introduced in GRPO, namely the length and std normalization terms, to reduce potential pretraining biases in GRPO and achieve unbiased optimization. In our framework, where the policy model is based on the image encoder and each reward corresponds to an independent image sample, the length degenerates to the batch size $bs$. Accordingly, the new RL loss is formulated as:

$$\mathcal{L}_{rl} = -\frac{1}{G} \sum_{j=1}^{G} \sum_{i=1}^{bs} \left( \mathcal{L}_p - \beta \mathbb{D}_{\text{KL}} \right), \tag{14}$$

where the advantage in $\mathcal{L}_p$ is redefined as $A_{i,j} = r_{i,j} - \text{mean}_j(r_{i,j})$, $j = 1, 2, \cdots, G$.

**GMPO.** GMPO modifies the aggregation strategy of GRPO by replacing the arithmetic mean of rewards with a geometric

*Table 10.* The average accuracy (%) of different RL algorithms applied into our RL stage over five datasets (OxfordPet, Flower102, CIFAR100, Caltech101, and Caltech256) under four data settings. Per-dataset results in Table 27.

| RL Algorithm | Few_IID | | | Few_Non-IID | | | Full_IID | | | Full_Non-IID | | |
|---|---|---|---|---|---|---|---|---|---|---|---|---|
| | Base | Novel | HM | Base | Novel | HM | Base | Novel | HM | Base | Novel | HM |
| GRPO | 94.11 | 84.04 | 88.43 | 92.38 | **85.03** | **88.31** | **94.78** | 83.26 | **88.29** | 93.10 | **82.96** | **87.43** |
| DR_GRPO | **94.22** | 82.87 | 87.72 | 92.18 | 83.76 | 87.41 | 94.55 | 82.35 | 87.67 | **93.20** | 81.59 | 86.67 |
| GMPO | 94.14 | 83.27 | 87.99 | 92.25 | 84.79 | 88.16 | 94.53 | **83.27** | 88.24 | 93.00 | 81.90 | 86.67 |
| DAPO | 94.06 | 84.10 | 88.49 | **92.45** | 84.03 | 87.71 | 94.60 | 82.38 | 87.76 | 92.75 | 81.34 | 86.37 |
| LitePPO | 93.95 | **84.39** | **88.63** | **92.45** | 84.91 | 88.30 | 94.65 | 82.99 | 88.13 | 92.89 | 81.38 | 86.33 |

mean. For numerical stability, both the product and clipping operations are implemented in log space. Moreover, GMPO adopts asymmetric clipping thresholds to better control estimation during policy updating. The new RL loss is defined as:

$$\mathcal{L}_p = \sum_{j=1}^{G} \left( \prod_{i=1}^{bs} |\min[\rho_{i,j}A_{i,j}, \text{clip}(\rho_{i,j}, 1-\epsilon_{low}, 1+\epsilon_{high})A_{i,j}]| \right)^{\frac{1}{bs}} \cdot sgn(A_{i,j}), \tag{15}$$

$$\mathcal{L}_{rl} = -\frac{1}{G}(\mathcal{L}_p - \beta \sum_{j=1}^{G} \frac{1}{bs} \sum_{i=1}^{bs} \mathbb{D}_{\text{KL}}), \tag{16}$$

where $sgn(A_{i,j})$ ensures the correct optimization direction, returning 1 when $sgn(A_{i,j})$ is positive and $-1$ otherwise.

**DAPO.** DAPO introduces four key techniques over GRPO, including CLIP-Higher strategy, dynamic sampling, token-level policy gradient loss, and overlong reward shaping. In our framework, the policy model operates on independent images without variable-length sequence generation. Therefore, overlong reward shaping is infeasible. Moreover, dynamic sampling is designed to filter textual token prompts with zero reward, which is not suitable for our instance-wise optimization setting. As a result, we do not perform dynamic sampling. Then, DAPO replaces the fixed clipping threshold $\epsilon$ with a CLIP-Higher strategy to improve system exploration. The new RL loss is computed as:

$$\mathcal{L}_{rl} = -\frac{1}{\sum_{j=1}^{G} bs_j} \sum_{j=1}^{G} \sum_{i=1}^{bs_j} (\min[\rho_{i,j}A_{i,j}, \text{clip}(\rho_{i,j}, 1-\epsilon_{low}, 1+\epsilon_{high})A_{i,j}] - \beta\mathbb{D}_{\text{KL}}). \tag{17}$$

In our setting, each training step samples a batch of independent images and performs an instance-wise policy update. Therefore, the token-level loss degenerates to an instance-level loss.

**LitePPO.** LitePPO enhances GRPO by replacing group-level reward normalization with batch-level standard deviation and adopting instance-level loss aggregation. The new RL loss is formulated as:

$$\mathcal{L}_{rl} = -\frac{1}{\sum_{j=1}^{G} bs_j} \sum_{j=1}^{G} \sum_{i=1}^{bs_j} (\min[\rho_{i,j}A_{i,j}, \text{clip}(\rho_{i,j}, 1-\epsilon, 1+\epsilon)A_{i,j}] - \beta\mathbb{D}_{\text{KL}}), \tag{18}$$

where $A_{i,j} = (r_{i,j} - \text{mean}_j(r_{i,j}))/\text{std}_{batch}(r_{i,j})$.

**Experiments.** As reported in Table 10, DR_GRPO, GMPO, DAPO, and LitePPO achieve performance comparable to the GRPO-based RL on both base and novel classes, indicating that moderate modifications to reward normalization or loss aggregation do not significantly affect model performance on our CLIP-based RL stage under the small-scale and accessible FL benchmarks. In some settings, some RL algorithms even slightly outperform GRPO-based RL on base accuracy and novel accuracy. In the future, these RL algorithms can improve model performance with more suitable stochasticity mechanisms.

### D.5. Text Encoder Backbone

To investigate whether a more powerful text encoder can enhance model performance, we evaluate different text encoder backbones (ViT-B/32, ViT-B/16, ViT-L/14) while fixing the image encoder backbone as ViT-B/16. Moreover, due to the dimension mismatch in embedding space across different backbones, we add a linear projection layer to the text encoder for modality alignment. As shown in Table 11, stronger text encoders can improve base accuracy in some data settings (e.g.,

*Table 11.* The average accuracy (%) of different text encoder backbones over five datasets (OxfordPet, Flower102, CIFAR100, Caltech101, and Caltech256) under four data settings. Per-dataset results in Table 28.

| TE Backbone | Few_IID | | | Few_Non-IID | | | Full_IID | | | Full_Non-IID | | |
|---|---|---|---|---|---|---|---|---|---|---|---|---|
| | Base | Novel | HM | Base | Novel | HM | Base | Novel | HM | Base | Novel | HM |
| ViT-B/32 | **94.65** | 56.37 | 69.33 | 91.02 | 47.00 | 60.95 | **94.87** | 52.81 | 66.27 | **93.19** | 46.57 | 60.90 |
| ViT-L/14 | 94.36 | 54.20 | 67.84 | 90.44 | 47.64 | 61.51 | 94.82 | 54.26 | 67.65 | 93.01 | 46.73 | 61.63 |
| ViT-B/16 | 94.11 | **84.04** | **88.43** | 92.38 | **85.03** | **88.31** | 94.78 | **83.26** | **88.29** | 93.10 | **82.96** | **87.43** |

*Table 12.* The average accuracy (%) of different client number $K$ across four datasets (Flower102, CIFAR100, Caltech101, Caltech256) under four data settings. Per-dataset results in Table 29.

| Client Number | Few_IID | | | Few_Non-IID | | | Full_IID | | | Full_Non-IID | | |
|---|---|---|---|---|---|---|---|---|---|---|---|---|
| | Base | Novel | HM | Base | Novel | HM | Base | Novel | HM | Base | Novel | HM |
| $K$=5 | 93.64 | 80.80 | 86.41 | 91.66 | **82.20** | 86.43 | 94.42 | 79.75 | 86.18 | 92.62 | 79.56 | 85.34 |
| $K$=10 | 94.56 | 80.70 | 86.72 | 92.04 | 81.47 | 86.14 | 94.73 | 79.93 | 86.29 | 91.81 | 79.65 | 84.86 |
| $K$=15 | **94.70** | **81.20** | **87.14** | **93.03** | 81.60 | **86.58** | **95.11** | **80.71** | **86.98** | **94.90** | **79.95** | **86.41** |

ViT-L/14 in IID settings). However, the accuracy on novel classes with mismatched backbones (i.e., ViT-L/14 and ViT-B/32) declines significantly, demonstrating generalization degradation. This is because mismatched text encoder backbones may amplify intra-client over-specialization under the projection layer and disrupt the alignment embedding space, leading to reduced generalization on unseen classes. Therefore, even if the text encoder is stronger, such as ViT-L/14, misaligned backbones will lead generalization degradation, thus weakening the balance of global task adaptation and generalization. As a result, we adopt the same backbone of the text encoder and image encoder to facilitate modality alignment in the decoupled encoder scheme. However, we believe that the model performance of generalization may be stabilized with stronger text encoders by using advanced backbone alignment methods. We leave the interesting direction to future work.

### D.6. Client Number

We next investigate the effect of the number of clients by varying $K \in \{5, 10, 15\}$. As shown in Table 12, the increased number of clients improves model performance in most settings. In the *few-shot* setting, the improvement is because the increased total number of training samples is benefit for model optimization. In the *full-data* setting, although the total training samples across all clients remain unchanged, distributed data across more clients leads to more diverse local updates, which mitigate inter-client optimization inconsistency and benefits global training. In contrast, in the Non-IID with *few-shot* setting, we observe a slight degradation on novel classes under increased client number. However, the HM accuracy still improves, suggesting that the effective balance between global task adaptation and generalization. This further validates the effectiveness of our decoupled encoder training scheme and two-stage local fine-tuning under various FL scenarios.

### D.7. More Experiments of Hyperparameter

We further investigate the impact of different training hyper-parameters on global task adaptation and generalization under five datasets (OxfordPet, Flower102, CIFAR100, Caltech101, and Caltech256), including the LoRA rank $r$, the LoRA starting layer $l$, the RL sampling number $G$, the noise scale $\sigma$ in our RL stage, and the coefficient between $\mathcal{L}_p$ and $\mathbb{D}_{KL}$ in RL loss $\beta$.

**LoRA Rank.** We conduct the effect of different LoRA ranks $r$ by varying $r \in \{2, 4, 8, 16\}$, and the results are shown in Figure 6. We observe that the base accuracy and novel accuracy remain relatively stable across different LoRA ranks. We also observe slight fluctuations of novel accuracy in the *Full_Non-IID* setting. However, these fluctuations remain within a small range and do not influence performance balance. Considering the trade-off between model performance and computational cost, we adopt $r$=4 as the default value in all experiments.

**LoRA Starting Layer.** We next investigate the effect of different LoRA starting layers $l$ by varying $l \in \{2, 6, 10\}$, and the results are shown in Figure 7. We observe that the base accuracy and novel accuracy remain relatively stable across different LoRA starting layers. We also observe that a slight degradation of novel accuracy when applying LoRA tuning started from the 6-th layer in *full-data* settings. Considering the trade-off between generalization and computational cost, we adopt the deeper starting layer $l = 10$ as the default value in all experiments.

**RL Sampling Count.** Next, we conduct the effect of different sampling counts in the RL stage by varying $G \in \{2, 3, 5, 7\}$.

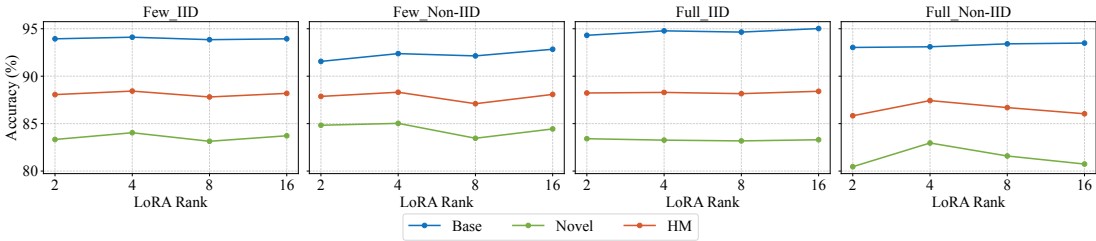

*Figure 6.* The average accuracy (%) of different LoRA ranks $r$ across five datasets under four data settings. Per-dataset results in Table 30.

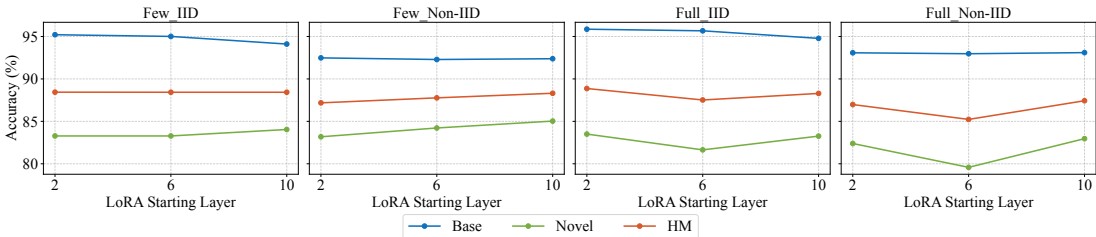

*Figure 7.* The average accuracy (%) of different LoRA starting layers $l$ across five datasets under four data settings. Per-dataset results in Table 31.

As shown in Figure 8, the model performance is relatively stable on base, novel, and HM in different data settings. We also observe that there are slight fluctuations of novel accuracy in the *Full_Non-IID* setting, but the overall trend remains consistent, suggesting that FedDTL is stable under different sampling counts within the evaluated range. Considering the balance between model performance and computational cost, we adopt $G = 3$ as the default value in all experiments.

**Noise Scale in our RL Stage.** We also investigate the effect of noise scale injected in our RL stage by varying $\sigma \in \{0.01, 0.05, 0.1, 0.2\}$, and the results are shown in Figure 9. We observe that the model performance on base, novel, and HM remains relatively stable across different noise scales. Although the accuracy varies within a tolerable range across different noise scales, we suggest injecting a small noise scale into the policy model to balance global task adaptation and generalization. Considering model performance and stable training, we adopt $\sigma = 0.1$ as the default value in all experiments.

**Coefficient in RL loss.** Finally, we study the effect of the loss coefficient $\beta$ of RL loss by varying it in $\{0.1, 0.3, 0.5, 0.7, 0.9\}$, and the results are shown in Figure 10. The model performance remains relatively stable across different $\beta$ within the evaluated range, indicating that the proposed framework can perform stable training with the KL regularization. We adopt $\beta = 0.5$ as the default value in all experiments. Because the KL regularization term is a part of RL optimization, we focus the ablation study on different meaningful values of $\beta$ while ensuring the KL regularization term exists.

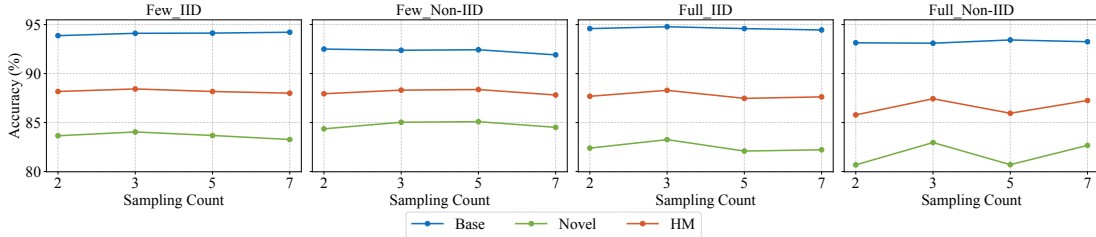

*Figure 8.* The average accuracy (%) of different sampling counts $G$ in the RL stage across five datasets under four data settings. Per-dataset results in Table 32.

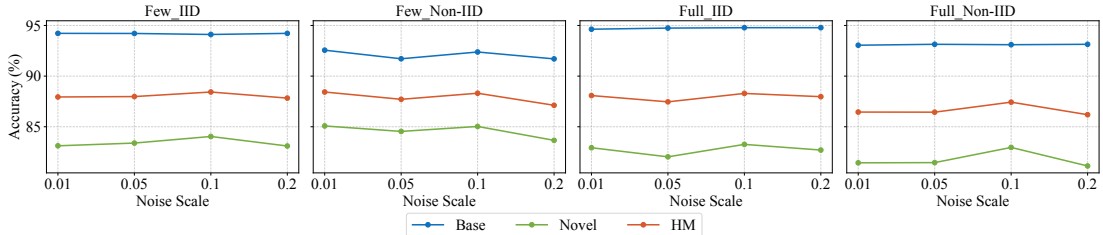

*Figure 9.* The average accuracy (%) of different noise scales $\sigma$ in the RL stage across five datasets under four data settings. Per-dataset results in Table 33.

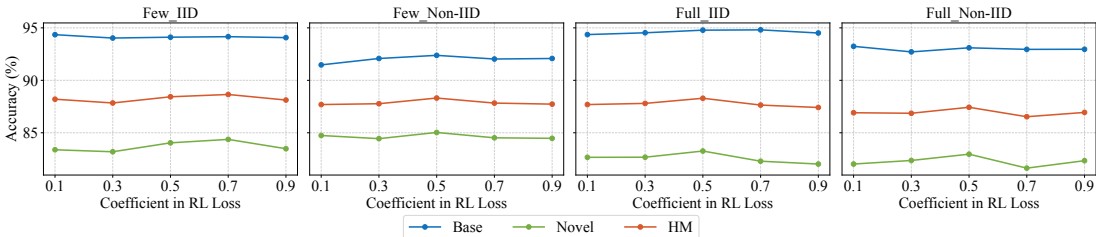

*Figure 10.* The average accuracy (%) of different coefficients $\beta$ in RL loss across five datasets under four data settings. Per-dataset results in Table 34.

# E. Full Experiment Results

*Table 13.* The per-dataset accuracy comparison results (%) under 16-shot with IID and Non-IID data distributions.

| IID | Average | | | CIFAR10 | | | CIFAR100 | | | EuroSAT | | | Tiny-ImageNet | | |
|---|---|---|---|---|---|---|---|---|---|---|---|---|---|---|---|
| | Local | Base | Novel | Local | Base | Novel | Local | Base | Novel | Local | Base | Novel | Local | Base | Novel |
| CLIP | 79.76 | 79.76 | 82.12 | 95.72 | 95.72 | 94.26 | 75.78 | 75.78 | 74.76 | 52.04 | 52.04 | 55.81 | 71.98 | 71.98 | 69.18 |
| pFedDC | 89.66 | 89.66 | 76.70 | 95.90 | 95.90 | 92.64 | 82.52 | 82.52 | 69.88 | 97.74 | 97.74 | 55.91 | 78.11 | 78.11 | 68.08 |
| FedPGP | 85.09 | 85.09 | 79.04 | 96.02 | 96.02 | 95.56 | 77.00 | 77.00 | 70.26 | 75.00 | 75.00 | 48.24 | 75.42 | 75.42 | 66.91 |
| pFedMMA | 90.68 | 90.68 | 76.25 | 97.11 | 97.11 | 96.64 | 82.84 | 82.84 | 67.62 | 93.98 | 93.98 | 43.37 | 75.94 | 75.94 | 59.03 |
| PromptFL | 86.24 | 86.24 | 80.52 | 96.15 | 96.15 | 94.35 | 77.59 | 77.59 | 69.34 | 81.86 | 81.86 | 64.19 | 75.44 | 75.44 | 67.76 |
| FedMaPLe | 90.85 | 90.85 | 81.65 | 97.07 | 97.07 | 95.83 | 85.79 | 85.79 | 73.79 | 91.02 | 91.02 | 66.57 | 81.58 | 81.58 | 70.79 |
| FedDTL | 92.58 | 92.58 | 82.53 | 96.95 | 96.95 | 97.49 | 85.09 | 85.09 | 77.18 | 93.92 | 93.92 | 67.91 | 80.15 | 80.15 | 68.22 |

| IID | OxfordPet | | | Flower102 | | | Caltech101 | | | Caltech256 | | | Food101 | | |
|---|---|---|---|---|---|---|---|---|---|---|---|---|---|---|---|
| | Local | Base | Novel | Local | Base | Novel | Local | Base | Novel | Local | Base | Novel | Local | Base | Novel |
| CLIP | 88.78 | 88.78 | 96.82 | 64.12 | 64.12 | 76.27 | 86.81 | 86.81 | 93.52 | 90.97 | 90.97 | 85.36 | 91.68 | 91.68 | 93.11 |
| pFedDC | 93.79 | 93.79 | 90.04 | 86.47 | 86.47 | 57.91 | 91.38 | 91.38 | 88.91 | 93.07 | 93.07 | 81.54 | 87.94 | 87.94 | 85.41 |
| FedPGP | 95.00 | 94.99 | 97.03 | 72.85 | 72.85 | 69.24 | 89.84 | 89.83 | 90.07 | 93.13 | 93.13 | 83.61 | 91.59 | 91.59 | 90.46 |
| pFedMMA | 95.65 | 95.65 | 96.95 | 93.62 | 93.62 | 62.30 | 93.46 | 93.46 | 92.25 | 93.11 | 93.11 | 80.92 | 90.37 | 90.37 | 87.21 |
| PromptFL | 95.32 | 95.32 | 95.30 | 75.36 | 75.36 | 67.25 | 89.42 | 89.42 | 92.44 | 93.47 | 93.47 | 83.96 | 91.56 | 91.56 | 90.13 |
| FedMaPLe | 95.79 | 95.79 | 96.93 | 88.63 | 88.63 | 64.25 | 91.24 | 91.24 | 92.18 | 94.47 | 94.47 | 83.83 | 92.03 | 92.03 | 90.70 |
| FedDTL | 95.98 | 95.98 | 97.00 | 97.19 | 97.19 | 67.32 | 97.42 | 97.42 | 94.08 | 94.88 | 94.88 | 84.61 | 91.60 | 91.60 | 88.98 |

| Non-IID | Average | | | CIFAR10 | | | CIFAR100 | | | EuroSAT | | | Tiny-ImageNet | | |
|---|---|---|---|---|---|---|---|---|---|---|---|---|---|---|---|
| | Local | Base | Novel | Local | Base | Novel | Local | Base | Novel | Local | Base | Novel | Local | Base | Novel |
| CLIP | 80.22 | 79.62 | 81.99 | 95.72 | 95.72 | 94.26 | 75.78 | 75.78 | 74.76 | 52.75 | 52.04 | 55.81 | 71.98 | 71.98 | 69.18 |
| pFedDC | 96.28 | 37.55 | 54.58 | 100.00 | 21.04 | 71.77 | 93.21 | 31.43 | 36.08 | 100.00 | 20.00 | 28.85 | 87.47 | 31.28 | 36.08 |
| FedPGP | 82.17 | 80.66 | 78.87 | 95.77 | 95.51 | 93.49 | 74.48 | 73.14 | 57.51 | 60.98 | 57.03 | 59.78 | 73.46 | 70.84 | 63.65 |
| pFedMMA | 97.23 | 56.55 | 72.47 | 99.79 | 82.13 | 93.20 | 95.03 | 52.42 | 64.73 | 99.99 | 27.94 | 44.40 | 89.63 | 37.68 | 45.80 |
| PromptFL | 82.69 | 82.23 | 79.84 | 95.70 | 95.70 | 96.03 | 75.98 | 75.98 | 68.40 | 71.18 | 69.82 | 57.98 | 70.47 | 70.47 | 63.27 |
| FedMaPLe | 83.89 | 83.63 | 77.56 | 96.22 | 96.22 | 96.47 | 80.22 | 80.22 | 75.68 | 75.38 | 74.27 | 54.42 | 68.26 | 68.26 | 49.49 |
| FedDTL | 89.76 | 89.58 | 83.01 | 96.61 | 96.61 | 97.31 | 82.21 | 82.21 | 77.90 | 79.83 | 79.12 | 64.72 | 77.10 | 77.10 | 68.99 |

| Non-IID | OxfordPet | | | Flower102 | | | Caltech101 | | | Caltech256 | | | Food101 | | |
|---|---|---|---|---|---|---|---|---|---|---|---|---|---|---|---|
| | Local | Base | Novel | Local | Base | Novel | Local | Base | Novel | Local | Base | Novel | Local | Base | Novel |
| CLIP | 86.70 | 86.84 | 96.71 | 64.20 | 64.20 | 75.38 | 91.70 | 86.81 | 93.52 | 91.47 | 91.55 | 85.21 | 91.68 | 91.68 | 93.11 |
| pFedDC | 98.53 | 46.63 | 77.97 | 99.20 | 28.40 | 41.42 | 97.60 | 63.40 | 76.43 | 96.12 | 62.14 | 62.84 | 94.40 | 33.63 | 59.80 |
| FedPGP | 91.90 | 91.27 | 94.79 | 66.53 | 65.71 | 71.94 | 92.07 | 89.10 | 93.72 | 92.42 | 91.97 | 84.54 | 91.89 | 91.38 | 90.42 |
| pFedMMA | 98.66 | 53.45 | 91.26 | 99.20 | 37.77 | 58.46 | 97.98 | 80.33 | 92.05 | 97.97 | 80.10 | 80.31 | 96.83 | 57.13 | 82.00 |
| PromptFL | 91.09 | 91.12 | 95.64 | 65.33 | 65.33 | 70.90 | 92.30 | 89.36 | 91.62 | 91.06 | 91.13 | 83.30 | 91.14 | 91.14 | 91.46 |
| FedMaPLe | 87.44 | 87.61 | 95.32 | 74.73 | 74.73 | 68.40 | 91.10 | 89.72 | 91.98 | 91.04 | 91.05 | 74.42 | 90.58 | 90.58 | 91.87 |
| FedDTL | 95.28 | 95.28 | 96.39 | 94.33 | 94.33 | 70.64 | 96.67 | 95.69 | 94.44 | 94.38 | 94.39 | 85.80 | 91.45 | 91.45 | 90.92 |

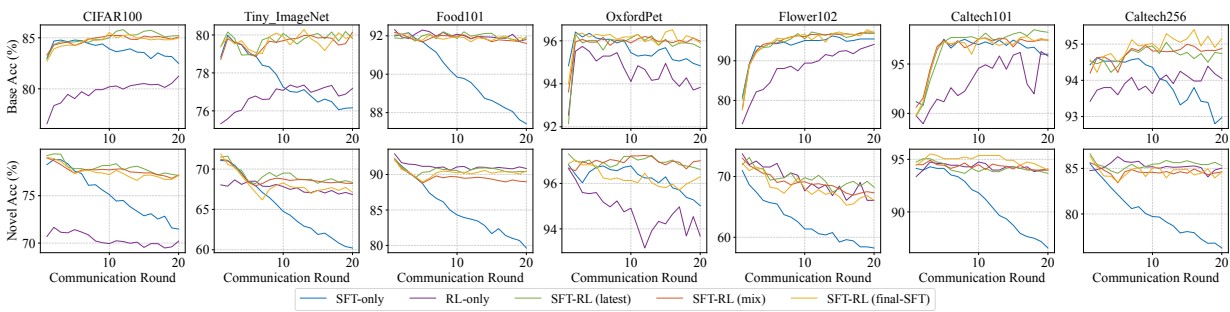

*Figure 11.* The per-dataset accuracy of ablation study on local fine-tuning strategies and reference model choices under few-shot with IID data setting.

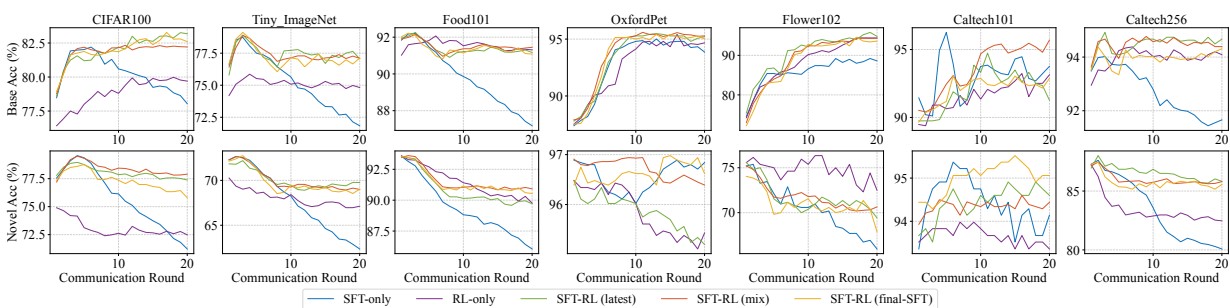

*Figure 12.* The per-dataset accuracy of ablation study on local fine-tuning strategies and reference model choices under few-shot with Non-IID data setting.

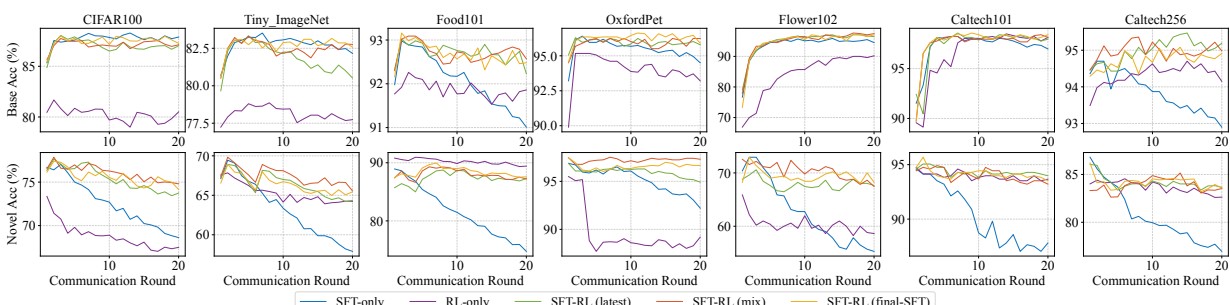

*Figure 13.* The per-dataset accuracy of ablation study on local fine-tuning strategies and reference model choices under full-data with IID data setting.

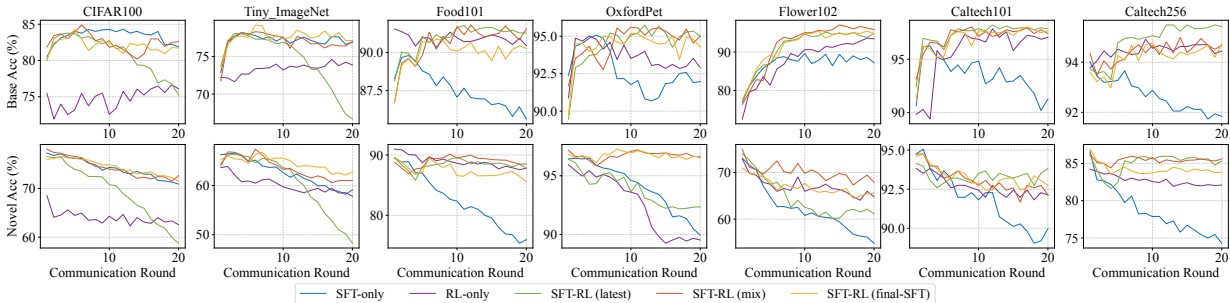

*Figure 14.* The per-dataset accuracy of ablation study on local fine-tuning strategies and reference model choices under full-data with Non-IID data setting.

*Table 14.* The per-dataset accuracy comparison results (%) under 16-shot with different Dirichlet data distributions.

| Dir(0.1) | Average | | | CIFAR10 | | | CIFAR100 | | | EuroSAT | | | Tiny-ImageNet | | |
|---|---|---|---|---|---|---|---|---|---|---|---|---|---|---|---|
| | Local | Base | Novel | Local | Base | Novel | Local | Base | Novel | Local | Base | Novel | Local | Base | Novel |
| CLIP | 79.44 | 79.76 | 82.12 | 95.75 | 95.72 | 94.26 | 76.17 | 75.78 | 74.76 | 49.79 | 52.04 | 55.81 | 71.80 | 71.98 | 69.18 |
| pFedDC | 87.36 | 69.96 | 69.54 | 95.32 | 87.57 | 92.05 | 81.39 | 62.40 | 59.61 | 87.43 | 58.84 | 40.94 | 74.94 | 60.98 | 56.02 |
| FedPGP | 81.43 | 79.75 | 78.48 | 96.06 | 96.04 | 96.29 | 73.58 | 70.87 | 62.25 | 55.05 | 47.23 | 51.60 | 73.24 | 71.67 | 62.42 |
| pFedMMA | 87.13 | 72.77 | 74.91 | 97.12 | 95.49 | 95.84 | 83.06 | 65.06 | 65.24 | 66.05 | 48.08 | 42.46 | 75.64 | 58.89 | 53.98 |
| PromptFL | 82.02 | 80.41 | 76.09 | 96.52 | 96.56 | 93.50 | 70.91 | 67.52 | 48.39 | 67.85 | 60.61 | 51.55 | 72.17 | 71.15 | 63.26 |
| FedMaPLe | 85.70 | 84.05 | 77.69 | 96.75 | 96.78 | 95.75 | 80.66 | 77.45 | 68.29 | 76.65 | 69.87 | 51.45 | 77.38 | 75.70 | 64.62 |
| FedDTL | 91.66 | 90.95 | 82.64 | 97.13 | 97.13 | 97.53 | 84.14 | 82.79 | 76.83 | 91.37 | 88.49 | 70.91 | 78.87 | 78.21 | 68.38 |

| Dir(0.1) | OxfordPet | | | Flower102 | | | Caltech101 | | | Caltech256 | | | Food101 | | |
|---|---|---|---|---|---|---|---|---|---|---|---|---|---|---|---|
| | Local | Base | Novel | Local | Base | Novel | Local | Base | Novel | Local | Base | Novel | Local | Base | Novel |
| CLIP | 89.79 | 88.78 | 96.82 | 63.39 | 64.12 | 76.27 | 86.16 | 86.81 | 93.52 | 90.81 | 90.97 | 85.36 | 91.27 | 91.68 | 93.11 |
| pFedDC | 92.24 | 67.83 | 89.48 | 86.99 | 52.60 | 50.76 | 93.72 | 82.52 | 83.59 | 90.05 | 82.59 | 74.44 | 84.15 | 74.30 | 78.97 |
| FedPGP | 89.96 | 89.06 | 94.62 | 70.87 | 69.10 | 72.23 | 90.73 | 90.52 | 93.64 | 92.26 | 92.01 | 83.15 | 91.11 | 91.21 | 90.11 |
| pFedMMA | 93.70 | 73.42 | 93.47 | 90.58 | 59.34 | 64.14 | 95.72 | 89.49 | 92.76 | 93.05 | 85.25 | 80.74 | 89.26 | 79.88 | 85.58 |
| PromptFL | 86.89 | 86.44 | 93.84 | 71.55 | 69.08 | 71.44 | 89.55 | 89.40 | 92.54 | 91.95 | 92.11 | 80.77 | 90.81 | 90.83 | 89.48 |
| FedMaPLe | 89.03 | 88.35 | 95.81 | 78.17 | 74.05 | 59.08 | 88.90 | 90.70 | 92.95 | 92.99 | 92.81 | 81.07 | 90.78 | 90.78 | 90.18 |
| FedDTL | 96.29 | 95.51 | 95.99 | 95.43 | 94.90 | 66.01 | 96.28 | 96.23 | 94.24 | 94.10 | 94.11 | 84.32 | 91.32 | 91.21 | 89.56 |

| Dir(0.3) | Average | | | CIFAR10 | | | CIFAR100 | | | EuroSAT | | | Tiny-ImageNet | | |
|---|---|---|---|---|---|---|---|---|---|---|---|---|---|---|---|
| | Local | Base | Novel | Local | Base | Novel | Local | Base | Novel | Local | Base | Novel | Local | Base | Novel |
| CLIP | 79.83 | 79.76 | 82.12 | 95.83 | 95.72 | 94.26 | 75.28 | 75.78 | 74.76 | 51.16 | 52.04 | 55.81 | 71.87 | 71.98 | 69.18 |
| pFedDC | 85.84 | 80.34 | 74.97 | 95.33 | 94.02 | 94.69 | 78.00 | 74.38 | 66.30 | 87.37 | 83.87 | 58.26 | 75.32 | 72.86 | 64.19 |
| FedPGP | 83.35 | 83.02 | 79.72 | 95.73 | 95.73 | 95.38 | 75.92 | 75.72 | 66.79 | 64.50 | 65.12 | 53.66 | 74.42 | 74.11 | 66.87 |
| pFedMMA | 86.33 | 81.45 | 76.69 | 96.82 | 96.36 | 96.68 | 78.76 | 75.32 | 67.29 | 79.28 | 77.07 | 50.96 | 72.36 | 68.49 | 56.63 |
| PromptFL | 83.79 | 83.40 | 80.06 | 96.35 | 96.34 | 96.36 | 75.51 | 75.22 | 69.45 | 68.02 | 68.25 | 54.92 | 74.25 | 73.99 | 66.95 |
| FedMaPLe | 88.47 | 88.18 | 80.40 | 96.85 | 96.89 | 96.73 | 84.51 | 84.27 | 76.33 | 73.79 | 73.99 | 52.62 | 80.36 | 80.07 | 70.13 |
| FedDTL | 92.18 | 91.94 | 83.61 | 96.88 | 96.93 | 97.82 | 84.38 | 84.17 | 76.11 | 93.37 | 93.25 | 78.31 | 79.39 | 79.03 | 68.02 |

| Dir(0.3) | OxfordPet | | | Flower102 | | | Caltech101 | | | Caltech256 | | | Food101 | | |
|---|---|---|---|---|---|---|---|---|---|---|---|---|---|---|---|
| | Local | Base | Novel | Local | Base | Novel | Local | Base | Novel | Local | Base | Novel | Local | Base | Novel |
| CLIP | 88.60 | 88.78 | 96.82 | 65.34 | 64.12 | 76.27 | 87.74 | 86.81 | 93.52 | 91.01 | 90.97 | 85.36 | 91.66 | 91.68 | 93.11 |
| pFedDC | 87.12 | 74.65 | 90.53 | 79.17 | 63.76 | 51.84 | 94.44 | 88.59 | 87.93 | 90.18 | 87.11 | 78.42 | 85.64 | 83.78 | 82.54 |
| FedPGP | 94.56 | 94.23 | 96.25 | 69.06 | 68.52 | 70.59 | 91.83 | 89.84 | 93.23 | 92.68 | 92.55 | 83.96 | 91.43 | 91.40 | 90.74 |
| pFedMMA | 90.20 | 81.87 | 96.21 | 88.39 | 72.14 | 63.49 | 94.47 | 90.38 | 92.86 | 91.80 | 88.84 | 81.62 | 84.85 | 82.56 | 84.44 |
| PromptFL | 93.87 | 93.32 | 95.37 | 71.41 | 70.26 | 71.63 | 91.14 | 89.81 | 93.11 | 92.20 | 92.04 | 82.45 | 91.38 | 91.40 | 90.28 |
| FedMaPLe | 92.96 | 92.18 | 96.03 | 90.08 | 89.15 | 64.51 | 91.65 | 91.39 | 93.11 | 94.16 | 93.93 | 83.50 | 91.86 | 91.79 | 90.68 |
| FedDTL | 95.23 | 94.78 | 96.64 | 96.45 | 96.08 | 67.58 | 97.28 | 96.88 | 94.29 | 94.90 | 94.69 | 84.49 | 91.74 | 91.64 | 89.20 |

| Dir(0.5) | Average | | | CIFAR10 | | | CIFAR100 | | | EuroSAT | | | Tiny-ImageNet | | |
|---|---|---|---|---|---|---|---|---|---|---|---|---|---|---|---|
| | Local | Base | Novel | Local | Base | Novel | Local | Base | Novel | Local | Base | Novel | Local | Base | Novel |
| CLIP | 79.77 | 79.76 | 82.12 | 95.72 | 95.72 | 94.26 | 75.72 | 75.78 | 74.76 | 52.04 | 52.04 | 55.81 | 71.92 | 71.98 | 69.18 |
| pFedDC | 88.02 | 85.77 | 75.52 | 93.86 | 93.86 | 93.02 | 80.35 | 79.06 | 67.16 | 97.48 | 97.48 | 53.73 | 76.79 | 75.77 | 65.72 |
| FedPGP | 84.40 | 84.31 | 80.42 | 96.13 | 96.13 | 93.58 | 76.66 | 76.60 | 68.25 | 71.51 | 71.51 | 63.68 | 75.12 | 75.09 | 67.85 |
| pFedMMA | 88.52 | 86.39 | 76.04 | 95.94 | 95.94 | 96.87 | 80.08 | 78.69 | 66.69 | 91.72 | 91.72 | 42.32 | 74.09 | 72.21 | 58.48 |
| PromptFL | 85.85 | 85.61 | 79.76 | 95.95 | 95.95 | 94.32 | 77.03 | 76.98 | 69.91 | 81.57 | 81.57 | 57.31 | 74.80 | 74.71 | 67.43 |
| FedMaPLe | 90.40 | 90.18 | 80.01 | 97.08 | 97.08 | 97.08 | 84.97 | 84.97 | 73.49 | 87.39 | 87.39 | 51.60 | 81.07 | 80.92 | 70.63 |
| FedDTL | 92.17 | 92.02 | 83.25 | 97.02 | 97.02 | 97.87 | 84.44 | 84.49 | 76.31 | 93.06 | 93.06 | 72.87 | 79.50 | 79.32 | 68.16 |

| Dir(0.5) | OxfordPet | | | Flower102 | | | Caltech101 | | | Caltech256 | | | Food101 | | |
|---|---|---|---|---|---|---|---|---|---|---|---|---|---|---|---|
| | Local | Base | Novel | Local | Base | Novel | Local | Base | Novel | Local | Base | Novel | Local | Base | Novel |
| CLIP | 88.39 | 88.78 | 96.82 | 64.51 | 64.12 | 76.27 | 87.15 | 86.81 | 93.52 | 90.88 | 90.97 | 85.36 | 91.64 | 91.68 | 93.11 |
| pFedDC | 90.75 | 85.01 | 91.12 | 81.27 | 73.77 | 55.85 | 93.67 | 90.49 | 88.52 | 91.14 | 90.35 | 80.59 | 86.89 | 86.10 | 83.95 |
| FedPGP | 90.20 | 90.29 | 93.56 | 75.46 | 74.48 | 70.89 | 90.09 | 90.31 | 93.90 | 92.66 | 92.61 | 82.84 | 91.80 | 91.80 | 89.23 |
| pFedMMA | 91.92 | 88.44 | 95.57 | 87.87 | 79.98 | 63.15 | 93.73 | 91.18 | 92.59 | 91.52 | 90.45 | 81.36 | 89.82 | 88.91 | 87.35 |
| PromptFL | 90.61 | 90.52 | 92.65 | 77.38 | 76.67 | 69.34 | 90.75 | 89.57 | 92.85 | 92.78 | 92.71 | 83.14 | 91.78 | 91.78 | 90.73 |
| FedMaPLe | 91.79 | 91.53 | 96.54 | 92.58 | 92.03 | 64.25 | 92.41 | 91.56 | 93.37 | 94.32 | 94.23 | 82.60 | 91.99 | 91.92 | 90.50 |
| FedDTL | 95.68 | 95.32 | 96.92 | 97.07 | 96.73 | 67.84 | 96.48 | 96.05 | 94.29 | 94.65 | 94.59 | 85.25 | 91.65 | 91.56 | 89.76 |

*Table 15.* The per-dataset accuracy comparison results (%) under full-data with IID and Non-IID data distributions.

| IID | Average | | | CIFAR10 | | | CIFAR100 | | | EuroSAT | | | Tiny-ImageNet | | |
|---|---|---|---|---|---|---|---|---|---|---|---|---|---|---|---|
| | Local | Base | Novel | Local | Base | Novel | Local | Base | Novel | Local | Base | Novel | Local | Base | Novel |
| CLIP | 79.76 | 79.76 | 82.12 | 95.72 | 95.72 | 94.26 | 75.78 | 75.78 | 74.76 | 52.04 | 52.04 | 55.81 | 71.98 | 71.98 | 69.18 |
| pFedDC | 91.51 | 91.51 | 79.82 | 97.86 | 97.86 | 96.90 | 85.73 | 85.73 | 75.90 | 98.96 | 98.96 | 64.12 | 79.97 | 79.97 | 70.68 |
| FedPGP | 87.76 | 87.76 | 78.10 | 96.96 | 96.96 | 95.23 | 79.51 | 79.51 | 66.53 | 93.32 | 93.32 | 74.15 | 76.96 | 76.96 | 58.80 |
| pFedMMA | 92.83 | 92.83 | 73.67 | 98.26 | 98.26 | 90.79 | 87.06 | 87.06 | 63.61 | 99.38 | 99.38 | 42.73 | 80.17 | 80.17 | 53.53 |
| PromptFL | 87.29 | 87.29 | 77.39 | 97.08 | 97.08 | 93.84 | 80.20 | 80.20 | 66.08 | 93.86 | 93.86 | 46.46 | 76.58 | 76.58 | 60.98 |
| FedMaPLe | 93.97 | 93.97 | 78.77 | 98.52 | 98.52 | 94.94 | 89.14 | 89.14 | 70.40 | 99.57 | 99.57 | 55.54 | 84.96 | 84.96 | 67.56 |
| FedDTL | 94.07 | 94.07 | 79.72 | 98.30 | 98.30 | 94.50 | 87.12 | 87.12 | 74.80 | 99.11 | 99.11 | 53.92 | 82.76 | 82.76 | 65.56 |

| IID | OxfordPet | | | Flower102 | | | Caltech101 | | | Caltech256 | | | Food101 | | |
|---|---|---|---|---|---|---|---|---|---|---|---|---|---|---|---|
| | Local | Base | Novel | Local | Base | Novel | Local | Base | Novel | Local | Base | Novel | Local | Base | Novel |
| CLIP | 88.78 | 88.78 | 96.82 | 64.12 | 64.12 | 76.27 | 86.81 | 86.81 | 93.52 | 90.97 | 90.97 | 85.36 | 91.68 | 91.68 | 93.11 |
| pFedDC | 92.02 | 92.02 | 92.72 | 87.57 | 87.57 | 55.33 | 96.70 | 96.70 | 89.44 | 93.33 | 93.33 | 82.65 | 91.42 | 91.42 | 90.60 |
| FedPGP | 95.59 | 95.59 | 94.28 | 69.57 | 69.57 | 60.71 | 92.00 | 92.00 | 89.60 | 93.46 | 93.46 | 81.87 | 92.50 | 92.50 | 81.76 |
| pFedMMA | 94.52 | 94.52 | 96.47 | 94.27 | 94.27 | 63.53 | 97.68 | 97.68 | 92.50 | 93.27 | 93.27 | 79.73 | 90.83 | 90.83 | 80.14 |
| PromptFL | 95.51 | 95.51 | 96.77 | 60.98 | 60.98 | 70.20 | 94.83 | 94.83 | 90.59 | 94.15 | 94.15 | 83.65 | 92.45 | 92.45 | 87.91 |
| FedMaPLe | 95.68 | 95.68 | 95.02 | 93.14 | 93.14 | 62.75 | 96.70 | 96.70 | 91.98 | 94.67 | 94.67 | 83.11 | 93.37 | 93.37 | 87.61 |
| FedDTL | 96.24 | 96.24 | 97.30 | 97.45 | 97.45 | 67.45 | 98.13 | 98.13 | 93.21 | 94.98 | 94.98 | 83.53 | 92.57 | 92.57 | 87.21 |

| Non-IID | Average | | | CIFAR10 | | | CIFAR100 | | | EuroSAT | | | Tiny-ImageNet | | |
|---|---|---|---|---|---|---|---|---|---|---|---|---|---|---|---|
| | Local | Base | Novel | Local | Base | Novel | Local | Base | Novel | Local | Base | Novel | Local | Base | Novel |
| CLIP | 80.22 | 79.62 | 81.99 | 95.72 | 95.72 | 94.26 | 75.78 | 75.78 | 74.76 | 52.75 | 52.04 | 55.81 | 71.98 | 71.98 | 69.18 |
| pFedDC | 97.44 | 28.75 | 40.16 | 100.00 | 20.00 | 31.64 | 95.90 | 28.60 | 33.39 | 100.00 | 20.00 | 24.18 | 91.16 | 21.26 | 24.05 |
| FedPGP | 83.74 | 48.26 | 49.02 | 99.58 | 33.32 | 56.95 | 84.12 | 31.98 | 25.15 | 51.03 | 20.40 | 25.45 | 82.98 | 25.56 | 23.05 |
| pFedMMA | 98.29 | 31.54 | 45.26 | 100.00 | 20.00 | 48.92 | 97.36 | 23.40 | 30.25 | 100.00 | 20.00 | 33.92 | 94.42 | 18.92 | 13.48 |
| PromptFL | 62.92 | 62.98 | 48.48 | 84.36 | 84.36 | 40.74 | 57.94 | 57.94 | 32.86 | 41.81 | 40.89 | 20.00 | 44.10 | 44.10 | 38.10 |
| FedMaPLe | 79.77 | 80.56 | 69.41 | 84.64 | 84.64 | 93.36 | 83.50 | 83.50 | 70.36 | 45.08 | 43.82 | 37.69 | 78.74 | 78.74 | 59.04 |
| FedDTL | 91.52 | 91.64 | 77.72 | 94.64 | 94.64 | 92.06 | 82.60 | 82.60 | 72.62 | 96.10 | 96.07 | 43.08 | 76.98 | 76.98 | 61.04 |

| Non-IID | OxfordPet | | | Flower102 | | | Caltech101 | | | Caltech256 | | | Food101 | | |
|---|---|---|---|---|---|---|---|---|---|---|---|---|---|---|---|
| | Local | Base | Novel | Local | Base | Novel | Local | Base | Novel | Local | Base | Novel | Local | Base | Novel |
| CLIP | 86.70 | 86.84 | 96.71 | 64.20 | 64.20 | 75.38 | 91.70 | 86.81 | 93.52 | 91.47 | 91.55 | 85.21 | 91.68 | 91.68 | 93.11 |
| pFedDC | 97.73 | 29.06 | 68.75 | 98.00 | 21.72 | 20.92 | 99.68 | 51.23 | 67.44 | 97.63 | 40.20 | 44.54 | 96.83 | 26.72 | 46.49 |
| FedPGP | 90.45 | 66.72 | 69.68 | 73.40 | 48.32 | 41.54 | 92.39 | 79.75 | 80.43 | 85.46 | 71.44 | 58.90 | 94.26 | 56.85 | 60.07 |
| pFedMMA | 98.33 | 45.38 | 90.51 | 99.40 | 29.16 | 47.23 | 99.58 | 71.85 | 81.91 | 98.12 | 35.63 | 41.10 | 97.43 | 19.49 | 20.04 |
| PromptFL | 65.69 | 65.72 | 71.74 | 47.60 | 47.60 | 58.27 | 83.46 | 84.94 | 77.16 | 69.35 | 69.28 | 47.89 | 71.98 | 71.98 | 49.54 |
| FedMaPLe | 85.56 | 85.76 | 76.22 | 83.60 | 83.60 | 61.15 | 80.38 | 88.41 | 75.00 | 86.73 | 86.82 | 70.63 | 89.74 | 89.74 | 81.27 |
| FedDTL | 95.01 | 95.01 | 96.57 | 95.80 | 95.80 | 67.88 | 96.35 | 97.42 | 92.13 | 94.62 | 94.67 | 85.59 | 91.56 | 91.56 | 88.50 |

*Table 16.* The per-dataset accuracy comparison results (%) under full-data with different Dirichlet data distributions.

| Dir(0.1) | Average | | | CIFAR10 | | | CIFAR100 | | | EuroSAT | | | Tiny-ImageNet | | |
|---|---|---|---|---|---|---|---|---|---|---|---|---|---|---|---|
| | Local | Base | Novel | Local | Base | Novel | Local | Base | Novel | Local | Base | Novel | Local | Base | Novel |
| CLIP | 79.44 | 79.76 | 82.12 | 95.75 | 95.72 | 94.26 | 76.17 | 75.78 | 74.76 | 49.79 | 52.04 | 55.81 | 71.80 | 71.98 | 69.18 |
| pFedDC | 80.12 | 60.08 | 63.55 | 93.19 | 82.42 | 89.85 | 74.22 | 49.88 | 55.87 | 77.41 | 51.64 | 34.87 | 59.76 | 39.90 | 40.37 |
| FedPGP | 74.62 | 65.23 | 57.61 | 94.85 | 93.38 | 93.82 | 52.82 | 31.98 | 24.92 | 77.30 | 64.01 | 33.81 | 62.03 | 47.42 | 38.04 |
| pFedMMA | 84.07 | 62.56 | 65.56 | 92.72 | 84.75 | 90.23 | 77.21 | 49.80 | 51.24 | 75.27 | 50.67 | 36.14 | 68.69 | 41.72 | 37.67 |
| PromptFL | 77.52 | 75.82 | 57.98 | 95.87 | 95.92 | 77.72 | 59.94 | 54.62 | 40.18 | 92.66 | 91.64 | 44.27 | 64.79 | 60.60 | 43.60 |
| FedMaPLe | 90.49 | 89.27 | 70.10 | 98.12 | 98.16 | 89.22 | 87.20 | 85.84 | 61.16 | 89.68 | 87.89 | 46.00 | 80.69 | 79.58 | 57.86 |
| FedDTL | 92.69 | 92.40 | 76.59 | 96.18 | 96.24 | 93.98 | 85.02 | 84.22 | 72.20 | 97.37 | 97.21 | 34.54 | 79.27 | 78.64 | 63.12 |

| Dir(0.1) | OxfordPet | | | Flower102 | | | Caltech101 | | | Caltech256 | | | Food101 | | |
|---|---|---|---|---|---|---|---|---|---|---|---|---|---|---|---|
| | Local | Base | Novel | Local | Base | Novel | Local | Base | Novel | Local | Base | Novel | Local | Base | Novel |
| CLIP | 89.79 | 88.78 | 96.82 | 63.39 | 64.12 | 76.27 | 86.16 | 86.81 | 93.52 | 90.81 | 90.97 | 85.36 | 91.27 | 91.68 | 93.11 |
| pFedDC | 88.56 | 56.46 | 77.74 | 80.95 | 47.88 | 44.94 | 91.90 | 79.36 | 83.61 | 88.25 | 79.49 | 73.14 | 66.84 | 53.66 | 71.56 |
| FedPGP | 92.29 | 90.08 | 88.52 | 53.09 | 48.55 | 44.71 | 89.49 | 87.45 | 89.88 | 86.14 | 75.31 | 53.48 | 63.61 | 48.89 | 51.31 |
| pFedMMA | 92.26 | 71.43 | 92.95 | 89.92 | 55.25 | 58.12 | 96.59 | 84.46 | 87.99 | 90.57 | 74.53 | 72.67 | 73.38 | 50.45 | 63.03 |
| PromptFL | 70.76 | 70.65 | 73.24 | 54.47 | 53.92 | 41.18 | 89.02 | 86.81 | 71.60 | 88.03 | 87.12 | 64.51 | 82.11 | 81.06 | 65.54 |
| FedMaPLe | 92.51 | 91.36 | 90.57 | 86.34 | 82.75 | 44.51 | 96.44 | 95.37 | 84.10 | 93.13 | 92.28 | 75.37 | 90.31 | 90.22 | 82.07 |
| FedDTL | 96.69 | 96.13 | 95.23 | 95.95 | 95.49 | 65.49 | 98.49 | 98.22 | 93.98 | 94.44 | 94.67 | 84.55 | 90.80 | 90.81 | 86.20 |

| Dir(0.3) | Average | | | CIFAR10 | | | CIFAR100 | | | EuroSAT | | | Tiny-ImageNet | | |
|---|---|---|---|---|---|---|---|---|---|---|---|---|---|---|---|
| | Local | Base | Novel | Local | Base | Novel | Local | Base | Novel | Local | Base | Novel | Local | Base | Novel |
| CLIP | 79.83 | 79.76 | 82.12 | 95.83 | 95.72 | 94.26 | 75.28 | 75.78 | 74.76 | 51.16 | 52.04 | 55.81 | 71.87 | 71.98 | 69.18 |
| pFedDC | 83.37 | 77.51 | 74.37 | 93.89 | 91.29 | 91.82 | 72.09 | 68.39 | 69.41 | 79.31 | 75.79 | 49.72 | 68.66 | 66.66 | 64.35 |
| FedPGP | 78.32 | 76.38 | 66.17 | 96.00 | 96.01 | 96.06 | 51.94 | 47.46 | 32.82 | 78.88 | 77.38 | 45.83 | 55.74 | 51.57 | 41.03 |
| pFedMMA | 84.93 | 78.34 | 71.42 | 94.99 | 93.16 | 92.06 | 74.39 | 68.93 | 58.62 | 84.74 | 81.23 | 42.59 | 67.69 | 61.64 | 49.03 |
| PromptFL | 81.62 | 81.00 | 63.06 | 96.34 | 96.48 | 59.58 | 66.56 | 65.42 | 46.82 | 86.16 | 85.86 | 72.85 | 70.29 | 69.28 | 43.20 |
| FedMaPLe | 92.28 | 91.70 | 74.39 | 98.54 | 98.56 | 96.62 | 87.41 | 86.96 | 67.52 | 98.92 | 98.89 | 40.04 | 83.05 | 82.72 | 59.00 |
| FedDTL | 93.05 | 92.93 | 79.87 | 96.98 | 97.06 | 94.76 | 85.75 | 85.70 | 73.32 | 96.51 | 96.64 | 58.58 | 80.26 | 80.02 | 63.74 |

| Dir(0.3) | OxfordPet | | | Flower102 | | | Caltech101 | | | Caltech256 | | | Food101 | | |
|---|---|---|---|---|---|---|---|---|---|---|---|---|---|---|---|
| | Local | Base | Novel | Local | Base | Novel | Local | Base | Novel | Local | Base | Novel | Local | Base | Novel |
| CLIP | 88.60 | 88.78 | 96.82 | 65.34 | 64.12 | 76.27 | 87.74 | 86.81 | 93.52 | 91.01 | 90.97 | 85.36 | 91.66 | 91.68 | 93.11 |
| pFedDC | 87.53 | 73.61 | 89.59 | 78.78 | 62.31 | 50.39 | 96.05 | 91.07 | 89.97 | 90.85 | 87.84 | 78.87 | 83.13 | 80.64 | 85.21 |
| FedPGP | 94.17 | 93.15 | 96.58 | 63.36 | 62.51 | 52.59 | 91.15 | 88.25 | 89.29 | 89.10 | 87.84 | 70.99 | 84.57 | 83.26 | 70.36 |
| pFedMMA | 87.84 | 78.65 | 95.93 | 87.63 | 70.20 | 60.51 | 97.39 | 91.98 | 91.91 | 89.91 | 85.15 | 76.12 | 79.78 | 74.13 | 76.00 |
| PromptFL | 89.44 | 90.12 | 85.43 | 55.60 | 55.29 | 45.69 | 90.48 | 87.88 | 83.80 | 91.92 | 91.24 | 71.02 | 87.83 | 87.44 | 59.13 |
| FedMaPLe | 90.73 | 89.00 | 94.33 | 91.88 | 90.98 | 57.65 | 93.61 | 92.34 | 90.59 | 94.29 | 93.91 | 79.30 | 92.09 | 91.94 | 84.43 |
| FedDTL | 96.14 | 96.02 | 97.62 | 96.38 | 96.08 | 65.88 | 98.26 | 97.95 | 94.44 | 95.30 | 95.12 | 85.27 | 91.83 | 91.75 | 85.22 |

| Dir(0.5) | Average | | | CIFAR10 | | | CIFAR100 | | | EuroSAT | | | Tiny-ImageNet | | |
|---|---|---|---|---|---|---|---|---|---|---|---|---|---|---|---|
| | Local | Base | Novel | Local | Base | Novel | Local | Base | Novel | Local | Base | Novel | Local | Base | Novel |
| CLIP | 79.77 | 79.76 | 82.12 | 95.72 | 95.72 | 94.26 | 75.72 | 75.78 | 74.76 | 52.04 | 52.04 | 55.81 | 71.92 | 71.98 | 69.18 |
| pFedDC | 86.25 | 83.87 | 77.46 | 95.00 | 95.00 | 94.72 | 77.73 | 76.71 | 71.51 | 94.25 | 94.25 | 63.22 | 72.17 | 71.16 | 66.36 |
| FedPGP | 82.77 | 82.04 | 66.37 | 96.83 | 96.83 | 95.42 | 65.42 | 63.98 | 41.04 | 91.12 | 91.12 | 37.83 | 67.48 | 66.45 | 38.98 |
| pFedMMA | 87.31 | 84.55 | 73.32 | 94.33 | 94.33 | 89.94 | 78.24 | 76.24 | 59.99 | 95.67 | 95.67 | 50.81 | 70.79 | 67.94 | 51.91 |
| PromptFL | 85.32 | 84.93 | 66.48 | 96.96 | 96.96 | 95.20 | 72.53 | 71.98 | 43.62 | 92.64 | 92.64 | 41.65 | 71.36 | 71.24 | 45.84 |
| FedMaPLe | 92.87 | 92.57 | 76.67 | 98.52 | 98.52 | 95.34 | 88.46 | 88.46 | 69.92 | 99.18 | 99.18 | 52.35 | 83.77 | 83.56 | 64.12 |
| FedDTL | 93.70 | 93.61 | 79.23 | 98.14 | 98.14 | 95.96 | 86.73 | 86.72 | 72.86 | 98.29 | 98.29 | 59.62 | 81.24 | 81.12 | 64.80 |

| Dir(0.5) | OxfordPet | | | Flower102 | | | Caltech101 | | | Caltech256 | | | Food101 | | |
|---|---|---|---|---|---|---|---|---|---|---|---|---|---|---|---|
| | Local | Base | Novel | Local | Base | Novel | Local | Base | Novel | Local | Base | Novel | Local | Base | Novel |
| CLIP | 88.39 | 88.78 | 96.82 | 64.51 | 64.12 | 76.27 | 87.15 | 86.81 | 93.52 | 90.88 | 90.97 | 85.36 | 91.64 | 91.68 | 93.11 |
| pFedDC | 88.29 | 81.90 | 90.77 | 80.20 | 72.08 | 52.71 | 94.97 | 91.84 | 88.67 | 90.82 | 89.95 | 82.05 | 82.81 | 81.98 | 87.10 |
| FedPGP | 89.30 | 89.45 | 92.64 | 63.48 | 61.84 | 52.82 | 91.09 | 89.57 | 90.09 | 92.63 | 92.47 | 79.96 | 87.61 | 86.66 | 68.53 |
| pFedMMA | 91.47 | 87.34 | 96.49 | 87.68 | 79.76 | 62.43 | 96.13 | 92.82 | 92.56 | 90.60 | 89.14 | 78.22 | 80.91 | 77.73 | 77.52 |
| PromptFL | 89.46 | 89.56 | 84.84 | 70.09 | 69.02 | 57.45 | 92.15 | 90.82 | 90.90 | 93.30 | 93.18 | 80.26 | 89.39 | 89.01 | 58.56 |
| FedMaPLe | 90.54 | 89.67 | 94.97 | 93.40 | 93.14 | 57.25 | 94.96 | 93.85 | 90.43 | 94.58 | 94.53 | 83.32 | 92.43 | 92.26 | 82.35 |
| FedDTL | 95.92 | 95.68 | 90.62 | 97.74 | 97.65 | 64.51 | 98.04 | 97.86 | 93.98 | 95.20 | 95.15 | 84.19 | 91.96 | 91.86 | 86.53 |

*Table 17.* The per-domain accuracy comparison results (%) in the feature shift setting ("one") and the feature shift with label skew (Dirichlet($\alpha$=0.1)) setting ("Dir(0.1)") under few-shot and full-data settings.

| Few-one | Office-Caltech10 | | | | | DomainNet | | | | | | |
|---|---|---|---|---|---|---|---|---|---|---|---|---|
| | Amazon | Caltech | Dslr | Webcam | Avg. | Clipart | Infograph | Painting | Quickdraw | Real | Sketch | Avg. |
| CLIP | 94.51 | 88.02 | 96.55 | 92.45 | 92.88 | 96.45 | 75.33 | 96.10 | 62.93 | **97.73** | 97.56 | 87.68 |
| pFedDC | 96.85 | 93.41 | 98.57 | 97.33 | 96.54 | 94.96 | 72.99 | 92.80 | 63.34 | 95.95 | 94.10 | 85.69 |
| FedPGP | 97.56 | **97.45** | 96.55 | **100.00** | 97.89 | 96.93 | 77.47 | 95.96 | 68.81 | 97.17 | 97.52 | 88.98 |
| pFedMMA | 98.37 | 95.66 | 99.14 | 99.69 | 98.22 | **97.02** | 77.47 | 96.08 | 68.83 | 97.10 | 97.15 | 88.94 |
| PromptFL | 97.15 | 95.81 | **100.00** | **100.00** | 98.24 | 96.53 | 77.74 | 95.99 | 68.18 | 97.21 | **97.64** | 88.88 |
| FedMaPLe | 97.36 | 96.41 | **100.00** | **100.00** | **98.44** | 96.83 | 77.85 | **96.82** | 67.76 | 97.29 | 97.39 | 88.99 |
| FedDTL | **99.39** | 95.81 | **100.00** | 98.11 | 98.33 | 96.70 | **80.70** | 96.13 | **77.87** | 96.84 | 97.03 | **90.88** |

| Full-one | Office-Caltech10 | | | | | DomainNet | | | | | | |
|---|---|---|---|---|---|---|---|---|---|---|---|---|
| | Amazon | Caltech | Dslr | Webcam | Avg. | Clipart | Infograph | Painting | Quickdraw | Real | Sketch | Avg. |
| CLIP | 94.51 | 88.02 | 96.55 | 92.45 | 92.88 | 96.45 | 75.33 | 96.10 | 62.93 | **97.73** | 97.56 | 87.68 |
| pFedDC | 97.10 | 94.31 | 99.14 | 99.06 | 97.40 | 95.56 | 74.91 | 93.66 | 67.49 | 96.11 | 94.91 | 87.11 |
| FedPGP | 97.56 | **97.01** | 100.00 | 100.00 | 98.64 | 96.63 | 77.08 | 96.24 | 70.67 | 97.37 | 96.48 | 89.08 |
| pFedMMA | 98.17 | 95.21 | **100.00** | 99.53 | 98.23 | 94.46 | 74.86 | 92.68 | 68.30 | 95.20 | 93.54 | 86.51 |
| PromptFL | 96.95 | **97.01** | 100.00 | 100.00 | 98.49 | 97.08 | 77.63 | 95.66 | 73.13 | 97.56 | **97.56** | 89.77 |
| FedMaPLe | 96.95 | **97.01** | 100.00 | 98.11 | 98.02 | **97.21** | 78.40 | **96.64** | 84.40 | 97.45 | **97.56** | 91.94 |
| FedDTL | **98.78** | 95.81 | 100.00 | 100.00 | **98.65** | 97.08 | **83.00** | **96.64** | **87.87** | **98.23** | 97.43 | **93.38** |

| Few-Dir(0.1) | Office-Caltech10 | | | | | DomainNet | | | | | | |
|---|---|---|---|---|---|---|---|---|---|---|---|---|
| | Amazon | Caltech | Dslr | Webcam | Avg. | Clipart | Infograph | Painting | Quickdraw | Real | Sketch | Avg. |
| CLIP | 94.51 | 88.02 | 96.55 | 92.45 | 92.88 | 96.45 | 75.33 | 96.10 | 62.93 | **97.73** | 97.56 | 87.68 |
| pFedDC | 73.20 | 68.46 | 73.95 | 72.43 | 72.01 | 78.44 | 58.90 | 75.12 | 44.03 | 82.54 | 76.83 | 69.31 |
| FedPGP | 97.05 | 95.72 | 98.95 | 99.27 | 97.75 | 96.58 | 77.79 | 96.33 | 68.32 | 97.41 | **97.63** | 89.01 |
| pFedMMA | 84.94 | 78.16 | 84.29 | 81.61 | 82.25 | 88.46 | 65.74 | 87.53 | 48.33 | 93.43 | 88.54 | 78.67 |
| PromptFL | 96.71 | 95.81 | 96.85 | 98.74 | 97.03 | 96.57 | 77.01 | 96.06 | 68.64 | 97.21 | 97.23 | 88.79 |
| FedMaPLe | 96.95 | **97.21** | 100.00 | 98.11 | 98.07 | 96.66 | 77.26 | 96.24 | 68.00 | 97.38 | 97.31 | 88.81 |
| FedDTL | **98.17** | 94.61 | **100.00** | **100.00** | **98.20** | **96.70** | **80.15** | 96.42 | **78.42** | 97.53 | 97.15 | **91.06** |

| Full-Dir(0.1) | Office-Caltech10 | | | | | DomainNet | | | | | | |
|---|---|---|---|---|---|---|---|---|---|---|---|---|
| | Amazon | Caltech | Dslr | Webcam | Avg. | Clipart | Infograph | Painting | Quickdraw | Real | Sketch | Avg. |
| CLIP | 94.51 | 88.02 | 96.55 | 92.45 | 92.88 | 96.45 | 75.33 | 96.10 | 62.93 | **97.73** | **97.56** | 87.68 |
| pFedDC | 72.15 | 66.22 | 71.84 | 72.64 | 70.71 | 67.00 | 52.00 | 63.64 | 37.67 | 72.41 | 66.49 | 59.87 |
| FedPGP | 95.42 | 94.36 | 99.71 | 94.97 | 96.12 | 95.73 | 75.23 | 95.10 | 66.95 | 96.94 | 96.26 | 87.70 |
| pFedMMA | 80.33 | 74.35 | 82.18 | 77.36 | 78.56 | 74.07 | 56.96 | 73.85 | 41.54 | 82.21 | 74.80 | 67.24 |
| PromptFL | 96.95 | **96.41** | 100.00 | 98.11 | 97.87 | 96.07 | 77.19 | 95.66 | 59.87 | 97.29 | 95.84 | 86.99 |
| FedMaPLe | 96.34 | 95.21 | 96.55 | 98.11 | 96.55 | **96.95** | 76.21 | 94.90 | 81.27 | 97.29 | 96.45 | 90.51 |
| FedDTL | **98.78** | 95.81 | **100.00** | **100.00** | **98.65** | **96.95** | **84.10** | **96.85** | **87.60** | **98.01** | 97.31 | **93.47** |

*Table 18.* The per-dataset accuracy (%) of ablation study on core modules under four data settings. "Few" and "Full" represent few-shot and full-data, while "IID" and "Non-IID" represent IID and Non-IID data setting.

| Few_IID | Average | | | CIFAR100 | | | Tiny-ImageNet | | | Food101 | | |
|---|---|---|---|---|---|---|---|---|---|---|---|---|
| | Base | Novel | HM | Base | Novel | HM | Base | Novel | HM | Base | Novel | HM |
| FedLoRA | 89.27 | 77.44 | 82.68 | 82.44 | 71.34 | 76.49 | 75.84 | 61.12 | 67.69 | 88.46 | 81.10 | 84.62 |
| + Decoupled Encoder Training | 89.34 | 75.35 | 81.33 | 82.50 | 71.48 | 76.60 | 76.16 | 60.22 | 67.26 | 87.42 | 79.64 | 83.35 |
| + Two-Stage Local Fine-Tuning | 90.48 | 83.11 | 86.46 | 83.82 | 76.76 | 80.13 | 79.06 | 68.82 | 73.59 | 91.14 | 90.64 | 90.89 |
| FedDTL | 91.76 | 82.48 | 86.59 | 85.09 | 77.18 | 80.94 | 80.15 | 68.22 | 73.71 | 91.60 | 88.98 | 90.27 |

| Few_IID | OxfordPet | | | Flower102 | | | Caltech101 | | | Caltech256 | | |
|---|---|---|---|---|---|---|---|---|---|---|---|---|
| | Base | Novel | HM | Base | Novel | HM | Base | Novel | HM | Base | Novel | HM |
| FedLoRA | 94.84 | 95.55 | 95.19 | 93.73 | 65.29 | 76.97 | 96.35 | 90.28 | 93.22 | 93.22 | 77.38 | 84.56 |
| + Decoupled Encoder Training | 94.84 | 95.02 | 94.93 | 95.49 | 58.24 | 72.35 | 95.99 | 86.57 | 91.04 | 92.97 | 76.27 | 83.80 |
| + Two-Stage Local Fine-Tuning | 95.74 | 97.03 | 96.38 | 93.33 | 70.78 | 80.51 | 97.06 | 93.52 | 95.26 | 93.18 | 84.19 | 88.46 |
| FedDTL | 95.98 | 97.00 | 96.49 | 97.19 | 67.32 | 79.54 | 97.42 | 94.08 | 95.72 | 94.88 | 84.61 | 89.45 |

| Few_Non-IID | Average | | | CIFAR100 | | | Tiny-ImageNet | | | Food101 | | |
|---|---|---|---|---|---|---|---|---|---|---|---|---|
| | Base | Novel | HM | Base | Novel | HM | Base | Novel | HM | Base | Novel | HM |
| FedLoRA | 78.32 | 78.86 | 78.56 | 77.16 | 73.24 | 75.15 | 62.62 | 60.90 | 61.75 | 82.66 | 85.90 | 84.25 |
| + Decoupled Encoder Training | 86.42 | 79.52 | 82.60 | 78.04 | 71.22 | 74.47 | 71.84 | 62.36 | 66.77 | 87.18 | 86.05 | 86.61 |
| + Two-Stage Local Fine-Tuning | 79.46 | 83.84 | 81.47 | 80.76 | 78.58 | 79.66 | 63.46 | 68.90 | 66.07 | 87.53 | 91.80 | 89.61 |
| FedDTL | 90.06 | 83.58 | 86.51 | 82.21 | 77.90 | 80.00 | 77.10 | 68.99 | 72.82 | 91.45 | 90.92 | 91.18 |

| Few_Non-IID | OxfordPet | | | Flower102 | | | Caltech101 | | | Caltech256 | | |
|---|---|---|---|---|---|---|---|---|---|---|---|---|
| | Base | Novel | HM | Base | Novel | HM | Base | Novel | HM | Base | Novel | HM |
| FedLoRA | 94.87 | 96.75 | 95.80 | 61.60 | 63.85 | 62.70 | 87.79 | 92.90 | 90.27 | 81.53 | 78.51 | 79.99 |
| + Decoupled Encoder Training | 93.86 | 96.84 | 95.33 | 88.60 | 65.96 | 75.62 | 93.76 | 94.14 | 93.95 | 91.66 | 80.08 | 85.48 |
| + Two-Stage Local Fine-Tuning | 89.00 | 97.03 | 92.84 | 55.00 | 69.42 | 61.37 | 91.71 | 94.75 | 93.21 | 88.76 | 86.38 | 87.55 |
| FedDTL | 95.28 | 96.39 | 95.83 | 94.33 | 70.64 | 80.78 | 95.69 | 94.44 | 95.06 | 94.39 | 85.80 | 89.89 |

| Full_IID | Average | | | CIFAR100 | | | Tiny-ImageNet | | | Food101 | | |
|---|---|---|---|---|---|---|---|---|---|---|---|---|
| | Base | Novel | HM | Base | Novel | HM | Base | Novel | HM | Base | Novel | HM |
| FedLoRA | 91.75 | 75.80 | 82.58 | 87.88 | 72.20 | 79.27 | 82.88 | 62.06 | 70.97 | 91.29 | 77.62 | 83.90 |
| + Decoupled Encoder Training | 91.42 | 73.36 | 80.93 | 87.88 | 68.62 | 77.04 | 82.26 | 57.90 | 67.93 | 91.01 | 74.76 | 82.09 |
| + Two-Stage Local Fine-Tuning | 91.71 | 81.86 | 86.25 | 87.24 | 75.24 | 80.80 | 81.08 | 66.74 | 73.21 | 90.51 | 88.11 | 89.29 |
| FedDTL | 92.75 | 81.29 | 86.35 | 87.12 | 74.80 | 80.49 | 82.76 | 65.56 | 73.16 | 92.57 | 87.21 | 89.81 |

| Full_IID | OxfordPet | | | Flower102 | | | Caltech101 | | | Caltech256 | | |
|---|---|---|---|---|---|---|---|---|---|---|---|---|
| | Base | Novel | HM | Base | Novel | HM | Base | Novel | HM | Base | Novel | HM |
| FedLoRA | 94.50 | 93.80 | 94.15 | 94.71 | 57.25 | 71.36 | 97.24 | 90.43 | 93.71 | 93.73 | 77.26 | 84.70 |
| + Decoupled Encoder Training | 94.50 | 92.21 | 93.34 | 94.51 | 55.29 | 69.77 | 96.97 | 87.81 | 92.16 | 92.90 | 76.96 | 84.18 |
| + Two-Stage Local Fine-Tuning | 95.74 | 97.62 | 96.67 | 94.31 | 67.84 | 78.91 | 98.04 | 92.59 | 95.24 | 95.02 | 84.85 | 89.65 |
| FedDTL | 96.24 | 97.30 | 96.77 | 97.45 | 67.45 | 79.72 | 98.13 | 93.21 | 95.61 | 94.98 | 83.53 | 88.89 |

| Full_Non-IID | Average | | | CIFAR100 | | | Tiny-ImageNet | | | Food101 | | |
|---|---|---|---|---|---|---|---|---|---|---|---|---|
| | Base | Novel | HM | Base | Novel | HM | Base | Novel | HM | Base | Novel | HM |
| FedLoRA | 58.11 | 70.51 | 63.12 | 47.80 | 64.38 | 54.86 | 43.94 | 52.82 | 47.97 | 43.01 | 69.79 | 53.22 |
| + Decoupled Encoder Training | 86.68 | 73.57 | 79.20 | 81.88 | 70.90 | 76.00 | 76.94 | 59.12 | 66.86 | 85.63 | 75.98 | 80.52 |
| + Two-Stage Local Fine-Tuning | 47.91 | 76.43 | 57.86 | 33.10 | 72.06 | 45.36 | 18.92 | 56.40 | 28.33 | 44.38 | 77.84 | 56.53 |
| FedDTL | 90.58 | 80.62 | 85.03 | 82.60 | 72.62 | 77.29 | 76.98 | 61.04 | 68.09 | 91.56 | 88.50 | 90.00 |

| Full_Non-IID | OxfordPet | | | Flower102 | | | Caltech101 | | | Caltech256 | | |
|---|---|---|---|---|---|---|---|---|---|---|---|---|
| | Base | Novel | HM | Base | Novel | HM | Base | Novel | HM | Base | Novel | HM |
| FedLoRA | 92.98 | 91.59 | 92.28 | 61.80 | 55.00 | 58.20 | 65.15 | 86.88 | 74.46 | 52.09 | 73.11 | 60.84 |
| + Decoupled Encoder Training | 91.97 | 89.94 | 90.94 | 87.20 | 54.81 | 67.31 | 91.27 | 89.97 | 90.62 | 91.84 | 74.28 | 82.13 |
| + Two-Stage Local Fine-Tuning | 88.80 | 96.62 | 92.55 | 42.20 | 61.54 | 50.07 | 57.22 | 88.12 | 69.39 | 50.73 | 82.44 | 62.81 |
| FedDTL | 95.01 | 96.57 | 95.78 | 95.80 | 67.88 | 79.46 | 97.42 | 92.13 | 94.70 | 94.67 | 85.59 | 89.90 |

*Table 19.* The per-dataset accuracy (%) of ablation study on different image embedding upload ratios under two data settings.

| Full_IID | OxfordPet | | | Flower102 | | | CIFAR100 | | | Caltech101 | | | Caltech256 | | | Food101 | | | Tiny-ImageNet | | |
|---|---|---|---|---|---|---|---|---|---|---|---|---|---|---|---|---|---|---|---|---|---|
| | Base | Novel | HM | Base | Novel | HM | Base | Novel | HM | Base | Novel | HM | Base | Novel | HM | Base | Novel | HM | Base | Novel | HM |
| ratio=0.2 | 95.12 | 96.61 | 95.86 | 93.33 | 67.06 | 78.04 | 87.14 | 74.48 | 80.31 | 97.95 | 91.20 | 94.45 | 94.63 | 84.10 | 89.05 | 92.53 | 88.06 | 90.24 | 81.82 | 66.50 | 73.37 |
| ratio=0.4 | 96.13 | 97.19 | 96.66 | 93.92 | 67.84 | 78.78 | 86.90 | 74.82 | 80.41 | 98.22 | 91.82 | 94.91 | 94.67 | 84.79 | 89.46 | 92.32 | 87.27 | 89.72 | 82.38 | 65.82 | 73.17 |
| ratio=0.6 | 95.74 | 96.98 | 96.36 | 95.69 | 62.75 | 75.80 | 87.24 | 74.02 | 80.09 | 98.40 | 91.98 | 95.08 | 94.53 | 83.35 | 88.59 | 92.48 | 87.10 | 89.71 | 82.56 | 65.88 | 73.28 |
| ratio=0.8 | 96.30 | 96.24 | 96.27 | 97.25 | 62.35 | 75.98 | 87.38 | 74.42 | 80.38 | 97.68 | 93.36 | 95.47 | 94.63 | 84.01 | 89.00 | 92.41 | 86.85 | 89.54 | 81.60 | 65.20 | 72.48 |
| ratio=1.0 | 96.24 | 97.30 | 96.77 | 97.45 | 67.45 | 79.72 | 87.12 | 74.80 | 80.49 | 98.13 | 93.21 | 95.61 | 94.98 | 83.53 | 88.89 | 92.57 | 87.21 | 89.81 | 82.76 | 65.56 | 73.16 |

| Full_Non-IID | OxfordPet | | | Flower102 | | | CIFAR100 | | | Caltech101 | | | Caltech256 | | | Food101 | | | Tiny-ImageNet | | |
|---|---|---|---|---|---|---|---|---|---|---|---|---|---|---|---|---|---|---|---|---|---|
| | Base | Novel | HM | Base | Novel | HM | Base | Novel | HM | Base | Novel | HM | Base | Novel | HM | Base | Novel | HM | Base | Novel | HM |
| ratio=0.2 | 95.21 | 96.57 | 95.89 | 91.00 | 66.54 | 76.87 | 81.88 | 72.42 | 76.86 | 97.06 | 91.98 | 94.45 | 94.92 | 83.20 | 88.67 | 90.45 | 86.01 | 88.17 | 75.70 | 61.44 | 67.83 |
| ratio=0.4 | 94.87 | 97.44 | 96.14 | 94.00 | 69.04 | 79.61 | 82.80 | 73.16 | 77.68 | 95.99 | 92.28 | 94.10 | 94.77 | 85.24 | 89.75 | 90.06 | 86.61 | 88.30 | 76.24 | 61.80 | 68.26 |
| ratio=0.6 | 95.28 | 95.52 | 95.40 | 95.20 | 64.62 | 76.98 | 82.58 | 72.56 | 77.25 | 98.31 | 91.82 | 94.95 | 94.74 | 84.89 | 89.54 | 90.34 | 84.60 | 87.38 | 76.74 | 62.70 | 69.01 |
| ratio=0.8 | 95.55 | 95.38 | 95.46 | 94.60 | 65.77 | 77.59 | 81.44 | 71.80 | 76.32 | 97.15 | 92.90 | 94.98 | 94.52 | 84.75 | 89.37 | 89.91 | 85.93 | 87.87 | 76.72 | 59.56 | 67.06 |
| ratio=1.0 | 95.01 | 96.57 | 95.78 | 95.80 | 67.88 | 79.46 | 82.60 | 72.62 | 77.29 | 97.42 | 92.13 | 94.70 | 94.67 | 85.59 | 89.90 | 91.56 | 88.50 | 90.00 | 76.98 | 61.04 | 68.09 |

*Table 20.* The per-dataset accuracy (%) of more experimental analysis on privacy protection in uploading embeddings under four data settings. "Few" and "Full" represent few-shot and full-data, while "Dir(0.1)" and "Non-IID" represent Dirichlet(0.1) and Non-IID data setting.

| Few_Dir(0.1) | Average | | | CIFAR10 | | | CIFAR100 | | | EuroSAT | | | Tiny-ImageNet | | |
|---|---|---|---|---|---|---|---|---|---|---|---|---|---|---|---|
| | Base | Novel | HM | Base | Novel | HM | Base | Novel | HM | Base | Novel | HM | Base | Novel | HM |
| FedDTL | 90.95 | 82.64 | 86.32 | 97.13 | 97.53 | 97.33 | 82.79 | 76.83 | 79.70 | 88.49 | 70.91 | 78.73 | 78.21 | 68.38 | 72.97 |
| + Noise Perturbation | 89.69 | 82.17 | 85.56 | 97.06 | 97.30 | 97.18 | 82.52 | 76.40 | 79.34 | 81.57 | 66.12 | 73.04 | 75.62 | 67.14 | 71.13 |
| + Embedding Aggregation | 88.71 | 81.21 | 84.56 | 96.80 | 97.22 | 97.01 | 81.14 | 75.64 | 78.29 | 75.36 | 55.88 | 64.17 | 76.32 | 66.04 | 70.81 |

| Few_Dir(0.1) | OxfordPet | | | Flower102 | | | Caltech101 | | | Caltech256 | | | Food101 | | |
|---|---|---|---|---|---|---|---|---|---|---|---|---|---|---|---|
| | Base | Novel | HM | Base | Novel | HM | Base | Novel | HM | Base | Novel | HM | Base | Novel | HM |
| FedDTL | 95.51 | 95.99 | 95.75 | 94.90 | 66.01 | 77.86 | 96.23 | 94.24 | 95.22 | 94.11 | 84.32 | 88.95 | 91.21 | 89.56 | 90.38 |
| + Noise Perturbation | 94.95 | 96.13 | 95.54 | 94.02 | 70.21 | 80.39 | 96.17 | 93.98 | 95.06 | 94.25 | 82.27 | 87.85 | 91.03 | 90.00 | 90.51 |
| + Embedding Aggregation | 95.29 | 96.45 | 95.87 | 91.96 | 70.39 | 79.74 | 96.43 | 94.60 | 95.51 | 93.80 | 84.61 | 88.97 | 91.32 | 90.05 | 90.68 |

| Few_Non-IID | Average | | | CIFAR10 | | | CIFAR100 | | | EuroSAT | | | Tiny-ImageNet | | |
|---|---|---|---|---|---|---|---|---|---|---|---|---|---|---|---|
| | Base | Novel | HM | Base | Novel | HM | Base | Novel | HM | Base | Novel | HM | Base | Novel | HM |
| FedDTL | 89.58 | 83.01 | 85.97 | 96.61 | 97.31 | 96.96 | 82.21 | 77.90 | 80.00 | 79.12 | 64.72 | 71.20 | 77.10 | 68.99 | 72.82 |
| + Noise Perturbation | 88.77 | 82.23 | 85.11 | 96.70 | 97.20 | 96.95 | 81.74 | 77.66 | 79.65 | 79.71 | 57.46 | 66.78 | 75.86 | 66.74 | 71.01 |
| + Embedding Aggregation | 88.64 | 83.20 | 85.69 | 96.26 | 96.58 | 96.42 | 82.32 | 78.60 | 80.42 | 80.82 | 68.85 | 74.36 | 75.20 | 67.66 | 71.23 |

| Few_Non-IID | OxfordPet | | | Flower102 | | | Caltech101 | | | Caltech256 | | | Food101 | | |
|---|---|---|---|---|---|---|---|---|---|---|---|---|---|---|---|
| | Base | Novel | HM | Base | Novel | HM | Base | Novel | HM | Base | Novel | HM | Base | Novel | HM |
| FedDTL | 95.28 | 96.39 | 95.83 | 94.33 | 70.64 | 80.78 | 95.69 | 94.44 | 95.06 | 94.39 | 85.80 | 89.89 | 91.45 | 90.92 | 91.18 |
| + Noise Perturbation | 95.61 | 97.12 | 96.36 | 91.60 | 72.49 | 80.93 | 92.42 | 94.44 | 93.42 | 94.13 | 85.48 | 89.60 | 91.18 | 91.47 | 91.32 |
| + Embedding Aggregation | 94.53 | 97.17 | 95.83 | 87.70 | 69.81 | 77.74 | 95.63 | 94.14 | 94.88 | 94.56 | 85.10 | 89.58 | 90.66 | 90.89 | 90.77 |

| Full_Dir(0.1) | Average | | | CIFAR10 | | | CIFAR100 | | | EuroSAT | | | Tiny-ImageNet | | |
|---|---|---|---|---|---|---|---|---|---|---|---|---|---|---|---|
| | Base | Novel | HM | Base | Novel | HM | Base | Novel | HM | Base | Novel | HM | Base | Novel | HM |
| FedDTL | 92.40 | 76.59 | 82.34 | 96.24 | 93.98 | 95.10 | 84.22 | 72.20 | 77.75 | 97.21 | 34.54 | 50.97 | 78.64 | 63.12 | 70.03 |
| + Noise Perturbation | 91.80 | 77.86 | 83.35 | 96.58 | 95.32 | 95.95 | 84.24 | 70.62 | 76.83 | 94.96 | 43.12 | 59.31 | 78.18 | 62.60 | 69.53 |
| + Embedding Aggregation | 90.91 | 77.79 | 82.84 | 96.60 | 94.44 | 95.51 | 81.50 | 73.90 | 77.51 | 92.39 | 40.27 | 56.09 | 77.68 | 66.16 | 71.46 |

| Full_Dir(0.1) | OxfordPet | | | Flower102 | | | Caltech101 | | | Caltech256 | | | Food101 | | |
|---|---|---|---|---|---|---|---|---|---|---|---|---|---|---|---|
| | Base | Novel | HM | Base | Novel | HM | Base | Novel | HM | Base | Novel | HM | Base | Novel | HM |
| FedDTL | 96.13 | 95.23 | 95.68 | 95.49 | 65.49 | 77.69 | 98.22 | 93.98 | 96.05 | 94.67 | 84.55 | 89.32 | 90.81 | 86.20 | 88.44 |
| + Noise Perturbation | 95.51 | 96.66 | 96.08 | 94.59 | 70.57 | 80.83 | 97.59 | 92.75 | 95.11 | 94.53 | 85.15 | 89.60 | 90.05 | 83.98 | 86.91 |
| + Embedding Aggregation | 95.51 | 95.92 | 95.71 | 94.51 | 67.06 | 78.45 | 97.50 | 93.36 | 95.39 | 93.94 | 84.91 | 89.20 | 88.56 | 84.06 | 86.25 |

| Full_Non-IID | Average | | | CIFAR10 | | | CIFAR100 | | | EuroSAT | | | Tiny-ImageNet | | |
|---|---|---|---|---|---|---|---|---|---|---|---|---|---|---|---|
| | Base | Novel | HM | Base | Novel | HM | Base | Novel | HM | Base | Novel | HM | Base | Novel | HM |
| FedDTL | 91.64 | 77.72 | 83.12 | 94.64 | 92.06 | 93.33 | 82.60 | 72.62 | 77.29 | 96.07 | 43.08 | 59.49 | 76.98 | 61.04 | 68.09 |
| + Noise Perturbation | 91.44 | 79.04 | 84.15 | 95.34 | 94.06 | 94.70 | 83.24 | 73.36 | 77.99 | 97.25 | 55.38 | 70.57 | 76.26 | 60.88 | 67.71 |
| + Embedding Aggregation | 81.92 | 72.50 | 76.65 | 32.84 | 30.28 | 31.51 | 81.36 | 73.90 | 77.45 | 80.25 | 54.92 | 65.21 | 76.54 | 63.56 | 69.45 |

| Full_Non-IID | OxfordPet | | | Flower102 | | | Caltech101 | | | Caltech256 | | | Food101 | | |
|---|---|---|---|---|---|---|---|---|---|---|---|---|---|---|---|
| | Base | Novel | HM | Base | Novel | HM | Base | Novel | HM | Base | Novel | HM | Base | Novel | HM |
| FedDTL | 95.01 | 96.57 | 95.78 | 95.80 | 67.88 | 79.46 | 97.42 | 92.13 | 94.70 | 94.67 | 85.59 | 89.90 | 91.56 | 88.50 | 90.00 |
| + Noise Perturbation | 94.87 | 96.62 | 95.74 | 94.98 | 66.51 | 78.24 | 96.88 | 93.52 | 95.17 | 94.92 | 84.75 | 89.55 | 89.26 | 86.24 | 87.72 |
| + Embedding Aggregation | 95.14 | 96.84 | 95.98 | 92.00 | 69.81 | 79.38 | 96.52 | 93.21 | 94.84 | 94.16 | 84.89 | 89.29 | 88.43 | 85.07 | 86.72 |

*Table 21.* The per-dataset accuracy comparison results (%) under 4-shot and 8-shot with IID and Non-IID data distributions.

### 4_IID

| | Average | | | CIFAR10 | | | CIFAR100 | | | EuroSAT | | | Tiny-ImageNet | | |
|---|---|---|---|---|---|---|---|---|---|---|---|---|---|---|---|
| | Local | Base | Novel | Local | Base | Novel | Local | Base | Novel | Local | Base | Novel | Local | Base | Novel |
| CLIP | 79.76 | 79.76 | 82.12 | 95.72 | 95.72 | 94.26 | 75.78 | 75.78 | 74.76 | 52.04 | 52.04 | 55.81 | 71.98 | 71.98 | 69.18 |
| pFedDC | 86.83 | 86.83 | 76.64 | 95.75 | 95.75 | 95.78 | 77.44 | 77.44 | 64.70 | 93.92 | 93.92 | 63.85 | 72.50 | 72.50 | 62.25 |
| FedPGP | 83.16 | 83.16 | 81.66 | 95.18 | 95.18 | 95.46 | 75.25 | 75.25 | 69.72 | 66.82 | 66.81 | 62.12 | 74.59 | 74.59 | 68.44 |
| pFedMMA | 85.58 | 85.58 | 79.89 | 96.29 | 96.29 | 95.00 | 80.01 | 80.01 | 71.54 | 61.69 | 61.69 | 58.84 | 72.34 | 72.34 | 61.51 |
| PromptFL | 82.98 | 82.98 | 81.40 | 95.40 | 95.40 | 96.06 | 76.62 | 76.62 | 73.88 | 61.61 | 61.61 | 50.81 | 75.00 | 75.00 | 69.32 |
| FedMaPLe | 86.69 | 86.69 | 82.76 | 96.66 | 96.66 | 94.28 | 82.34 | 82.34 | 77.70 | 71.43 | 71.43 | 61.23 | 78.82 | 78.82 | 71.24 |
| FedDTL | 89.83 | 89.83 | 84.12 | 96.90 | 96.90 | 97.18 | 83.64 | 83.64 | 77.42 | 81.04 | 81.04 | 75.50 | 76.62 | 76.62 | 68.22 |

| | OxfordPet | | | Flower102 | | | Caltech101 | | | Caltech256 | | | Food101 | | |
|---|---|---|---|---|---|---|---|---|---|---|---|---|---|---|---|
| | Local | Base | Novel | Local | Base | Novel | Local | Base | Novel | Local | Base | Novel | Local | Base | Novel |
| CLIP | 88.78 | 88.78 | 96.82 | 64.12 | 64.12 | 76.27 | 86.81 | 86.81 | 93.52 | 90.97 | 90.97 | 85.36 | 91.68 | 91.68 | 93.11 |
| pFedDC | 92.76 | 92.76 | 93.82 | 84.04 | 84.04 | 58.24 | 89.22 | 89.22 | 87.87 | 90.49 | 90.49 | 80.38 | 85.31 | 85.31 | 82.84 |
| FedPGP | 94.31 | 94.31 | 95.18 | 69.02 | 69.02 | 73.65 | 89.37 | 89.38 | 94.32 | 91.98 | 91.98 | 84.07 | 91.88 | 91.88 | 91.96 |
| pFedMMA | 95.15 | 95.15 | 97.20 | 90.35 | 90.35 | 67.45 | 91.05 | 91.05 | 92.75 | 93.04 | 93.04 | 84.00 | 90.33 | 90.33 | 90.70 |
| PromptFL | 94.28 | 94.28 | 96.98 | 68.43 | 68.43 | 75.10 | 91.44 | 91.44 | 93.36 | 92.21 | 92.21 | 84.43 | 91.79 | 91.79 | 92.69 |
| FedMaPLe | 93.88 | 93.88 | 97.72 | 81.57 | 81.57 | 72.94 | 90.02 | 90.02 | 94.44 | 93.63 | 93.63 | 83.23 | 91.82 | 91.82 | 92.10 |
| FedDTL | 95.74 | 95.74 | 96.98 | 94.51 | 94.51 | 70.20 | 95.19 | 95.19 | 94.75 | 94.05 | 94.05 | 85.48 | 90.81 | 90.81 | 91.36 |

### 4_Non-IID

| | Average | | | CIFAR10 | | | CIFAR100 | | | EuroSAT | | | Tiny-ImageNet | | |
|---|---|---|---|---|---|---|---|---|---|---|---|---|---|---|---|
| | Local | Base | Novel | Local | Base | Novel | Local | Base | Novel | Local | Base | Novel | Local | Base | Novel |
| CLIP | 80.22 | 79.62 | 81.99 | 95.72 | 95.72 | 94.26 | 75.78 | 75.78 | 74.76 | 52.75 | 52.04 | 55.81 | 71.98 | 71.98 | 69.18 |
| pFedDC | 95.19 | 47.66 | 61.53 | 100.00 | 30.44 | 67.61 | 90.78 | 47.12 | 53.99 | 100.00 | 20.00 | 33.18 | 85.42 | 44.21 | 48.47 |
| FedPGP | 80.42 | 79.57 | 76.64 | 95.88 | 95.74 | 94.35 | 74.04 | 73.88 | 58.44 | 52.13 | 49.95 | 28.75 | 73.70 | 73.50 | 69.54 |
| pFedMMA | 95.92 | 68.46 | 77.76 | 99.60 | 84.74 | 93.55 | 92.84 | 66.64 | 73.60 | 100.00 | 26.96 | 41.27 | 88.68 | 60.52 | 64.51 |
| PromptFL | 80.99 | 80.37 | 81.63 | 93.94 | 93.94 | 86.48 | 75.54 | 75.54 | 69.48 | 56.68 | 54.25 | 64.23 | 73.20 | 73.20 | 70.14 |
| FedMaPLe | 84.97 | 84.47 | 81.58 | 96.14 | 96.14 | 96.48 | 78.30 | 78.30 | 76.06 | 78.74 | 77.68 | 50.15 | 73.56 | 73.56 | 69.32 |
| FedDTL | 88.03 | 87.85 | 82.87 | 96.60 | 96.60 | 96.30 | 81.42 | 81.42 | 78.02 | 78.15 | 77.14 | 62.62 | 76.02 | 76.02 | 69.06 |

| | OxfordPet | | | Flower102 | | | Caltech101 | | | Caltech256 | | | Food101 | | |
|---|---|---|---|---|---|---|---|---|---|---|---|---|---|---|---|
| | Local | Base | Novel | Local | Base | Novel | Local | Base | Novel | Local | Base | Novel | Local | Base | Novel |
| CLIP | 86.70 | 86.84 | 96.71 | 64.20 | 64.20 | 75.38 | 91.70 | 86.81 | 93.52 | 91.47 | 91.55 | 85.21 | 91.68 | 91.68 | 93.11 |
| pFedDC | 98.26 | 43.50 | 72.58 | 98.20 | 34.84 | 47.54 | 95.78 | 80.94 | 87.78 | 94.65 | 74.31 | 72.20 | 93.66 | 53.62 | 70.39 |
| FedPGP | 87.16 | 86.29 | 96.50 | 64.80 | 64.28 | 71.38 | 92.21 | 88.82 | 92.35 | 91.72 | 91.56 | 85.08 | 92.17 | 92.15 | 93.39 |
| pFedMMA | 97.32 | 82.36 | 95.14 | 95.80 | 40.80 | 64.58 | 95.87 | 88.95 | 94.44 | 97.37 | 88.05 | 84.88 | 95.83 | 77.12 | 87.83 |
| PromptFL | 87.03 | 87.18 | 96.43 | 66.20 | 66.20 | 76.15 | 93.08 | 89.75 | 93.52 | 91.42 | 91.44 | 85.21 | 91.86 | 91.86 | 92.99 |
| FedMaPLe | 89.71 | 89.81 | 97.90 | 70.40 | 70.40 | 73.46 | 93.65 | 90.11 | 93.06 | 92.61 | 92.59 | 84.89 | 91.63 | 91.63 | 92.90 |
| FedDTL | 93.43 | 93.45 | 96.11 | 86.00 | 86.00 | 71.35 | 94.84 | 94.12 | 94.14 | 94.43 | 94.49 | 85.83 | 91.39 | 91.39 | 92.43 |

### 8_IID

| | Average | | | CIFAR10 | | | CIFAR100 | | | EuroSAT | | | Tiny-ImageNet | | |
|---|---|---|---|---|---|---|---|---|---|---|---|---|---|---|---|
| | Local | Base | Novel | Local | Base | Novel | Local | Base | Novel | Local | Base | Novel | Local | Base | Novel |
| CLIP | 79.76 | 79.76 | 82.12 | 95.72 | 95.72 | 94.26 | 75.78 | 75.78 | 74.76 | 52.04 | 52.04 | 55.81 | 71.98 | 71.98 | 69.18 |
| pFedDC | 88.74 | 88.74 | 77.21 | 96.07 | 96.07 | 95.96 | 79.39 | 79.39 | 65.94 | 96.26 | 96.26 | 65.59 | 76.04 | 76.04 | 64.58 |
| FedPGP | 84.56 | 84.56 | 81.93 | 94.50 | 94.50 | 95.65 | 76.77 | 76.77 | 72.35 | 68.91 | 68.91 | 61.67 | 74.85 | 74.85 | 69.24 |
| pFedMMA | 88.29 | 88.29 | 78.65 | 96.68 | 96.68 | 95.48 | 80.22 | 80.22 | 68.94 | 80.41 | 80.41 | 59.20 | 73.88 | 73.88 | 60.61 |
| PromptFL | 85.86 | 85.86 | 79.19 | 95.50 | 95.50 | 96.74 | 76.86 | 76.86 | 69.60 | 82.11 | 82.11 | 42.46 | 74.76 | 74.76 | 68.96 |
| FedMaPLe | 89.61 | 89.61 | 81.61 | 96.08 | 96.08 | 97.26 | 84.32 | 84.32 | 74.16 | 83.96 | 83.96 | 59.31 | 80.02 | 80.02 | 70.26 |
| FedDTL | 91.94 | 91.94 | 82.05 | 97.26 | 97.26 | 96.78 | 84.46 | 84.46 | 76.44 | 92.50 | 92.50 | 66.88 | 78.70 | 78.70 | 68.86 |

| | OxfordPet | | | Flower102 | | | Caltech101 | | | Caltech256 | | | Food101 | | |
|---|---|---|---|---|---|---|---|---|---|---|---|---|---|---|---|
| | Local | Base | Novel | Local | Base | Novel | Local | Base | Novel | Local | Base | Novel | Local | Base | Novel |
| CLIP | 88.78 | 88.78 | 96.82 | 64.12 | 64.12 | 76.27 | 86.81 | 86.81 | 93.52 | 90.97 | 90.97 | 85.36 | 91.68 | 91.68 | 93.11 |
| pFedDC | 93.92 | 93.92 | 92.42 | 88.47 | 88.47 | 55.69 | 91.02 | 91.02 | 89.35 | 91.04 | 91.03 | 81.25 | 86.42 | 86.42 | 84.12 |
| FedPGP | 94.64 | 94.64 | 96.92 | 76.04 | 76.04 | 71.80 | 90.27 | 90.27 | 94.10 | 92.79 | 92.79 | 82.88 | 92.23 | 92.23 | 92.72 |
| pFedMMA | 95.82 | 95.81 | 97.35 | 92.16 | 92.16 | 63.22 | 92.07 | 92.07 | 92.41 | 93.01 | 93.01 | 82.39 | 90.36 | 90.36 | 88.28 |
| PromptFL | 94.78 | 94.78 | 97.30 | 74.31 | 74.31 | 67.25 | 90.11 | 90.11 | 93.98 | 92.59 | 92.59 | 85.00 | 91.69 | 91.69 | 91.43 |
| FedMaPLe | 95.01 | 95.01 | 96.03 | 90.00 | 90.00 | 66.08 | 90.82 | 90.82 | 94.29 | 94.29 | 94.29 | 85.45 | 91.96 | 91.96 | 91.92 |
| FedDTL | 95.40 | 95.40 | 97.09 | 96.67 | 96.67 | 62.35 | 96.61 | 96.61 | 94.91 | 94.46 | 94.46 | 84.61 | 91.38 | 91.38 | 90.57 |

### 8_Non-IID

| | Average | | | CIFAR10 | | | CIFAR100 | | | EuroSAT | | | Tiny-ImageNet | | |
|---|---|---|---|---|---|---|---|---|---|---|---|---|---|---|---|
| | Local | Base | Novel | Local | Base | Novel | Local | Base | Novel | Local | Base | Novel | Local | Base | Novel |
| CLIP | 80.22 | 79.62 | 81.99 | 95.72 | 95.72 | 94.26 | 75.78 | 75.78 | 74.76 | 52.75 | 52.04 | 55.81 | 71.98 | 71.98 | 69.18 |
| pFedDC | 95.58 | 42.17 | 56.78 | 100.00 | 21.94 | 64.37 | 92.14 | 38.95 | 44.66 | 100.00 | 20.00 | 24.63 | 86.10 | 34.67 | 38.67 |
| FedPGP | 82.66 | 81.71 | 79.02 | 96.12 | 95.79 | 95.99 | 76.26 | 75.58 | 67.18 | 66.26 | 64.11 | 43.20 | 74.02 | 73.53 | 68.42 |
| pFedMMA | 96.51 | 63.74 | 75.24 | 99.78 | 85.59 | 92.40 | 94.20 | 61.04 | 71.01 | 100.00 | 26.84 | 41.11 | 88.64 | 51.56 | 57.68 |
| PromptFL | 82.01 | 81.60 | 80.03 | 95.74 | 95.74 | 96.24 | 73.40 | 73.40 | 60.16 | 66.52 | 65.07 | 56.58 | 73.58 | 73.58 | 67.86 |
| FedMaPLe | 84.92 | 84.62 | 80.09 | 95.80 | 95.80 | 97.32 | 78.44 | 78.44 | 72.86 | 79.97 | 79.29 | 44.65 | 75.00 | 75.00 | 65.52 |
| FedDTL | 88.48 | 88.04 | 82.83 | 96.46 | 96.46 | 96.10 | 81.98 | 81.98 | 78.10 | 72.65 | 71.32 | 59.19 | 76.42 | 76.42 | 69.98 |

| | OxfordPet | | | Flower102 | | | Caltech101 | | | Caltech256 | | | Food101 | | |
|---|---|---|---|---|---|---|---|---|---|---|---|---|---|---|---|
| | Local | Base | Novel | Local | Base | Novel | Local | Base | Novel | Local | Base | Novel | Local | Base | Novel |
| CLIP | 86.70 | 86.84 | 96.71 | 64.20 | 64.20 | 75.38 | 91.70 | 86.81 | 93.52 | 91.47 | 91.55 | 85.21 | 91.68 | 91.68 | 93.11 |
| pFedDC | 97.32 | 42.40 | 72.18 | 99.00 | 32.48 | 45.65 | 94.58 | 72.92 | 83.15 | 96.40 | 66.04 | 66.28 | 94.70 | 50.11 | 71.41 |
| FedPGP | 87.48 | 87.46 | 96.35 | 66.80 | 66.04 | 74.04 | 93.75 | 90.02 | 93.52 | 91.61 | 91.31 | 81.61 | 91.62 | 91.51 | 90.85 |
| pFedMMA | 97.38 | 82.36 | 95.88 | 98.40 | 38.84 | 60.23 | 95.75 | 85.08 | 93.52 | 97.99 | 86.08 | 83.98 | 96.47 | 56.25 | 81.41 |
| PromptFL | 86.76 | 86.91 | 97.62 | 68.00 | 68.00 | 72.69 | 91.40 | 88.95 | 92.59 | 91.28 | 91.34 | 85.42 | 91.42 | 91.42 | 91.07 |
| FedMaPLe | 88.70 | 88.87 | 97.03 | 72.20 | 72.20 | 75.77 | 90.47 | 88.32 | 91.36 | 92.14 | 92.16 | 83.09 | 91.54 | 91.54 | 93.23 |
| FedDTL | 94.75 | 94.74 | 96.57 | 93.60 | 93.60 | 72.88 | 95.13 | 92.42 | 94.14 | 93.99 | 94.02 | 86.38 | 91.36 | 91.36 | 92.16 |

*Table 22.* The per-dataset accuracy comparison results (%) under 4-shot with different Dirichlet data distributions.

| Dir(0.1) | Average | | | CIFAR10 | | | CIFAR100 | | | EuroSAT | | | Tiny-ImageNet | | |
|---|---|---|---|---|---|---|---|---|---|---|---|---|---|---|---|
| | Local | Base | Novel | Local | Base | Novel | Local | Base | Novel | Local | Base | Novel | Local | Base | Novel |
| CLIP | 79.44 | 79.76 | 82.12 | 95.75 | 95.72 | 94.26 | 76.17 | 75.78 | 74.76 | 49.79 | 52.04 | 55.81 | 71.80 | 71.98 | 69.18 |
| pFedDC | 87.73 | 73.28 | 72.91 | 95.58 | 88.33 | 92.12 | 79.18 | 64.04 | 59.50 | 91.11 | 60.86 | 50.82 | 71.97 | 60.11 | 53.74 |
| FedPGP | 82.91 | 82.72 | 81.56 | 95.43 | 95.49 | 94.34 | 76.94 | 75.83 | 68.01 | 67.13 | 65.04 | 62.42 | 74.05 | 74.23 | 69.87 |
| pFedMMA | 88.14 | 79.56 | 79.57 | 96.36 | 95.71 | 94.42 | 84.55 | 74.03 | 73.23 | 73.35 | 57.14 | 52.03 | 76.85 | 66.69 | 62.77 |
| PromptFL | 81.11 | 80.37 | 78.86 | 95.79 | 95.74 | 97.08 | 77.18 | 76.32 | 71.82 | 46.40 | 44.21 | 37.42 | 73.64 | 73.06 | 66.32 |
| FedMaPLe | 85.23 | 84.67 | 80.46 | 96.23 | 96.20 | 96.96 | 81.58 | 80.92 | 77.00 | 74.22 | 69.54 | 44.19 | 75.07 | 74.70 | 69.56 |
| FedDTL | 88.73 | 87.33 | 81.60 | 96.73 | 96.74 | 97.04 | 82.42 | 80.94 | 77.12 | 78.06 | 70.86 | 54.73 | 77.05 | 76.32 | 68.44 |

| Dir(0.1) | OxfordPet | | | Flower102 | | | Caltech101 | | | Caltech256 | | | Food101 | | |
|---|---|---|---|---|---|---|---|---|---|---|---|---|---|---|---|
| | Local | Base | Novel | Local | Base | Novel | Local | Base | Novel | Local | Base | Novel | Local | Base | Novel |
| CLIP | 89.79 | 88.78 | 96.82 | 63.39 | 64.12 | 76.27 | 86.16 | 86.81 | 93.52 | 90.81 | 90.97 | 85.36 | 91.27 | 91.68 | 93.11 |
| pFedDC | 96.07 | 77.71 | 93.25 | 86.72 | 57.88 | 59.14 | 92.22 | 85.99 | 90.15 | 89.50 | 82.71 | 73.96 | 87.19 | 81.89 | 83.52 |
| FedPGP | 90.58 | 90.31 | 94.71 | 69.42 | 69.37 | 73.61 | 89.48 | 91.02 | 93.61 | 91.19 | 91.16 | 85.31 | 91.94 | 92.07 | 92.18 |
| pFedMMA | 95.42 | 90.19 | 96.52 | 88.98 | 64.67 | 68.82 | 91.82 | 89.45 | 93.86 | 94.45 | 90.34 | 84.07 | 91.44 | 87.78 | 90.37 |
| PromptFL | 93.33 | 92.54 | 94.01 | 68.60 | 67.45 | 72.94 | 91.36 | 90.29 | 93.67 | 92.15 | 92.11 | 83.80 | 91.50 | 91.60 | 92.69 |
| FedMaPLe | 90.88 | 90.52 | 96.98 | 77.12 | 75.49 | 70.39 | 88.12 | 90.55 | 94.14 | 92.26 | 92.35 | 81.61 | 91.57 | 91.78 | 93.29 |
| FedDTL | 95.88 | 94.78 | 97.03 | 92.60 | 90.78 | 70.00 | 90.60 | 90.46 | 94.14 | 94.06 | 93.94 | 85.63 | 91.20 | 91.12 | 90.31 |

| Dir(0.3) | Average | | | CIFAR10 | | | CIFAR100 | | | EuroSAT | | | Tiny-ImageNet | | |
|---|---|---|---|---|---|---|---|---|---|---|---|---|---|---|---|
| | Local | Base | Novel | Local | Base | Novel | Local | Base | Novel | Local | Base | Novel | Local | Base | Novel |
| CLIP | 79.83 | 79.76 | 82.12 | 95.83 | 95.72 | 94.26 | 75.28 | 75.78 | 74.76 | 51.16 | 52.04 | 55.81 | 71.87 | 71.98 | 69.18 |
| pFedDC | 85.68 | 81.11 | 75.78 | 95.49 | 94.07 | 92.73 | 75.75 | 72.29 | 60.47 | 83.25 | 80.08 | 58.78 | 71.62 | 69.70 | 60.33 |
| FedPGP | 84.39 | 83.97 | 83.46 | 95.39 | 95.37 | 94.46 | 75.19 | 75.23 | 71.46 | 76.13 | 76.68 | 72.52 | 73.55 | 73.35 | 70.54 |
| pFedMMA | 85.29 | 82.72 | 80.28 | 96.29 | 95.96 | 94.93 | 79.72 | 77.98 | 73.22 | 61.76 | 61.74 | 58.18 | 71.14 | 68.84 | 60.14 |
| PromptFL | 82.36 | 82.12 | 82.08 | 95.67 | 95.68 | 97.08 | 76.39 | 76.46 | 72.10 | 54.67 | 54.07 | 62.31 | 76.01 | 75.82 | 68.84 |
| FedMaPLe | 85.39 | 85.15 | 81.92 | 96.00 | 95.90 | 96.48 | 79.92 | 79.82 | 71.48 | 69.21 | 69.46 | 54.42 | 76.54 | 76.36 | 71.62 |
| FedDTL | 89.71 | 89.25 | 84.28 | 96.65 | 96.74 | 96.86 | 82.64 | 82.44 | 77.06 | 81.78 | 81.25 | 78.27 | 77.52 | 77.10 | 69.20 |

| Dir(0.3) | OxfordPet | | | Flower102 | | | Caltech101 | | | Caltech256 | | | Food101 | | |
|---|---|---|---|---|---|---|---|---|---|---|---|---|---|---|---|
| | Local | Base | Novel | Local | Base | Novel | Local | Base | Novel | Local | Base | Novel | Local | Base | Novel |
| CLIP | 88.60 | 88.78 | 96.82 | 65.34 | 64.12 | 76.27 | 87.74 | 86.81 | 93.52 | 91.01 | 90.97 | 85.36 | 91.66 | 91.68 | 93.11 |
| pFedDC | 92.22 | 81.71 | 93.84 | 84.81 | 72.35 | 63.61 | 92.24 | 87.72 | 89.75 | 90.30 | 87.82 | 79.31 | 85.44 | 84.27 | 83.17 |
| FedPGP | 92.71 | 92.35 | 96.81 | 69.55 | 68.35 | 75.92 | 92.53 | 89.96 | 93.21 | 92.57 | 92.48 | 86.02 | 91.87 | 91.92 | 90.22 |
| pFedMMA | 94.34 | 91.01 | 97.22 | 87.10 | 76.67 | 69.41 | 93.19 | 90.36 | 93.77 | 93.19 | 91.71 | 84.49 | 90.90 | 90.21 | 91.19 |
| PromptFL | 94.46 | 93.71 | 96.71 | 69.63 | 68.63 | 74.51 | 90.16 | 90.55 | 93.83 | 92.27 | 92.18 | 81.49 | 92.01 | 91.99 | 91.84 |
| FedMaPLe | 93.97 | 93.55 | 98.04 | 77.15 | 75.88 | 72.75 | 90.94 | 90.64 | 94.44 | 92.85 | 92.80 | 84.70 | 91.91 | 91.93 | 93.37 |
| FedDTL | 95.73 | 95.40 | 97.30 | 93.66 | 93.33 | 69.41 | 93.93 | 91.71 | 93.98 | 93.65 | 93.56 | 85.60 | 91.79 | 91.68 | 90.80 |

| Dir(0.5) | Average | | | CIFAR10 | | | CIFAR100 | | | EuroSAT | | | Tiny-ImageNet | | |
|---|---|---|---|---|---|---|---|---|---|---|---|---|---|---|---|
| | Local | Base | Novel | Local | Base | Novel | Local | Base | Novel | Local | Base | Novel | Local | Base | Novel |
| CLIP | 79.77 | 79.76 | 82.12 | 95.72 | 95.72 | 94.26 | 75.72 | 75.78 | 74.76 | 52.04 | 52.04 | 55.81 | 71.92 | 71.98 | 69.18 |
| pFedDC | 86.49 | 84.72 | 77.04 | 94.02 | 94.02 | 93.96 | 76.71 | 75.71 | 65.18 | 91.39 | 91.39 | 67.13 | 72.79 | 71.55 | 61.65 |
| FedPGP | 83.57 | 83.46 | 80.75 | 95.84 | 95.84 | 92.05 | 77.26 | 77.14 | 72.59 | 70.14 | 70.14 | 51.85 | 75.14 | 75.14 | 69.51 |
| pFedMMA | 85.29 | 83.97 | 80.19 | 96.11 | 96.11 | 94.55 | 79.86 | 78.98 | 72.40 | 62.14 | 62.14 | 59.08 | 73.18 | 72.03 | 62.16 |
| PromptFL | 82.00 | 82.05 | 80.59 | 96.06 | 96.06 | 95.76 | 77.06 | 77.02 | 72.58 | 52.61 | 52.61 | 51.42 | 74.94 | 74.94 | 68.52 |
| FedMaPLe | 87.26 | 87.20 | 83.05 | 95.76 | 95.76 | 95.48 | 80.38 | 80.40 | 76.48 | 82.46 | 82.46 | 62.00 | 78.05 | 77.88 | 70.54 |
| FedDTL | 89.79 | 89.50 | 82.45 | 96.62 | 96.62 | 96.84 | 83.27 | 83.24 | 77.10 | 83.14 | 83.14 | 62.62 | 77.46 | 77.24 | 68.94 |

| Dir(0.5) | OxfordPet | | | Flower102 | | | Caltech101 | | | Caltech256 | | | Food101 | | |
|---|---|---|---|---|---|---|---|---|---|---|---|---|---|---|---|
| | Local | Base | Novel | Local | Base | Novel | Local | Base | Novel | Local | Base | Novel | Local | Base | Novel |
| CLIP | 88.39 | 88.78 | 96.82 | 64.51 | 64.12 | 76.27 | 87.15 | 86.81 | 93.52 | 90.88 | 90.97 | 85.36 | 91.64 | 91.68 | 93.11 |
| pFedDC | 92.94 | 89.35 | 93.68 | 82.79 | 76.51 | 58.75 | 91.92 | 89.41 | 89.66 | 88.85 | 88.11 | 78.75 | 87.04 | 86.47 | 84.62 |
| FedPGP | 89.14 | 89.78 | 96.85 | 69.14 | 68.59 | 74.55 | 91.54 | 90.62 | 93.77 | 91.93 | 91.98 | 84.55 | 91.97 | 91.95 | 91.01 |
| pFedMMA | 93.44 | 92.39 | 97.44 | 86.04 | 80.24 | 68.27 | 92.98 | 91.00 | 93.43 | 93.10 | 92.53 | 84.57 | 90.74 | 90.32 | 89.84 |
| PromptFL | 93.52 | 93.60 | 97.72 | 69.66 | 69.41 | 73.53 | 89.71 | 90.37 | 92.90 | 92.58 | 92.52 | 80.80 | 91.87 | 91.88 | 92.25 |
| FedMaPLe | 89.77 | 90.24 | 97.09 | 83.37 | 82.75 | 74.31 | 90.39 | 90.29 | 94.60 | 92.94 | 92.80 | 83.62 | 92.19 | 92.18 | 93.33 |
| FedDTL | 95.56 | 95.23 | 95.50 | 94.84 | 94.31 | 70.39 | 92.12 | 90.82 | 94.44 | 93.47 | 93.32 | 85.24 | 91.61 | 91.55 | 90.97 |

*Table 23.* The per-dataset accuracy comparison results (%) under 8-shot with different Dirichlet data distributions.

| Dir(0.1) | Average | | | CIFAR10 | | | CIFAR100 | | | EuroSAT | | | Tiny-ImageNet | | |
|---|---|---|---|---|---|---|---|---|---|---|---|---|---|---|---|
| | Local | Base | Novel | Local | Base | Novel | Local | Base | Novel | Local | Base | Novel | Local | Base | Novel |
| CLIP | 79.44 | 79.76 | 82.12 | 95.75 | 95.72 | 94.26 | 76.17 | 75.78 | 74.76 | 49.79 | 52.04 | 55.81 | 71.80 | 71.98 | 69.18 |
| pFedDC | 88.03 | 72.37 | 70.02 | 96.52 | 91.84 | 94.32 | 80.80 | 64.34 | 60.42 | 90.75 | 60.29 | 32.65 | 75.15 | 61.03 | 55.88 |
| FedPGP | 82.47 | 81.92 | 79.48 | 96.00 | 95.99 | 96.11 | 76.14 | 75.60 | 68.94 | 62.41 | 57.75 | 43.28 | 75.00 | 74.24 | 66.42 |
| pFedMMA | 87.11 | 75.62 | 77.28 | 96.41 | 95.92 | 95.16 | 83.84 | 69.18 | 68.64 | 65.31 | 48.06 | 48.20 | 75.95 | 62.73 | 56.84 |
| PromptFL | 83.44 | 82.23 | 78.88 | 95.80 | 95.86 | 94.20 | 76.43 | 75.68 | 69.64 | 67.96 | 59.71 | 50.81 | 74.02 | 73.16 | 66.06 |
| FedMaPLe | 85.52 | 84.44 | 78.16 | 96.29 | 96.28 | 95.06 | 82.33 | 81.06 | 74.06 | 67.42 | 63.14 | 30.62 | 77.57 | 76.24 | 68.50 |
| FedDTL | 90.62 | 89.71 | 82.29 | 97.27 | 97.28 | 96.74 | 83.58 | 82.02 | 76.08 | 87.32 | 84.75 | 63.88 | 78.25 | 77.38 | 68.58 |

| Dir(0.1) | OxfordPet | | | Flower102 | | | Caltech101 | | | Caltech256 | | | Food101 | | |
|---|---|---|---|---|---|---|---|---|---|---|---|---|---|---|---|
| | Local | Base | Novel | Local | Base | Novel | Local | Base | Novel | Local | Base | Novel | Local | Base | Novel |
| CLIP | 89.79 | 88.78 | 96.82 | 63.39 | 64.12 | 76.27 | 86.16 | 86.81 | 93.52 | 90.81 | 90.97 | 85.36 | 91.27 | 91.68 | 93.11 |
| pFedDC | 95.28 | 74.50 | 90.16 | 87.42 | 56.75 | 54.24 | 90.54 | 83.12 | 86.45 | 90.66 | 83.27 | 75.23 | 85.12 | 76.17 | 80.84 |
| FedPGP | 90.89 | 90.51 | 95.70 | 68.83 | 68.75 | 71.57 | 88.56 | 90.14 | 93.89 | 92.81 | 92.54 | 86.61 | 91.59 | 91.80 | 92.76 |
| pFedMMA | 94.67 | 78.47 | 94.99 | 90.23 | 62.90 | 66.43 | 92.91 | 88.63 | 93.12 | 93.82 | 88.73 | 82.80 | 90.83 | 85.98 | 89.31 |
| PromptFL | 93.68 | 92.48 | 92.69 | 70.19 | 69.22 | 70.00 | 89.63 | 90.37 | 91.36 | 91.92 | 91.97 | 83.44 | 91.36 | 91.61 | 91.73 |
| FedMaPLe | 92.01 | 91.58 | 97.40 | 78.70 | 75.29 | 68.43 | 91.04 | 92.34 | 93.52 | 93.20 | 92.90 | 84.61 | 91.11 | 91.09 | 91.23 |
| FedDTL | 95.98 | 94.89 | 96.50 | 94.45 | 92.94 | 67.84 | 94.29 | 93.58 | 95.06 | 93.37 | 93.42 | 85.00 | 91.11 | 91.09 | 90.96 |

| Dir(0.3) | Average | | | CIFAR10 | | | CIFAR100 | | | EuroSAT | | | Tiny-ImageNet | | |
|---|---|---|---|---|---|---|---|---|---|---|---|---|---|---|---|
| | Local | Base | Novel | Local | Base | Novel | Local | Base | Novel | Local | Base | Novel | Local | Base | Novel |
| CLIP | 79.83 | 79.76 | 82.12 | 95.83 | 95.72 | 94.26 | 75.28 | 75.78 | 74.76 | 51.16 | 52.04 | 55.81 | 71.87 | 71.98 | 69.18 |
| pFedDC | 85.76 | 80.76 | 74.98 | 94.58 | 93.56 | 95.30 | 78.23 | 74.70 | 64.57 | 85.83 | 82.49 | 59.92 | 72.64 | 70.26 | 58.74 |
| FedPGP | 83.56 | 83.57 | 81.79 | 95.98 | 95.92 | 94.10 | 76.00 | 76.02 | 71.73 | 67.57 | 67.44 | 63.14 | 75.33 | 75.17 | 67.60 |
| pFedMMA | 86.28 | 82.54 | 79.13 | 96.78 | 96.40 | 95.68 | 79.21 | 76.36 | 69.78 | 68.64 | 68.45 | 58.20 | 73.57 | 70.55 | 60.05 |
| PromptFL | 84.37 | 84.07 | 80.11 | 96.25 | 96.22 | 94.60 | 75.81 | 75.86 | 71.96 | 68.85 | 68.54 | 49.27 | 75.12 | 74.98 | 68.68 |
| FedMaPLe | 86.83 | 86.68 | 81.73 | 96.65 | 96.66 | 95.28 | 82.65 | 82.40 | 74.42 | 71.84 | 71.89 | 57.35 | 78.29 | 78.06 | 69.90 |
| FedDTL | 90.88 | 90.49 | 83.79 | 96.68 | 96.76 | 97.32 | 83.13 | 82.92 | 77.54 | 86.57 | 86.54 | 75.42 | 78.63 | 78.18 | 68.72 |

| Dir(0.3) | OxfordPet | | | Flower102 | | | Caltech101 | | | Caltech256 | | | Food101 | | |
|---|---|---|---|---|---|---|---|---|---|---|---|---|---|---|---|
| | Local | Base | Novel | Local | Base | Novel | Local | Base | Novel | Local | Base | Novel | Local | Base | Novel |
| CLIP | 88.60 | 88.78 | 96.82 | 65.34 | 64.12 | 76.27 | 87.74 | 86.81 | 93.52 | 91.01 | 90.97 | 85.36 | 91.66 | 91.68 | 93.11 |
| pFedDC | 91.07 | 81.04 | 93.04 | 82.42 | 67.41 | 56.20 | 92.62 | 87.01 | 88.52 | 89.45 | 86.64 | 77.41 | 85.02 | 83.71 | 81.16 |
| FedPGP | 94.69 | 94.34 | 97.47 | 70.04 | 69.14 | 72.86 | 87.70 | 89.55 | 93.73 | 93.03 | 92.84 | 83.73 | 91.71 | 91.74 | 91.71 |
| pFedMMA | 93.21 | 87.46 | 96.85 | 87.79 | 73.29 | 67.25 | 94.10 | 90.52 | 92.96 | 92.77 | 90.47 | 82.29 | 90.48 | 89.39 | 89.13 |
| PromptFL | 94.31 | 93.43 | 92.69 | 74.12 | 73.53 | 75.49 | 90.25 | 89.66 | 93.67 | 92.38 | 92.38 | 82.90 | 92.24 | 92.24 | 91.77 |
| FedMaPLe | 95.09 | 94.67 | 97.40 | 80.44 | 79.02 | 71.37 | 90.52 | 91.62 | 94.14 | 93.84 | 93.67 | 84.28 | 92.18 | 92.13 | 91.46 |
| FedDTL | 95.12 | 94.28 | 95.34 | 96.15 | 95.69 | 71.57 | 95.69 | 94.30 | 93.98 | 94.49 | 94.36 | 83.83 | 91.49 | 91.36 | 90.38 |

| Dir(0.5) | Average | | | CIFAR10 | | | CIFAR100 | | | EuroSAT | | | Tiny-ImageNet | | |
|---|---|---|---|---|---|---|---|---|---|---|---|---|---|---|---|
| | Local | Base | Novel | Local | Base | Novel | Local | Base | Novel | Local | Base | Novel | Local | Base | Novel |
| CLIP | 79.77 | 79.76 | 82.12 | 95.72 | 95.72 | 94.26 | 75.72 | 75.78 | 74.76 | 52.04 | 52.04 | 55.81 | 71.92 | 71.98 | 69.18 |
| pFedDC | 87.25 | 85.25 | 76.93 | 94.07 | 94.07 | 94.15 | 78.49 | 76.95 | 63.55 | 95.92 | 95.92 | 65.37 | 74.25 | 73.09 | 63.13 |
| FedPGP | 83.19 | 83.01 | 82.17 | 96.22 | 96.22 | 94.07 | 74.42 | 74.42 | 71.82 | 67.72 | 67.71 | 64.32 | 74.39 | 74.32 | 67.87 |
| pFedMMA | 86.91 | 85.12 | 79.12 | 95.98 | 95.98 | 96.01 | 80.32 | 79.09 | 70.44 | 75.60 | 75.60 | 58.52 | 73.07 | 71.46 | 60.43 |
| PromptFL | 85.90 | 85.74 | 81.06 | 96.00 | 96.00 | 96.70 | 77.87 | 77.92 | 72.76 | 81.25 | 81.25 | 61.73 | 74.87 | 74.78 | 68.36 |
| FedMaPLe | 88.77 | 88.60 | 80.87 | 96.70 | 96.70 | 95.66 | 84.29 | 84.28 | 73.42 | 86.00 | 86.00 | 52.88 | 80.35 | 80.24 | 70.42 |
| FedDTL | 90.87 | 90.70 | 83.27 | 96.84 | 96.84 | 97.72 | 83.19 | 83.20 | 75.58 | 86.36 | 86.36 | 73.31 | 78.89 | 78.70 | 67.94 |

| Dir(0.5) | OxfordPet | | | Flower102 | | | Caltech101 | | | Caltech256 | | | Food101 | | |
|---|---|---|---|---|---|---|---|---|---|---|---|---|---|---|---|
| | Local | Base | Novel | Local | Base | Novel | Local | Base | Novel | Local | Base | Novel | Local | Base | Novel |
| CLIP | 88.39 | 88.78 | 96.82 | 64.51 | 64.12 | 76.27 | 87.15 | 86.81 | 93.52 | 90.88 | 90.97 | 85.36 | 91.64 | 91.68 | 93.11 |
| pFedDC | 92.34 | 87.92 | 94.48 | 83.35 | 76.35 | 59.45 | 91.42 | 88.95 | 89.75 | 89.21 | 88.48 | 78.59 | 86.24 | 85.56 | 83.90 |
| FedPGP | 88.77 | 89.19 | 96.79 | 73.48 | 72.67 | 73.96 | 89.88 | 88.84 | 93.49 | 92.15 | 92.06 | 84.19 | 91.66 | 91.67 | 93.04 |
| pFedMMA | 92.87 | 90.30 | 96.92 | 88.79 | 81.57 | 64.59 | 92.76 | 90.75 | 93.15 | 92.17 | 91.30 | 82.44 | 90.62 | 90.01 | 89.54 |
| PromptFL | 93.48 | 93.55 | 89.19 | 73.84 | 73.53 | 74.31 | 91.22 | 90.02 | 91.98 | 92.49 | 92.52 | 83.11 | 92.05 | 92.06 | 91.36 |
| FedMaPLe | 89.26 | 89.56 | 97.03 | 85.38 | 85.10 | 71.76 | 91.28 | 89.93 | 93.06 | 94.20 | 94.15 | 84.04 | 91.43 | 91.40 | 89.57 |
| FedDTL | 95.51 | 95.06 | 96.45 | 95.95 | 95.88 | 69.02 | 94.70 | 94.03 | 94.44 | 94.54 | 94.46 | 84.49 | 91.87 | 91.76 | 90.46 |

*Table 24.* The per-dataset accuracy (%) of ablation study on further communication analysis (uploading 16 embeddings per class) over seven base-to-novel benchmarks under the full-data setting."IID", "Dir(0.1)" and "Non-IID" represent IID, Dirichlet(0.1) and Non-IID data setting.

| Setting | OxfordPet | | | Flower102 | | | CIFAR100 | | | Caltech101 | | | Caltech256 | | | Food101 | | | Tiny-ImageNet | | |
|---|---|---|---|---|---|---|---|---|---|---|---|---|---|---|---|---|---|---|---|---|---|
| | Base | Novel | HM | Base | Novel | HM | Base | Novel | HM | Base | Novel | HM | Base | Novel | HM | Base | Novel | HM | Base | Novel | HM |
| IID | 96.07 | 96.61 | 96.34 | 96.47 | 66.27 | 78.57 | 87.36 | 74.36 | 80.34 | 97.59 | 92.90 | 95.19 | 95.36 | 84.94 | 89.85 | 92.21 | 87.18 | 89.62 | 82.24 | 66.12 | 73.30 |
| Dir(0.1) | 96.07 | 97.09 | 96.58 | 95.73 | 69.95 | 80.83 | 81.90 | 72.58 | 76.96 | 97.77 | 93.83 | 95.76 | 93.80 | 84.85 | 89.10 | 89.95 | 87.38 | 88.65 | 77.92 | 65.54 | 71.20 |
| Non-IID | 95.14 | 96.71 | 95.92 | 95.00 | 69.04 | 79.97 | 81.36 | 73.70 | 77.34 | 95.01 | 89.51 | 92.18 | 94.56 | 85.68 | 89.90 | 89.50 | 84.67 | 87.02 | 76.10 | 63.62 | 69.30 |

*Table 25.* The per-dataset accuracy (%) of some baselines with our GRPO-inspired RL strategy across nine base-to-novel datasets under four data settings. "Few" and "Full" represent few-shot and full-data, while "Dir(0.1)" and "Non-IID" represent Dirichlet(0.1) and Non-IID data setting.

| Few_Dir(0.1) | Average | | | CIFAR10 | | | CIFAR100 | | | EuroSAT | | | Tiny-ImageNet | | |
|---|---|---|---|---|---|---|---|---|---|---|---|---|---|---|---|
| | Base | Novel | HM | Base | Novel | HM | Base | Novel | HM | Base | Novel | HM | Base | Novel | HM |
| pFedDC | 76.06 | 77.68 | 76.78 | 92.46 | 97.05 | 94.70 | 74.65 | 70.64 | 72.59 | 42.49 | 44.14 | 43.30 | 68.22 | 65.14 | 66.64 |
| pFedMMA | 78.10 | 80.09 | 79.01 | 95.88 | 94.85 | 95.36 | 77.53 | 76.56 | 77.04 | 52.60 | 56.42 | 54.44 | 57.25 | 54.89 | 56.05 |
| FedMaPLe | 83.03 | 80.96 | 81.83 | 95.52 | 96.74 | 96.13 | 78.24 | 75.64 | 76.92 | 63.18 | 47.69 | 54.35 | 74.68 | 70.64 | 72.60 |

| Few_Dir(0.1) | OxfordPet | | | Flower102 | | | Caltech101 | | | Caltech256 | | | Food101 | | |
|---|---|---|---|---|---|---|---|---|---|---|---|---|---|---|---|
| | Base | Novel | HM | Base | Novel | HM | Base | Novel | HM | Base | Novel | HM | Base | Novel | HM |
| pFedDC | 79.74 | 91.96 | 85.42 | 59.10 | 64.45 | 61.66 | 88.82 | 92.22 | 90.49 | 90.42 | 84.67 | 87.45 | 88.66 | 88.87 | 88.76 |
| pFedMMA | 89.36 | 96.53 | 92.81 | 61.73 | 70.24 | 65.71 | 89.11 | 93.70 | 91.35 | 91.09 | 84.45 | 87.64 | 88.39 | 93.20 | 90.73 |
| FedMaPLe | 88.89 | 94.28 | 91.51 | 73.53 | 71.76 | 72.63 | 89.30 | 94.44 | 91.80 | 92.25 | 84.22 | 88.05 | 91.67 | 93.23 | 92.44 |

| Few_Non-IID | Average | | | CIFAR10 | | | CIFAR100 | | | EuroSAT | | | Tiny-ImageNet | | |
|---|---|---|---|---|---|---|---|---|---|---|---|---|---|---|---|
| | Base | Novel | HM | Base | Novel | HM | Base | Novel | HM | Base | Novel | HM | Base | Novel | HM |
| pFedDC | 60.46 | 74.27 | 65.67 | 59.71 | 90.65 | 72.00 | 56.30 | 58.82 | 57.53 | 23.38 | 55.98 | 32.98 | 62.19 | 61.77 | 61.98 |
| pFedMMA | 79.51 | 82.39 | 80.85 | 95.90 | 94.41 | 95.15 | 78.96 | 77.34 | 78.14 | 53.23 | 56.96 | 55.03 | 66.90 | 68.59 | 67.73 |
| FedMaPLe | 82.67 | 83.39 | 82.97 | 95.64 | 96.02 | 95.83 | 77.62 | 76.36 | 76.98 | 65.04 | 66.73 | 65.87 | 72.34 | 67.78 | 69.99 |

| Few_Non-IID | OxfordPet | | | Flower102 | | | Caltech101 | | | Caltech256 | | | Food101 | | |
|---|---|---|---|---|---|---|---|---|---|---|---|---|---|---|---|
| | Base | Novel | HM | Base | Novel | HM | Base | Novel | HM | Base | Novel | HM | Base | Novel | HM |
| pFedDC | 55.28 | 85.86 | 67.26 | 41.48 | 58.19 | 48.43 | 81.59 | 90.31 | 85.73 | 88.03 | 81.74 | 84.77 | 76.17 | 85.07 | 80.37 |
| pFedMMA | 87.25 | 96.94 | 91.84 | 64.32 | 75.46 | 69.45 | 88.86 | 93.80 | 91.26 | 89.55 | 85.23 | 87.34 | 90.61 | 92.76 | 91.67 |
| FedMaPLe | 88.93 | 97.21 | 92.89 | 70.80 | 74.62 | 72.66 | 90.11 | 93.52 | 91.78 | 92.30 | 85.24 | 88.63 | 91.24 | 93.00 | 92.11 |

| Full_Dir(0.1) | Average | | | CIFAR10 | | | CIFAR100 | | | EuroSAT | | | Tiny-ImageNet | | |
|---|---|---|---|---|---|---|---|---|---|---|---|---|---|---|---|
| | Base | Novel | HM | Base | Novel | HM | Base | Novel | HM | Base | Novel | HM | Base | Novel | HM |
| pFedDC | 69.08 | 71.55 | 70.18 | 55.59 | 59.16 | 57.32 | 67.91 | 65.91 | 66.90 | 41.99 | 36.57 | 39.09 | 67.38 | 67.99 | 67.68 |
| pFedMMA | 68.14 | 71.39 | 69.49 | 95.92 | 94.49 | 95.20 | 43.23 | 44.34 | 43.78 | 42.42 | 33.83 | 37.64 | 41.61 | 41.74 | 41.67 |
| FedMaPLe | 82.76 | 77.80 | 79.93 | 97.28 | 94.06 | 95.64 | 78.84 | 67.78 | 72.89 | 77.07 | 64.04 | 69.95 | 78.42 | 68.56 | 73.16 |

| Full_Dir(0.1) | OxfordPet | | | Flower102 | | | Caltech101 | | | Caltech256 | | | Food101 | | |
|---|---|---|---|---|---|---|---|---|---|---|---|---|---|---|---|
| | Base | Novel | HM | Base | Novel | HM | Base | Novel | HM | Base | Novel | HM | Base | Novel | HM |
| pFedDC | 79.33 | 91.76 | 85.09 | 55.61 | 63.10 | 59.12 | 88.40 | 92.28 | 90.30 | 89.78 | 84.33 | 86.97 | 75.71 | 82.88 | 79.13 |
| pFedMMA | 86.86 | 96.58 | 91.46 | 64.60 | 75.38 | 69.57 | 88.73 | 93.95 | 91.27 | 90.20 | 85.00 | 87.52 | 59.68 | 77.16 | 67.30 |
| FedMaPLe | 87.32 | 95.28 | 91.13 | 61.18 | 63.92 | 62.52 | 86.99 | 92.59 | 89.70 | 87.44 | 66.04 | 75.25 | 90.32 | 87.89 | 89.09 |

| Full_Non-IID | Average | | | CIFAR10 | | | CIFAR100 | | | EuroSAT | | | Tiny-ImageNet | | |
|---|---|---|---|---|---|---|---|---|---|---|---|---|---|---|---|
| | Base | Novel | HM | Base | Novel | HM | Base | Novel | HM | Base | Novel | HM | Base | Novel | HM |
| pFedDC | 56.74 | 62.56 | 59.31 | 26.01 | 33.79 | 29.39 | 58.78 | 59.86 | 59.32 | 20.00 | 23.42 | 21.58 | 59.29 | 60.24 | 59.76 |
| pFedMMA | 39.41 | 53.63 | 43.61 | 20.00 | 54.81 | 29.31 | 30.00 | 30.79 | 30.39 | 20.00 | 34.62 | 25.35 | 2.06 | 1.53 | 1.76 |
| FedMaPLe | 79.83 | 73.96 | 76.36 | 95.74 | 93.00 | 94.35 | 81.40 | 68.52 | 74.41 | 54.46 | 30.42 | 39.04 | 75.20 | 65.56 | 70.05 |

| Full_Non-IID | OxfordPet | | | Flower102 | | | Caltech101 | | | Caltech256 | | | Food101 | | |
|---|---|---|---|---|---|---|---|---|---|---|---|---|---|---|---|
| | Base | Novel | HM | Base | Novel | HM | Base | Novel | HM | Base | Novel | HM | Base | Novel | HM |
| pFedDC | 66.84 | 83.90 | 74.40 | 42.34 | 55.09 | 47.88 | 82.58 | 89.94 | 86.10 | 90.35 | 83.04 | 86.54 | 64.46 | 73.73 | 68.78 |
| pFedMMA | 32.16 | 83.76 | 46.48 | 38.79 | 62.78 | 47.95 | 75.54 | 90.22 | 82.23 | 72.14 | 75.63 | 73.84 | 64.01 | 48.49 | 55.18 |
| FedMaPLe | 78.41 | 81.07 | 79.72 | 64.45 | 69.13 | 66.71 | 87.43 | 89.35 | 88.38 | 92.09 | 80.52 | 85.92 | 89.26 | 88.10 | 88.68 |

*Table 26.* The per-dataset test accuracy (%) of different training accuracy threshold $\varepsilon_{acc}$ under four data settings. "Few" and "Full" represent few-shot and full-data, while "IID" and "Non-IID" represent IID and Non-IID data setting.

| Few_IID | OxfordPet | | | Flower102 | | | CIFAR100 | | | Caltech101 | | | Caltech256 | | |
|---|---|---|---|---|---|---|---|---|---|---|---|---|---|---|---|
| | Base | Novel | HM | Base | Novel | HM | Base | Novel | HM | Base | Novel | HM | Base | Novel | HM |
| $\varepsilon_{acc}$=0.003 | 95.68 | 96.18 | 95.93 | 97.19 | 67.32 | 79.54 | 85.06 | 75.96 | 80.25 | 97.42 | 94.08 | 95.72 | 94.88 | 84.61 | 89.45 |
| $\varepsilon_{acc}$=0.005 | 96.07 | 96.66 | 96.36 | 97.06 | 72.13 | 82.76 | 84.96 | 76.04 | 80.25 | 97.42 | 94.14 | 95.75 | 94.95 | 84.91 | 89.65 |
| $\varepsilon_{acc}$=0.007 | 96.41 | 95.34 | 95.87 | 96.96 | 68.09 | 80.00 | 85.42 | 77.36 | 81.19 | 97.59 | 93.67 | 95.59 | 94.81 | 84.19 | 89.18 |
| $\varepsilon_{acc}$=0.009 | 95.98 | 97.00 | 96.49 | 97.25 | 68.04 | 80.06 | 85.09 | 77.18 | 80.94 | 96.26 | 93.67 | 94.95 | 95.22 | 84.49 | 89.53 |

| Few_Non-IID | OxfordPet | | | Flower102 | | | CIFAR100 | | | Caltech101 | | | Caltech256 | | |
|---|---|---|---|---|---|---|---|---|---|---|---|---|---|---|---|
| | Base | Novel | HM | Base | Novel | HM | Base | Novel | HM | Base | Novel | HM | Base | Novel | HM |
| $\varepsilon_{acc}$=0.003 | 95.28 | 96.39 | 95.83 | 94.33 | 70.64 | 80.78 | 82.21 | 77.90 | 80.00 | 95.69 | 94.44 | 95.06 | 94.39 | 85.80 | 89.89 |
| $\varepsilon_{acc}$=0.005 | 95.14 | 96.48 | 95.81 | 93.60 | 70.19 | 80.22 | 82.18 | 77.14 | 79.58 | 95.81 | 94.60 | 95.20 | 94.56 | 85.33 | 89.71 |
| $\varepsilon_{acc}$=0.007 | 94.80 | 95.29 | 95.04 | 96.00 | 69.20 | 80.43 | 83.12 | 77.00 | 79.94 | 95.63 | 94.60 | 95.11 | 94.31 | 85.54 | 89.71 |
| $\varepsilon_{acc}$=0.009 | 94.94 | 96.11 | 95.52 | 94.77 | 70.07 | 80.57 | 82.84 | 77.72 | 80.20 | 95.45 | 94.60 | 95.02 | 94.41 | 85.07 | 89.50 |

| Full_IID | OxfordPet | | | Flower102 | | | CIFAR100 | | | Caltech101 | | | Caltech256 | | |
|---|---|---|---|---|---|---|---|---|---|---|---|---|---|---|---|
| | Base | Novel | HM | Base | Novel | HM | Base | Novel | HM | Base | Novel | HM | Base | Novel | HM |
| $\varepsilon_{acc}$=0.003 | 96.18 | 94.22 | 95.19 | 97.45 | 67.45 | 79.72 | 87.36 | 74.28 | 80.29 | 98.13 | 93.21 | 95.61 | 94.98 | 83.53 | 88.89 |
| $\varepsilon_{acc}$=0.005 | 96.18 | 97.19 | 96.68 | 96.96 | 65.82 | 78.41 | 87.38 | 73.76 | 79.99 | 97.95 | 93.06 | 95.44 | 94.81 | 85.00 | 89.64 |
| $\varepsilon_{acc}$=0.007 | 96.02 | 96.66 | 96.34 | 96.39 | 67.09 | 79.11 | 87.32 | 75.52 | 80.99 | 98.13 | 94.29 | 96.17 | 94.70 | 84.79 | 89.47 |
| $\varepsilon_{acc}$=0.009 | 96.24 | 97.30 | 96.77 | 97.06 | 67.65 | 79.73 | 87.12 | 74.80 | 80.49 | 98.04 | 93.36 | 95.64 | 95.15 | 83.56 | 88.98 |

| Full_Non-IID | OxfordPet | | | Flower102 | | | CIFAR100 | | | Caltech101 | | | Caltech256 | | |
|---|---|---|---|---|---|---|---|---|---|---|---|---|---|---|---|
| | Base | Novel | HM | Base | Novel | HM | Base | Novel | HM | Base | Novel | HM | Base | Novel | HM |
| $\varepsilon_{acc}$=0.003 | 95.01 | 96.57 | 95.78 | 95.80 | 67.88 | 79.46 | 82.60 | 72.62 | 77.29 | 97.42 | 92.13 | 94.70 | 94.67 | 85.59 | 89.90 |
| $\varepsilon_{acc}$=0.005 | 95.01 | 96.30 | 95.65 | 97.00 | 64.81 | 77.70 | 83.52 | 71.56 | 77.08 | 97.68 | 90.74 | 94.08 | 94.67 | 83.73 | 88.86 |
| $\varepsilon_{acc}$=0.007 | 95.07 | 96.52 | 95.79 | 96.31 | 70.68 | 81.53 | 81.38 | 71.36 | 76.04 | 97.50 | 90.74 | 94.00 | 94.77 | 83.90 | 89.00 |
| $\varepsilon_{acc}$=0.009 | 95.61 | 96.66 | 96.13 | 96.11 | 69.13 | 80.42 | 82.00 | 72.28 | 76.83 | 97.50 | 92.59 | 94.98 | 94.77 | 84.49 | 89.34 |

*Table 27.* The per-dataset test accuracy (%) of different RL algorithms applied into our RL stage under four data settings. "Few" and "Full" represent few-shot and full-data, while "IID" and "Non-IID" represent IID and Non-IID data settings.

| Few_IID | OxfordPet | | | Flower102 | | | CIFAR100 | | | Caltech101 | | | Caltech256 | | |
|---|---|---|---|---|---|---|---|---|---|---|---|---|---|---|---|
| | Base | Novel | HM | Base | Novel | HM | Base | Novel | HM | Base | Novel | HM | Base | Novel | HM |
| GRPO | 95.98 | 97.00 | 96.49 | 97.19 | 67.32 | 79.54 | 85.09 | 77.18 | 80.94 | 97.42 | 94.08 | 95.72 | 94.88 | 84.61 | 89.45 |
| DR_GRPO | 96.41 | 97.56 | 96.98 | 97.65 | 64.51 | 77.69 | 84.64 | 76.00 | 80.09 | 97.50 | 94.14 | 95.79 | 94.88 | 82.15 | 88.06 |
| GMPO | 96.02 | 96.98 | 96.50 | 97.25 | 66.47 | 78.97 | 85.14 | 75.02 | 79.76 | 97.50 | 94.29 | 95.87 | 94.77 | 83.59 | 88.83 |
| DAPO | 95.45 | 96.82 | 96.13 | 97.45 | 69.61 | 81.21 | 85.58 | 75.26 | 80.09 | 96.79 | 93.67 | 95.20 | 95.02 | 85.15 | 89.81 |
| LitePPO | 96.07 | 97.35 | 96.71 | 96.67 | 70.00 | 81.20 | 84.60 | 76.30 | 80.24 | 97.59 | 93.67 | 95.59 | 94.81 | 84.61 | 89.42 |

| Few_Non-IID | OxfordPet | | | Flower102 | | | CIFAR100 | | | Caltech101 | | | Caltech256 | | |
|---|---|---|---|---|---|---|---|---|---|---|---|---|---|---|---|
| | Base | Novel | HM | Base | Novel | HM | Base | Novel | HM | Base | Novel | HM | Base | Novel | HM |
| GRPO | 95.28 | 96.39 | 95.83 | 94.33 | 70.64 | 80.78 | 82.21 | 77.90 | 80.00 | 95.69 | 94.44 | 95.06 | 94.39 | 85.80 | 89.89 |
| DR_GRPO | 95.28 | 95.79 | 95.53 | 94.20 | 65.77 | 77.46 | 82.22 | 77.46 | 79.77 | 94.83 | 94.29 | 94.56 | 94.38 | 85.51 | 89.73 |
| GMPO | 95.61 | 96.16 | 95.88 | 93.80 | 71.35 | 81.05 | 81.96 | 77.02 | 79.41 | 95.63 | 93.83 | 94.72 | 94.27 | 85.59 | 89.72 |
| DAPO | 95.07 | 96.48 | 95.77 | 94.20 | 66.92 | 78.25 | 83.02 | 76.88 | 79.83 | 95.28 | 94.14 | 94.71 | 94.70 | 85.71 | 89.98 |
| LitePPO | 95.41 | 94.97 | 95.19 | 94.80 | 71.35 | 81.42 | 82.40 | 77.90 | 80.09 | 95.63 | 94.44 | 95.03 | 94.02 | 85.89 | 89.77 |

| Full_IID | OxfordPet | | | Flower102 | | | CIFAR100 | | | Caltech101 | | | Caltech256 | | |
|---|---|---|---|---|---|---|---|---|---|---|---|---|---|---|---|
| | Base | Novel | HM | Base | Novel | HM | Base | Novel | HM | Base | Novel | HM | Base | Novel | HM |
| GRPO | 96.24 | 97.30 | 96.77 | 97.45 | 67.45 | 79.72 | 87.12 | 74.80 | 80.49 | 98.13 | 93.21 | 95.61 | 94.98 | 83.53 | 88.89 |
| DR_GRPO | 95.57 | 95.55 | 95.56 | 97.25 | 66.27 | 78.83 | 86.62 | 74.12 | 79.88 | 98.40 | 92.90 | 95.57 | 94.91 | 82.93 | 88.52 |
| GMPO | 96.13 | 96.24 | 96.18 | 96.27 | 68.04 | 79.73 | 87.26 | 74.30 | 80.26 | 98.22 | 93.36 | 95.73 | 94.77 | 84.40 | 89.28 |
| DAPO | 96.07 | 95.81 | 95.94 | 96.67 | 68.24 | 80.00 | 87.28 | 71.88 | 78.83 | 98.13 | 92.59 | 95.28 | 94.84 | 83.38 | 88.74 |
| LitePPO | 96.02 | 96.45 | 96.23 | 96.67 | 68.43 | 80.13 | 87.44 | 73.86 | 80.08 | 98.22 | 92.13 | 95.08 | 94.88 | 84.07 | 89.15 |

| Full_Non-IID | OxfordPet | | | Flower102 | | | CIFAR100 | | | Caltech101 | | | Caltech256 | | |
|---|---|---|---|---|---|---|---|---|---|---|---|---|---|---|---|
| | Base | Novel | HM | Base | Novel | HM | Base | Novel | HM | Base | Novel | HM | Base | Novel | HM |
| GRPO | 95.01 | 96.57 | 95.78 | 95.80 | 67.88 | 79.46 | 82.60 | 72.62 | 77.29 | 97.42 | 92.13 | 94.70 | 94.67 | 85.59 | 89.90 |
| DR_GRPO | 95.14 | 96.80 | 95.96 | 96.40 | 67.31 | 79.27 | 82.46 | 69.00 | 75.13 | 96.79 | 89.66 | 93.09 | 95.20 | 85.16 | 89.90 |
| GMPO | 94.60 | 96.80 | 95.69 | 96.60 | 64.42 | 77.29 | 82.88 | 72.30 | 77.23 | 96.17 | 91.05 | 93.54 | 94.74 | 84.95 | 89.58 |
| DAPO | 94.87 | 92.96 | 93.91 | 96.40 | 66.54 | 78.73 | 80.16 | 72.26 | 76.01 | 97.50 | 90.90 | 94.08 | 94.84 | 84.05 | 89.12 |
| LitePPO | 95.01 | 95.02 | 95.01 | 94.40 | 62.31 | 75.07 | 82.44 | 70.82 | 76.19 | 97.77 | 93.36 | 95.51 | 94.81 | 85.39 | 89.85 |

*Table 28.* The per-dataset test accuracy (%) of different text encoder backbones under four data settings. "Few" and "Full" represent few-shot and full-data, while "IID" and "Non-IID" represent IID and Non-IID data settings.

| Few_IID | OxfordPet | | | Flower102 | | | CIFAR100 | | | Caltech101 | | | Caltech256 | | |
|---|---|---|---|---|---|---|---|---|---|---|---|---|---|---|---|
| | Base | Novel | HM | Base | Novel | HM | Base | Novel | HM | Base | Novel | HM | Base | Novel | HM |
| ViT-B/32 | 94.84 | 77.58 | 85.35 | 98.43 | 31.76 | 48.02 | 86.00 | 48.10 | 61.69 | 98.75 | 62.65 | 76.66 | 95.22 | 61.75 | 74.92 |
| ViT-L/14 | 95.57 | 62.80 | 75.79 | 98.24 | 31.76 | 48.00 | 85.70 | 46.96 | 60.67 | 96.88 | 64.35 | 77.33 | 95.40 | 65.11 | 77.40 |
| ViT-B/16 | 95.98 | 97.00 | 96.49 | 97.19 | 67.32 | 79.54 | 85.09 | 77.18 | 80.94 | 97.42 | 94.08 | 95.72 | 94.88 | 84.61 | 89.45 |

| Few_Non-IID | OxfordPet | | | Flower102 | | | CIFAR100 | | | Caltech101 | | | Caltech256 | | |
|---|---|---|---|---|---|---|---|---|---|---|---|---|---|---|---|
| | Base | Novel | HM | Base | Novel | HM | Base | Novel | HM | Base | Novel | HM | Base | Novel | HM |
| ViT-B/32 | 88.06 | 53.82 | 66.81 | 96.40 | 25.96 | 40.90 | 82.50 | 42.06 | 55.72 | 94.03 | 57.10 | 71.05 | 94.09 | 56.05 | 70.25 |
| ViT-L/14 | 85.22 | 46.91 | 60.51 | 96.60 | 29.62 | 45.34 | 82.92 | 41.36 | 55.19 | 93.32 | 61.88 | 74.42 | 94.13 | 58.44 | 72.11 |
| ViT-B/16 | 95.28 | 96.39 | 95.83 | 94.33 | 70.64 | 80.78 | 82.21 | 77.90 | 80.00 | 95.69 | 94.44 | 95.06 | 94.39 | 85.80 | 89.89 |

| Full_IID | OxfordPet | | | Flower102 | | | CIFAR100 | | | Caltech101 | | | Caltech256 | | |
|---|---|---|---|---|---|---|---|---|---|---|---|---|---|---|---|
| | Base | Novel | HM | Base | Novel | HM | Base | Novel | HM | Base | Novel | HM | Base | Novel | HM |
| ViT-B/32 | 94.73 | 69.90 | 80.44 | 97.84 | 25.10 | 39.95 | 87.84 | 47.20 | 61.40 | 98.66 | 58.49 | 73.44 | 95.26 | 63.37 | 76.11 |
| ViT-L/14 | 95.40 | 76.84 | 85.12 | 97.84 | 31.76 | 47.95 | 87.66 | 42.04 | 56.83 | 98.31 | 56.79 | 71.99 | 94.88 | 63.85 | 76.33 |
| ViT-B/16 | 96.24 | 97.30 | 96.77 | 97.45 | 67.45 | 79.72 | 87.12 | 74.80 | 80.49 | 98.13 | 93.21 | 95.61 | 94.98 | 83.53 | 88.89 |

| Full_Non-IID | OxfordPet | | | Flower102 | | | CIFAR100 | | | Caltech101 | | | Caltech256 | | |
|---|---|---|---|---|---|---|---|---|---|---|---|---|---|---|---|
| | Base | Novel | HM | Base | Novel | HM | Base | Novel | HM | Base | Novel | HM | Base | Novel | HM |
| ViT-B/32 | 94.40 | 54.73 | 69.29 | 97.40 | 23.46 | 37.81 | 81.52 | 40.58 | 54.19 | 97.68 | 54.78 | 70.19 | 94.95 | 59.32 | 73.02 |
| ViT-L/14 | 95.21 | 56.24 | 70.71 | 97.60 | 31.54 | 47.67 | 79.52 | 38.36 | 51.75 | 97.50 | 54.01 | 69.51 | 95.24 | 53.51 | 68.52 |
| ViT-B/16 | 95.01 | 96.57 | 95.78 | 95.80 | 67.88 | 79.46 | 82.60 | 72.62 | 77.29 | 97.42 | 92.13 | 94.70 | 94.67 | 85.59 | 89.90 |

*Table 29.* The per-dataset test accuracy (%) of different client number $K$ under four data settings. "Few" and "Full" represent few-shot and full-data, while "IID" and "Non-IID" represent IID and Non-IID data settings.

| Few_IID | Flower102 | | | CIFAR100 | | | Caltech101 | | | Caltech256 | | |
|---|---|---|---|---|---|---|---|---|---|---|---|---|
| | Base | Novel | HM | Base | Novel | HM | Base | Novel | HM | Base | Novel | HM |
| $K$=5 | 97.19 | 67.32 | 79.54 | 85.09 | 77.18 | 80.94 | 97.42 | 94.08 | 95.72 | 94.88 | 84.61 | 89.45 |
| $K$=10 | 97.25 | 66.08 | 78.69 | 87.06 | 79.42 | 83.06 | 98.31 | 94.29 | 96.26 | 95.60 | 82.99 | 88.85 |
| $K$=15 | 97.25 | 68.04 | 80.06 | 87.46 | 78.50 | 82.74 | 98.48 | 94.14 | 96.26 | 95.60 | 84.13 | 89.50 |

| Few_Non-IID | Flower102 | | | CIFAR100 | | | Caltech101 | | | Caltech256 | | |
|---|---|---|---|---|---|---|---|---|---|---|---|---|
| | Base | Novel | HM | Base | Novel | HM | Base | Novel | HM | Base | Novel | HM |
| $K$=5 | 94.33 | 70.64 | 80.78 | 82.21 | 77.90 | 80.00 | 95.69 | 94.44 | 95.06 | 94.39 | 85.80 | 89.89 |
| $K$=10 | 94.00 | 67.88 | 78.83 | 83.54 | 77.78 | 80.56 | 95.45 | 93.98 | 94.71 | 95.17 | 86.23 | 90.48 |
| $K$=15 | 96.67 | 67.89 | 79.76 | 85.02 | 77.36 | 81.01 | 94.67 | 95.13 | 94.90 | 95.77 | 86.01 | 90.63 |

| Full_IID | Flower102 | | | CIFAR100 | | | Caltech101 | | | Caltech256 | | |
|---|---|---|---|---|---|---|---|---|---|---|---|---|
| | Base | Novel | HM | Base | Novel | HM | Base | Novel | HM | Base | Novel | HM |
| $K$=5 | 97.45 | 67.45 | 79.72 | 87.12 | 74.80 | 80.49 | 98.13 | 93.21 | 95.61 | 94.98 | 83.53 | 88.89 |
| $K$=10 | 97.25 | 63.92 | 77.14 | 88.24 | 77.62 | 82.59 | 97.95 | 93.21 | 95.52 | 95.47 | 84.97 | 89.91 |
| $K$=15 | 97.45 | 66.08 | 78.76 | 88.28 | 77.48 | 82.53 | 98.93 | 94.29 | 96.55 | 95.78 | 85.00 | 90.07 |

| Full_Non-IID | Flower102 | | | CIFAR100 | | | Caltech101 | | | Caltech256 | | |
|---|---|---|---|---|---|---|---|---|---|---|---|---|
| | Base | Novel | HM | Base | Novel | HM | Base | Novel | HM | Base | Novel | HM |
| $K$=5 | 95.80 | 67.88 | 79.46 | 82.60 | 72.62 | 77.29 | 97.42 | 92.13 | 94.70 | 94.67 | 85.59 | 89.90 |
| $K$=10 | 96.80 | 66.35 | 78.73 | 84.88 | 74.98 | 79.62 | 90.02 | 92.28 | 91.14 | 95.54 | 84.99 | 89.96 |
| $K$=15 | 97.78 | 65.09 | 78.15 | 87.76 | 75.93 | 81.42 | 98.19 | 93.46 | 95.77 | 95.88 | 85.33 | 90.30 |

*Table 30.* The per-dataset test accuracy (%) of different LoRA ranks $r$ under four data settings. "Few" and "Full" represent few-shot and full-data, while "IID" and "Non-IID" represent IID and Non-IID data settings.

| Few_IID | OxfordPet | | | Flower102 | | | CIFAR100 | | | Caltech101 | | | Caltech256 | | |
|---|---|---|---|---|---|---|---|---|---|---|---|---|---|---|---|
| | Base | Novel | HM | Base | Novel | HM | Base | Novel | HM | Base | Novel | HM | Base | Novel | HM |
| r=2 | 96.46 | 94.65 | 95.55 | 96.86 | 69.61 | 81.00 | 84.12 | 76.38 | 80.06 | 97.59 | 93.52 | 95.51 | 94.67 | 82.48 | 88.16 |
| r=4 | 95.98 | 97.00 | 96.49 | 97.19 | 67.32 | 79.54 | 85.09 | 77.18 | 80.94 | 97.42 | 94.08 | 95.72 | 94.88 | 84.61 | 89.45 |
| r=8 | 95.85 | 96.71 | 96.28 | 96.47 | 66.67 | 78.85 | 85.00 | 75.34 | 79.88 | 96.79 | 93.52 | 95.13 | 95.12 | 83.44 | 88.90 |
| r=16 | 95.57 | 96.87 | 96.22 | 97.06 | 67.84 | 79.86 | 85.38 | 76.04 | 80.44 | 96.52 | 93.52 | 95.00 | 95.19 | 84.31 | 89.42 |

| Few_Non-IID | OxfordPet | | | Flower102 | | | CIFAR100 | | | Caltech101 | | | Caltech256 | | |
|---|---|---|---|---|---|---|---|---|---|---|---|---|---|---|---|
| | Base | Novel | HM | Base | Novel | HM | Base | Novel | HM | Base | Novel | HM | Base | Novel | HM |
| r=2 | 94.87 | 95.38 | 95.12 | 92.80 | 71.73 | 80.92 | 82.08 | 77.62 | 79.79 | 94.03 | 93.52 | 93.77 | 94.02 | 85.89 | 89.77 |
| r=4 | 95.28 | 96.39 | 95.83 | 94.33 | 70.64 | 80.78 | 82.21 | 77.90 | 80.00 | 95.69 | 94.44 | 95.06 | 94.39 | 85.80 | 89.89 |
| r=8 | 94.94 | 96.52 | 95.72 | 95.40 | 63.46 | 76.22 | 82.24 | 77.36 | 79.73 | 93.94 | 94.14 | 94.04 | 94.20 | 85.80 | 89.80 |
| r=16 | 95.07 | 96.39 | 95.73 | 96.60 | 68.08 | 79.87 | 82.90 | 77.74 | 80.24 | 95.10 | 94.44 | 94.77 | 94.52 | 85.54 | 89.81 |

| Full_IID | OxfordPet | | | Flower102 | | | CIFAR100 | | | Caltech101 | | | Caltech256 | | |
|---|---|---|---|---|---|---|---|---|---|---|---|---|---|---|---|
| | Base | Novel | HM | Base | Novel | HM | Base | Novel | HM | Base | Novel | HM | Base | Novel | HM |
| r=2 | 96.18 | 97.14 | 96.66 | 96.27 | 68.82 | 80.26 | 86.60 | 74.02 | 79.82 | 97.77 | 92.75 | 95.19 | 94.74 | 84.31 | 89.22 |
| r=4 | 96.24 | 97.30 | 96.77 | 97.45 | 67.45 | 79.72 | 87.12 | 74.80 | 80.49 | 98.13 | 93.21 | 95.61 | 94.98 | 83.53 | 88.89 |
| r=8 | 95.90 | 97.09 | 96.49 | 96.47 | 65.88 | 78.29 | 87.84 | 74.94 | 80.88 | 97.77 | 93.36 | 95.51 | 95.26 | 84.61 | 89.62 |
| r=16 | 96.07 | 97.30 | 96.68 | 97.45 | 67.25 | 79.58 | 88.14 | 75.12 | 81.11 | 98.48 | 93.52 | 95.94 | 94.95 | 83.29 | 88.74 |

| Full_Non-IID | OxfordPet | | | Flower102 | | | CIFAR100 | | | Caltech101 | | | Caltech256 | | |
|---|---|---|---|---|---|---|---|---|---|---|---|---|---|---|---|
| | Base | Novel | HM | Base | Novel | HM | Base | Novel | HM | Base | Novel | HM | Base | Novel | HM |
| r=2 | 95.14 | 96.94 | 96.03 | 95.20 | 62.88 | 75.74 | 82.36 | 66.22 | 73.41 | 97.59 | 91.82 | 94.62 | 94.88 | 84.46 | 89.37 |
| r=4 | 95.01 | 96.57 | 95.78 | 95.80 | 67.88 | 79.46 | 82.60 | 72.62 | 77.29 | 97.42 | 92.13 | 94.70 | 94.67 | 85.59 | 89.90 |
| r=8 | 94.94 | 96.94 | 95.93 | 96.80 | 65.19 | 77.91 | 83.34 | 72.28 | 77.42 | 97.06 | 91.67 | 94.29 | 94.92 | 81.86 | 87.91 |
| r=16 | 95.41 | 97.49 | 96.44 | 96.80 | 59.04 | 73.35 | 83.54 | 72.50 | 77.63 | 96.97 | 89.97 | 93.34 | 94.74 | 84.69 | 89.43 |

*Table 31.* The per-dataset test accuracy (%) of different LoRA starting layers $l$ under four data settings. "Few" and "Full" represent few-shot and full-data, while "IID" and "Non-IID" represent IID and Non-IID data settings.

| Few_IID | OxfordPet | | | Flower102 | | | CIFAR100 | | | Caltech101 | | | Caltech256 | | |
|---|---|---|---|---|---|---|---|---|---|---|---|---|---|---|---|
| | Base | Novel | HM | Base | Novel | HM | Base | Novel | HM | Base | Novel | HM | Base | Novel | HM |
| l=2 | 95.68 | 96.82 | 96.25 | 98.04 | 65.49 | 78.53 | 88.38 | 77.70 | 82.70 | 98.75 | 92.13 | 95.33 | 95.22 | 84.28 | 89.42 |
| l=6 | 96.24 | 95.65 | 95.94 | 97.45 | 67.06 | 79.45 | 87.02 | 77.66 | 82.07 | 98.48 | 92.28 | 95.28 | 95.85 | 83.77 | 89.40 |
| l=10 | 95.98 | 97.00 | 96.49 | 97.19 | 67.32 | 79.54 | 85.09 | 77.18 | 80.94 | 97.42 | 94.08 | 95.72 | 94.88 | 84.61 | 89.45 |

| Few_Non-IID | OxfordPet | | | Flower102 | | | CIFAR100 | | | Caltech101 | | | Caltech256 | | |
|---|---|---|---|---|---|---|---|---|---|---|---|---|---|---|---|
| | Base | Novel | HM | Base | Novel | HM | Base | Novel | HM | Base | Novel | HM | Base | Novel | HM |
| l=2 | 94.20 | 95.34 | 94.77 | 95.40 | 65.58 | 77.73 | 85.10 | 77.32 | 81.02 | 93.49 | 93.52 | 93.50 | 94.20 | 84.14 | 88.89 |
| l=6 | 94.26 | 96.20 | 95.22 | 95.20 | 69.23 | 80.16 | 83.62 | 76.68 | 80.00 | 94.30 | 93.52 | 93.91 | 94.09 | 85.48 | 89.58 |
| l=10 | 95.28 | 96.39 | 95.83 | 94.33 | 70.64 | 80.78 | 82.21 | 77.90 | 80.00 | 95.69 | 94.44 | 95.06 | 94.39 | 85.80 | 89.89 |

| Full_IID | OxfordPet | | | Flower102 | | | CIFAR100 | | | Caltech101 | | | Caltech256 | | |
|---|---|---|---|---|---|---|---|---|---|---|---|---|---|---|---|
| | Base | Novel | HM | Base | Novel | HM | Base | Novel | HM | Base | Novel | HM | Base | Novel | HM |
| l=2 | 96.02 | 96.61 | 96.31 | 97.84 | 66.67 | 79.30 | 91.16 | 76.12 | 82.96 | 98.57 | 93.83 | 96.14 | 95.71 | 84.28 | 89.63 |
| l=6 | 95.34 | 95.97 | 95.65 | 99.02 | 61.18 | 75.63 | 89.48 | 74.46 | 81.28 | 98.48 | 92.90 | 95.61 | 96.05 | 83.68 | 89.44 |
| l=10 | 96.24 | 97.30 | 96.77 | 97.45 | 67.45 | 79.72 | 87.12 | 74.80 | 80.49 | 98.13 | 93.21 | 95.61 | 94.98 | 83.53 | 88.89 |

| Full_Non-IID | OxfordPet | | | Flower102 | | | CIFAR100 | | | Caltech101 | | | Caltech256 | | |
|---|---|---|---|---|---|---|---|---|---|---|---|---|---|---|---|
| | Base | Novel | HM | Base | Novel | HM | Base | Novel | HM | Base | Novel | HM | Base | Novel | HM |
| l=2 | 93.59 | 96.16 | 94.86 | 96.00 | 64.42 | 77.10 | 83.66 | 73.56 | 78.29 | 96.97 | 92.28 | 94.57 | 95.20 | 85.51 | 90.10 |
| l=6 | 93.52 | 91.59 | 92.54 | 96.40 | 59.42 | 73.52 | 83.22 | 72.90 | 77.72 | 97.06 | 90.59 | 93.71 | 94.63 | 83.41 | 88.67 |
| l=10 | 95.01 | 96.57 | 95.78 | 95.80 | 67.88 | 79.46 | 82.60 | 72.62 | 77.29 | 97.42 | 92.13 | 94.70 | 94.67 | 85.59 | 89.90 |

*Table 32.* The per-dataset test accuracy (%) of different sampling counts $G$ in the RL stage under four data settings. "Few" and "Full" represent few-shot and full-data, while "IID" and "Non-IID" represent IID and Non-IID data setting.

| Few_IID | OxfordPet | | | Flower102 | | | CIFAR100 | | | Caltech101 | | | Caltech256 | | |
|---|---|---|---|---|---|---|---|---|---|---|---|---|---|---|---|
| | Base | Novel | HM | Base | Novel | HM | Base | Novel | HM | Base | Novel | HM | Base | Novel | HM |
| G=2 | 95.96 | 95.71 | 95.83 | 96.47 | 68.63 | 80.20 | 85.04 | 77.06 | 80.85 | 97.06 | 93.06 | 95.02 | 94.84 | 83.77 | 88.96 |
| G=3 | 95.98 | 97.00 | 96.49 | 97.19 | 67.32 | 79.54 | 85.09 | 77.18 | 80.94 | 97.42 | 94.08 | 95.72 | 94.88 | 84.61 | 89.45 |
| G=5 | 96.24 | 96.13 | 96.18 | 97.25 | 65.29 | 78.13 | 85.70 | 77.62 | 81.46 | 96.70 | 95.06 | 95.87 | 94.74 | 84.31 | 89.22 |
| G=7 | 96.02 | 95.92 | 95.97 | 97.65 | 65.49 | 78.40 | 84.90 | 76.78 | 80.64 | 97.33 | 93.83 | 95.55 | 95.22 | 84.31 | 89.43 |

| Few_Non-IID | OxfordPet | | | Flower102 | | | CIFAR100 | | | Caltech101 | | | Caltech256 | | |
|---|---|---|---|---|---|---|---|---|---|---|---|---|---|---|---|
| | Base | Novel | HM | Base | Novel | HM | Base | Novel | HM | Base | Novel | HM | Base | Novel | HM |
| G=2 | 95.48 | 96.16 | 95.82 | 95.40 | 69.23 | 80.23 | 82.22 | 77.46 | 79.77 | 94.92 | 94.75 | 94.83 | 94.49 | 84.19 | 89.04 |
| G=3 | 95.28 | 96.39 | 95.83 | 94.33 | 70.64 | 80.78 | 82.21 | 77.90 | 80.00 | 95.69 | 94.44 | 95.06 | 94.39 | 85.80 | 89.89 |
| G=5 | 94.94 | 95.79 | 95.36 | 95.20 | 71.73 | 81.82 | 82.70 | 76.92 | 79.71 | 95.19 | 95.22 | 95.20 | 94.13 | 85.80 | 89.77 |
| G=7 | 95.28 | 94.88 | 95.08 | 95.20 | 71.92 | 81.94 | 81.92 | 77.60 | 79.70 | 93.14 | 94.44 | 93.79 | 94.02 | 83.70 | 88.56 |

| Full_IID | OxfordPet | | | Flower102 | | | CIFAR100 | | | Caltech101 | | | Caltech256 | | |
|---|---|---|---|---|---|---|---|---|---|---|---|---|---|---|---|
| | Base | Novel | HM | Base | Novel | HM | Base | Novel | HM | Base | Novel | HM | Base | Novel | HM |
| G=2 | 96.13 | 96.50 | 96.31 | 96.86 | 65.49 | 78.14 | 87.32 | 74.04 | 80.13 | 98.04 | 93.67 | 95.81 | 94.60 | 82.24 | 87.99 |
| G=3 | 96.24 | 97.30 | 96.77 | 97.45 | 67.45 | 79.72 | 87.12 | 74.80 | 80.49 | 98.13 | 93.21 | 95.61 | 94.98 | 83.53 | 88.89 |
| G=5 | 96.30 | 96.77 | 96.53 | 96.47 | 63.53 | 76.61 | 87.16 | 74.54 | 80.36 | 98.31 | 92.13 | 95.12 | 94.70 | 83.50 | 88.75 |
| G=7 | 95.96 | 93.48 | 94.70 | 95.88 | 66.67 | 78.65 | 87.50 | 74.10 | 80.24 | 97.95 | 93.06 | 95.44 | 94.98 | 83.80 | 89.04 |

| Full_Non-IID | OxfordPet | | | Flower102 | | | CIFAR100 | | | Caltech101 | | | Caltech256 | | |
|---|---|---|---|---|---|---|---|---|---|---|---|---|---|---|---|
| | Base | Novel | HM | Base | Novel | HM | Base | Novel | HM | Base | Novel | HM | Base | Novel | HM |
| G=2 | 94.87 | 97.07 | 95.96 | 96.60 | 57.69 | 72.24 | 81.86 | 71.22 | 76.17 | 97.95 | 92.75 | 95.28 | 94.41 | 84.66 | 89.27 |
| G=3 | 95.01 | 96.57 | 95.78 | 95.80 | 67.88 | 79.46 | 82.60 | 72.62 | 77.29 | 97.42 | 92.13 | 94.70 | 94.67 | 85.59 | 89.90 |
| G=5 | 95.07 | 95.75 | 95.41 | 97.00 | 58.08 | 72.66 | 83.24 | 73.00 | 77.78 | 97.33 | 91.20 | 94.17 | 94.49 | 85.48 | 89.76 |
| G=7 | 95.14 | 97.62 | 96.36 | 95.20 | 64.81 | 77.12 | 83.24 | 73.14 | 77.86 | 97.24 | 92.28 | 94.70 | 95.42 | 85.51 | 90.19 |

*Table 33.* The per-dataset test accuracy (%) of different noise scales $\sigma$ in the RL stage under four data settings. "Few" and "Full" represent few-shot and full-data, while "IID" and "Non-IID" represent IID and Non-IID data settings.

| Few_IID | OxfordPet | | | Flower102 | | | CIFAR100 | | | Caltech101 | | | Caltech256 | | |
|---|---|---|---|---|---|---|---|---|---|---|---|---|---|---|---|
| | Base | Novel | HM | Base | Novel | HM | Base | Novel | HM | Base | Novel | HM | Base | Novel | HM |
| $\sigma$=0.01 | 95.85 | 96.98 | 96.41 | 97.65 | 67.06 | 79.51 | 85.24 | 76.56 | 80.67 | 97.33 | 93.67 | 95.46 | 95.02 | 81.34 | 87.65 |
| $\sigma$=0.05 | 96.07 | 97.14 | 96.60 | 97.84 | 63.73 | 77.18 | 84.38 | 77.50 | 80.79 | 97.86 | 93.52 | 95.64 | 94.88 | 85.06 | 89.70 |
| $\sigma$=0.1 | 95.98 | 97.00 | 96.49 | 97.19 | 67.32 | 79.54 | 85.09 | 77.18 | 80.94 | 97.42 | 94.08 | 95.72 | 94.88 | 84.61 | 89.45 |
| $\sigma$=0.2 | 96.13 | 97.09 | 96.61 | 97.84 | 63.92 | 77.32 | 84.92 | 76.04 | 80.24 | 97.33 | 94.44 | 95.86 | 94.88 | 84.01 | 89.11 |

| Few_Non-IID | OxfordPet | | | Flower102 | | | CIFAR100 | | | Caltech101 | | | Caltech256 | | |
|---|---|---|---|---|---|---|---|---|---|---|---|---|---|---|---|
| | Base | Novel | HM | Base | Novel | HM | Base | Novel | HM | Base | Novel | HM | Base | Novel | HM |
| $\sigma$=0.01 | 95.34 | 95.15 | 95.24 | 95.40 | 71.73 | 81.89 | 82.36 | 78.34 | 80.30 | 95.72 | 94.75 | 95.23 | 93.98 | 85.42 | 89.50 |
| $\sigma$=0.05 | 95.01 | 96.66 | 95.83 | 94.40 | 70.58 | 80.77 | 81.96 | 76.36 | 79.06 | 93.23 | 93.98 | 93.60 | 93.95 | 85.10 | 89.31 |
| $\sigma$=0.1 | 95.28 | 96.39 | 95.83 | 94.33 | 70.64 | 80.78 | 82.21 | 77.90 | 80.00 | 95.69 | 94.44 | 95.06 | 94.39 | 85.80 | 89.89 |
| $\sigma$=0.2 | 95.41 | 95.70 | 95.55 | 94.40 | 66.35 | 77.93 | 82.12 | 77.26 | 79.62 | 92.69 | 94.91 | 93.79 | 93.88 | 84.08 | 88.71 |

| Full_IID | OxfordPet | | | Flower102 | | | CIFAR100 | | | Caltech101 | | | Caltech256 | | |
|---|---|---|---|---|---|---|---|---|---|---|---|---|---|---|---|
| | Base | Novel | HM | Base | Novel | HM | Base | Novel | HM | Base | Novel | HM | Base | Novel | HM |
| $\sigma$=0.01 | 96.18 | 96.56 | 96.37 | 96.67 | 68.63 | 80.27 | 87.52 | 72.26 | 79.16 | 98.13 | 93.52 | 95.77 | 94.63 | 83.68 | 88.82 |
| $\sigma$=0.05 | 95.85 | 95.92 | 95.88 | 97.25 | 62.94 | 76.42 | 87.58 | 74.20 | 80.34 | 97.95 | 92.75 | 95.28 | 95.08 | 84.34 | 89.39 |
| $\sigma$=0.1 | 96.24 | 97.30 | 96.77 | 97.45 | 67.45 | 79.72 | 87.12 | 74.80 | 80.49 | 98.13 | 93.21 | 95.61 | 94.98 | 83.53 | 88.89 |
| $\sigma$=0.2 | 96.35 | 95.92 | 96.13 | 96.67 | 65.88 | 78.36 | 87.36 | 74.84 | 80.62 | 98.40 | 93.67 | 95.98 | 95.12 | 83.20 | 88.76 |

| Full_Non-IID | OxfordPet | | | Flower102 | | | CIFAR100 | | | Caltech101 | | | Caltech256 | | |
|---|---|---|---|---|---|---|---|---|---|---|---|---|---|---|---|
| | Base | Novel | HM | Base | Novel | HM | Base | Novel | HM | Base | Novel | HM | Base | Novel | HM |
| $\sigma$=0.01 | 94.80 | 95.24 | 95.02 | 96.00 | 63.46 | 76.41 | 81.62 | 72.16 | 76.60 | 97.95 | 91.36 | 94.54 | 94.88 | 84.98 | 89.66 |
| $\sigma$=0.05 | 95.14 | 96.34 | 95.74 | 96.20 | 62.50 | 75.77 | 81.96 | 72.16 | 76.75 | 97.77 | 91.98 | 94.79 | 94.63 | 84.31 | 89.17 |
| $\sigma$=0.1 | 95.01 | 96.57 | 95.78 | 95.80 | 67.88 | 79.46 | 82.60 | 72.62 | 77.29 | 97.42 | 92.13 | 94.70 | 94.67 | 85.59 | 89.90 |
| $\sigma$=0.2 | 95.34 | 96.94 | 96.13 | 95.40 | 60.00 | 73.67 | 82.52 | 72.80 | 77.36 | 97.86 | 91.82 | 94.74 | 94.59 | 84.08 | 89.03 |

*Table 34.* The per-dataset test accuracy (%) of different coefficients $\beta$ in RL loss under four data settings. "Few" and "Full" represent few-shot and full-data, while "IID" and "Non-IID" represent IID and Non-IID data settings.

| Few_IID | OxfordPet | | | Flower102 | | | CIFAR100 | | | Caltech101 | | | Caltech256 | | |
|---|---|---|---|---|---|---|---|---|---|---|---|---|---|---|---|
| | Base | Novel | HM | Base | Novel | HM | Base | Novel | HM | Base | Novel | HM | Base | Novel | HM |
| $\beta$=0.1 | 95.79 | 95.39 | 95.59 | 97.84 | 67.84 | 80.12 | 85.28 | 76.28 | 80.53 | 97.86 | 93.21 | 95.48 | 94.98 | 84.19 | 89.26 |
| $\beta$=0.3 | 95.74 | 96.34 | 96.04 | 96.86 | 64.12 | 77.16 | 85.30 | 76.80 | 80.83 | 97.15 | 93.83 | 95.46 | 95.08 | 84.88 | 89.69 |
| $\beta$=0.5 | 95.98 | 97.00 | 96.49 | 97.19 | 67.32 | 79.54 | 85.09 | 77.18 | 80.94 | 97.42 | 94.08 | 95.72 | 94.88 | 84.61 | 89.45 |
| $\beta$=0.7 | 95.79 | 97.14 | 96.46 | 97.45 | 68.43 | 80.40 | 85.22 | 77.44 | 81.14 | 97.15 | 93.98 | 95.54 | 95.19 | 84.85 | 89.72 |
| $\beta$=0.9 | 96.18 | 96.71 | 96.44 | 96.08 | 67.06 | 78.99 | 85.66 | 76.74 | 80.96 | 97.50 | 93.83 | 95.63 | 94.91 | 83.08 | 88.60 |

| Few_Non-IID | OxfordPet | | | Flower102 | | | CIFAR100 | | | Caltech101 | | | Caltech256 | | |
|---|---|---|---|---|---|---|---|---|---|---|---|---|---|---|---|
| | Base | Novel | HM | Base | Novel | HM | Base | Novel | HM | Base | Novel | HM | Base | Novel | HM |
| $\beta$=0.1 | 95.55 | 96.71 | 96.13 | 94.60 | 70.38 | 80.71 | 81.80 | 77.82 | 79.76 | 91.53 | 93.83 | 92.67 | 93.88 | 84.98 | 89.21 |
| $\beta$=0.3 | 94.67 | 96.30 | 95.48 | 94.60 | 67.88 | 79.04 | 82.46 | 77.42 | 79.86 | 94.74 | 94.60 | 94.67 | 93.95 | 86.00 | 89.80 |
| $\beta$=0.5 | 95.28 | 96.39 | 95.83 | 94.33 | 70.64 | 80.78 | 82.21 | 77.90 | 80.00 | 95.69 | 94.44 | 95.06 | 94.39 | 85.80 | 89.89 |
| $\beta$=0.7 | 94.26 | 95.38 | 94.82 | 94.40 | 69.23 | 79.88 | 82.40 | 77.84 | 80.06 | 95.19 | 94.60 | 94.89 | 93.88 | 85.54 | 89.52 |
| $\beta$=0.9 | 94.67 | 96.62 | 95.64 | 95.60 | 67.50 | 79.13 | 82.30 | 78.16 | 80.18 | 93.58 | 94.75 | 94.16 | 94.27 | 85.30 | 89.56 |

| Full_IID | OxfordPet | | | Flower102 | | | CIFAR100 | | | Caltech101 | | | Caltech256 | | |
|---|---|---|---|---|---|---|---|---|---|---|---|---|---|---|---|
| | Base | Novel | HM | Base | Novel | HM | Base | Novel | HM | Base | Novel | HM | Base | Novel | HM |
| $\beta$=0.1 | 95.45 | 96.82 | 96.13 | 95.88 | 63.92 | 76.70 | 87.52 | 75.20 | 80.89 | 98.04 | 94.14 | 96.05 | 94.91 | 83.20 | 88.67 |
| $\beta$=0.3 | 95.68 | 96.40 | 96.04 | 95.88 | 64.71 | 77.27 | 87.86 | 73.90 | 80.28 | 98.31 | 94.29 | 96.26 | 94.91 | 84.07 | 89.16 |
| $\beta$=0.5 | 96.24 | 97.30 | 96.77 | 97.45 | 67.45 | 79.72 | 87.12 | 74.80 | 80.49 | 98.13 | 93.21 | 95.61 | 94.98 | 83.53 | 88.89 |
| $\beta$=0.7 | 96.02 | 95.55 | 95.78 | 97.25 | 63.14 | 76.57 | 87.78 | 74.18 | 80.41 | 98.13 | 93.98 | 96.01 | 94.88 | 84.61 | 89.45 |
| $\beta$=0.9 | 96.07 | 95.92 | 95.99 | 96.67 | 64.12 | 77.10 | 86.98 | 74.82 | 80.44 | 98.04 | 93.52 | 95.73 | 94.81 | 81.70 | 87.77 |

| Full_Non-IID | OxfordPet | | | Flower102 | | | CIFAR100 | | | Caltech101 | | | Caltech256 | | |
|---|---|---|---|---|---|---|---|---|---|---|---|---|---|---|---|
| | Base | Novel | HM | Base | Novel | HM | Base | Novel | HM | Base | Novel | HM | Base | Novel | HM |
| $\beta$=0.1 | 95.07 | 96.43 | 95.75 | 95.80 | 65.58 | 77.86 | 82.86 | 72.36 | 77.25 | 98.04 | 91.98 | 94.91 | 94.41 | 83.76 | 88.77 |
| $\beta$=0.3 | 94.33 | 96.16 | 95.24 | 96.00 | 66.15 | 78.33 | 80.98 | 71.66 | 76.04 | 97.50 | 91.98 | 94.66 | 94.74 | 85.83 | 90.07 |
| $\beta$=0.5 | 95.01 | 96.57 | 95.78 | 95.80 | 67.88 | 79.46 | 82.60 | 72.62 | 77.29 | 97.42 | 92.13 | 94.70 | 94.67 | 85.59 | 89.90 |
| $\beta$=0.7 | 95.14 | 96.52 | 95.83 | 96.40 | 65.00 | 77.65 | 80.76 | 71.30 | 75.74 | 97.95 | 91.67 | 94.71 | 94.52 | 83.64 | 88.75 |
| $\beta$=0.9 | 95.01 | 96.66 | 95.83 | 96.00 | 65.38 | 77.79 | 82.94 | 73.24 | 77.79 | 96.17 | 92.13 | 94.11 | 94.70 | 84.28 | 89.19 |

