# OpenReview forum: "Decoupled Training with Local Reinforcement Fine-Tuning in Federated Learning"
_ICML.cc/2026/Conference — ICML 2026 regular_

### Official Review · Reviewer_XUs8 · 2026-03-09

**Soundness:** 4
**Presentation:** 3
**Significance:** 3
**Originality:** 3
**Overall Recommendation:** 5
**Confidence:** 4

**Summary:**

Addressing the challenges of inter-client optimization inconsistency and intra-client over-specialization in Federated Vision-Language Models under Non-IID data, this paper proposes FedDTL. The framework innovatively decouples the training of image and text encoders: the image encoder is trained locally to preserve privacy, while the text encoder learns global semantics on the server, reducing client drift through modality alignment. Furthermore, a two-stage fine-tuning strategy is introduced, utilizing supervised learning for a rapid warm-start followed by reinforcement learning to suppress overfitting and enhance generalization. Experiments demonstrate that FedDTL effectively balances task adaptation and generalization in both few-shot and full-data scenarios.

**Compliance With Llm Reviewing Policy:**

Affirmed.

**Final Justification:**

I appreciate the authors' detailed responses and finalize to accept.

**Key Questions For Authors:**

NO

**Limitations:**

Yes

**Strengths And Weaknesses:**

Strengths

1.	Mitigates Inter-Client Inconsistency: By decoupling encoder training and centralizing text semantics, the framework avoids the "coordinate system" drift caused by traditional parameter averaging, ensuring consistency in global semantic updates.

2.	Suppresses Intra-Client Over-specialization: The unique "Supervised Fine-Tuning + Reinforcement Learning" two-stage strategy guarantees convergence speed while effectively preventing excessive memorization of local data, significantly improving generalization performance.

3.	Robust Across Data Regimes: Under various heterogeneous settings, including label skew and feature shift, the method achieves an optimal balance between task adaptation and generalization in both few-shot and full-data scenarios.

.
Weakness

1.	The motivation of the alignment between the server text embedding and the client image embedding is unclear. Intuitively, aligning the same modality will achieve the better performance.

2.	It is recommended to add the preliminaries about the federated VLA.

3.	Many previous works [1,2,3] have been proposed to address the NonIID in Federated learning for general models. The paper should discuss whether these methods can solve the NonIID problem in the Federated VLA.

[1] FedCDA: Federated Learning with Cross-rounds Divergence-aware Aggregation. In The Twelfth International Conference on Learning Representations (ICLR 2024), Vienna, Austria, May 7th to May 11th, 2024.

[2] DaFKD: Domain-aware Federated Knowledge Distillation. The Thirty-Fourth IEEE/CVF Conference on Computer Vision and Pattern Recognition (CVPR 2023), Vancouver, Canada, June 18-22, 2023.

[3] FedNLR: Federated Learning with Neuron-wise Learning Rates. The 30th SIGKDD Conference on Knowledge Discovery and Data Mining (KDD 2024), Barcelona, Spain, 25-29 August 2024.

---

> ### Author Rebuttal · Authors · 2026-03-31
>
> We thank Reviewer XUs8 for the insightful comments. The reviewer highlights that our method "**effectively mitigates the inter-client inconsistency by decoupling training**" and "**suppresses the intra-client over-specialization by SFT-RL fine-tuning**", achieving **an optimal balance** between task adaptation and generalization in **few-shot and full-data settings under various heterogeneous experimental settings**. We address the concerns below.
>
> > **W1: Unclear motivation of alignment between text and image embeddings.**
>
> Thanks. We clarify a potential misunderstanding in alignment motivation. ***Aligning server text embeddings with client image embeddings is to address inter-client optimization inconsistency under heterogeneous FL***, which is a core challenge in our work. Specifically, the server updates a global text encoder to produce semantic anchor embeddings, defining a unified embedding space shared across heterogeneous clients. Compared to conventional simple parameter aggregation, this centralized semantic guidance enables local image encoders to align their representations toward the unified embedding space, mitigating optimization inconsistency and improving global task adaptation under heterogeneous FL scenarios. ***Our framework not only performs cross-modal alignment (the core motivation) but also performs intra-modal alignment***. Intra-modal alignment is achieved in two ways: 1) vision alignment through shared LoRA parameter aggregation across clients; 2) text alignment through centralized server-side optimization. **Prior federated VLM works mainly focus on single-modality alignment or multimodal alignment via local contrastive learning**. In contrast, our method jointly performs cross-modal and intra-modal alignment between clients and the server, improving performance under heterogeneous data.
>
> > **W2: Suggestion of adding preliminaries about federated VLA (VLM?).**
>
> Thanks for your suggestion. We have included the implementation details of VLM training under our proposed decoupled framework (cf. `L142–159 in Sec. 3.2`). Following the suggestion, we will further add a more explicit and structured preliminary of federated VLMs (e.g., text and image encoding in CLIP and their interaction in FL) at the beginning of the "Method" section in the revision. **We wonder if this might be a minor typo error between VLM and VLA? If not, we would greatly appreciate the reviewer’s corrections and welcome further in-depth discussion**.
>
>
> > **W3: Discuss previous works in the Non-IID setting.**
>
> Thanks for your suggestion. These prior works address Non-IID challenges in FL from different perspectives: 1) **FedCDA** investigates ***cross-round local model aggregation*** by selecting a suitable local model across rounds that minimizes its divergence from local models of other clients, ensuring that the aggregated global model remains close to all selected local models. 2) **DaFKD** endows ***the model with different importance*** by computing the correlation between the distillation sample and the training domain with a per-client domain discriminator, optimizing the federated distillation across heterogeneous clients. 3) **FedNLR** proposes ***neuron-wise learning rates*** during local training, enhancing neurons bound learning of local classes while reducing the decay of non-local classes knowledge by setting a smaller learning rate.
>
> Although these methods are designed for general models and cannot be directly applied to federated VLMs, ***their core ideas provides valuable insights for handling Non-IID challenges in federated VLMs***: 1) The cross-round selection in FedCDA could inspire ***adaptive selection of per-client local parameters (image encoders) from local parameter pools (store local parameters of the recent rounds)***, improving parameter consistency on the global aggregation at the intra-client level. 2) The correlation-aware weighting in DaFKD can be extended to ***estimate inter-client similarity and assign adaptive aggregation coefficients***, facilitating better aggregation at the inter-client level. 3) The neuron-level optimization in FedNLR could inspire ***fine-grained neuron adaptation in VLMs***, such as assigning different importance to encoder components for local/non-local classes, improving intra-model level optimization. We will include this discussion in the revision, and this also provides an interesting direction for further research in federated VLMs from different levels.

---

> > ### Author Rebuttal · Reviewer_XUs8 · 2026-04-02
> >
> > Thank you for the rebuttal . You have addressed my main questions.  I keep my original rating.

---

> > > ### Author Response · Authors · 2026-04-07
> > >
> > > We sincerely thank the reviewer for the positive recommendation and for acknowledging that our responses have addressed the main questions. Following your suggestion, we will further add a more explicit and structured preliminary of federated VLMs and include the discussion of three related works in the revision. We are grateful for your constructive discussion and engagement.

---

### Official Review · Reviewer_bLBC · 2026-03-12

**Soundness:** 3
**Presentation:** 3
**Significance:** 3
**Originality:** 3
**Overall Recommendation:** 5
**Confidence:** 3

**Summary:**

This study addresses the challenge of balancing task adaptability and generalization in federated learning for visual-language models (VLMs) under data heterogeneity and full-data scenarios by proposing the FedDTL framework. Its core innovations comprise two components: decoupled encoder training and two-stage local fine-tuning. FedDTL demonstrates outstanding performance across various federated learning data distributions, both in few-shot and full-data settings.

**Compliance With Llm Reviewing Policy:**

Affirmed.

**Final Justification:**

Most of my concerns have been adequately addressed as stated in the acknowledgement part. Thus, I increase my rating.

**Key Questions For Authors:**

1.Since $\mathcal{T}_g$ is updated globally on the server using features from the previous round of client uploads, the reward function used by the client in round $t$ is effectively derived from a "stale" version of the global semantic space. How does this asynchronous update affect the stability of the policy gradient? Specifically, if the server-side update shifts the semantic clusters significantly, could the local GRPO process be guided by inaccurate or lagging advantage estimates?

2.1.Can the author provide specific comparative data from the experiment regarding client computational power consumption (FLOPs) and memory footprint to demonstrate the feasibility of this process on edge devices?

3.When updating the global text encoder, the author requires clients to upload the feature $\{\overline{z}_{v,k}\}$ of a single image sample. This instance-level semantic representation directly exposes the distributional characteristics of users' private data, revealing more information about the original sample than parameter updates alone. Even when employing random sampling of partial embeddings to reduce communication overhead, as mentioned in the paper, this merely diminishes the total volume of leaked data without altering the inherent risk of individual leaks. Once the server acquires a sufficient number of embeddings, it can perform membership inference or attribute inference attacks.Can you provide the difficulty of reconstructing images from the uploaded features?

**Limitations:**

As mentioned above, uploading individual embeddings may pose data leakage risks. Feature aggregation is recommended: instead of uploading individual embeddings, upload aggregated values of central prototypes or feature gradients for that category within the client to provide stronger statistical masking protection.

**Strengths And Weaknesses:**

### Strengths
1.A clever and novel method design, featuring an innovative two-stage training mechanism that decouples understanding from local SFT-RL, systematically addresses representation drift in non-IID scenarios and enhances model generalization under statistical heterogeneity.

2.Most existing studies focus on few-shot scenarios, whereas this paper demonstrates that FedDTL exhibits greater stability in full-data and long-trajectory local training, effectively mitigating the significant performance fluctuations commonly observed in baseline methods.

### Weaknesses
1.In federated learning scenarios, clients are typically edge devices with limited computational resources. The requirement to perform supervised fine-tuning locally before proceeding to reinforcement learning significantly extends the local training time per round, resulting in overall convergence inefficiency.

2.The paper claims that since $\mathcal{T}_g$ only operates on label descriptions and highly compressed image features, global training can prevent privacy leakage. However, it is now common for generative models (such as GANs or diffusion models) to reconstruct original images from feature vectors. This may pose a risk of information leakage.

3.The paper introduces a two-stage training approach using SFT-RL, which imposes certain convergence requirements to enhance model performance. While traditional FedAvg exhibits well-defined convergence boundaries, the introduction of decoupled training and RL objectives makes it mathematically more challenging to guarantee convergence of the global model within the non-convex optimization space.

---

> ### Author Rebuttal · Authors · 2026-03-31
>
> We thank Reviewer bLBC for the valuable comments. The reviewer notes that our "**method is clever and novel, effectively mitigating performance fluctuations in few-shot and full-data settings (overlooked in prior works) under data heterogeneity**". We address the concerns below.
>
> > **W1: Local training efficiency under SFT with RL.**
>
> Thanks. We clarify a potential misunderstanding in our local SFT-RL fine-tuning. ***Our two-stage training does not interleave executing SFT and RL within each round***. In early global rounds, users perform **SFT-only local optimization** for fast and stable adaptation. After adaptation saturates, users switch to **RL-only local generalization enhancement**, without further SFT. Thus, our two-stage design does not increase per-round local computation but accelerates model convergence via SFT warm-up and avoids long-term RL training, making it suitable for limited edge devices.
>
> > **W2/Q3: (I) Privacy concern about uploading individual embeddings; (II) Suggestion of embedding aggregation.**
>
> Good point. **(I)** ***We only upload the visual class token embedding (that means $\bar{z}_v$ in the paper is the visual class token embedding instead of full feature embeddings), while keeping patch embeddings local***. The server only needs the visual class token embedding (only contains high-level classification knowledge) to optimize the text encoder, while the patch token embeddings (contain fine-grained private features) remain local, mitigating image reconstruction attacks. The full analysis including potential attack analysis and noise-based defense can refer to `W1(II) in Reviewer NxRm`.
>
> **(II)** We further explore the suggested feature aggregation and report the average results below. We also include the noise perturbation. **Both strategies have competitive performance while enhancing stronger privacy protection, especially with noise perturbation**.
>
> |Base/Novel|Few-NonIID|Few-Dir(0.1)|Full-NonIID|Full-Dir(0.1)|
> |-|-|-|-|-|
> |PromptFL|82.23/79.84|80.41/76.09|62.98/48.48|75.82/57.98|
> |FedMaPLe|83.63/77.56|84.05/77.69|80.56/69.41|89.27/70.10|
> |FedDTL|89.58/83.01|90.95/82.64|91.64/77.72|92.40/76.59|
> |FedDTL+agg|88.64/83.20|88.71/81.21|81.92/72.50|90.91/77.79|
> |FedDTL+noise|88.77/82.23|89.69/82.17|91.44/79.04|91.80/77.86|
>
> In summary, **FedDTL achieves a trade-off between performance and privacy. Incorporating a privacy protection scheme is a promising future direction**.
>
> > **W3: Convergence guarantee.**
>
> Good point. We clarify that the convergence requirement in our paper ***does not aim to find an optimal SFT convergence point, but serves as a simple signal indicating that local adaptation has saturated and provide a stable initialization for the subsequent RL stage*** (cf. `Fig.5`). From a theoretical perspective inspired from split FL, our decoupled framework can be viewed as a bi-level optimization problem with client-side (image encoder, IE) and server-side (text encoder, TE) separately. IE* and TE* denote the optimal solution. Due to decoupling, there exists a coupling error $\epsilon$.
>
> * Global objective: $min_{TE}\mathbb{E}[\mathcal{L}(TE, IE^*)]$
> * Local objective: $min_{IE_k}\mathbb{E}[\mathcal{L}(IE_k, TE^*)]$
>
> For **client**, the SFT stage reduces to standard FL, yielding a convergence rate of approximately $O(1/T)+O(\epsilon)$ [R1], where T is SGD number. The convergence analysis on general GRPO-variants optimization within the non-convex space is still a challenging topic, we will follow related works in the future. For **server**, under uploading embeddings, the convergence rate similarly follows $O(1/T)+O(\epsilon)$ [R1]. Our FedDTL is an empirical method and can serve as the basis to encourage more rigorous theoretical analysis in the future.
>
> [R1] On the Convergence of FedAvg on Non-IID Data, ICLR 2020.
>
> > **Q1: Whether asynchronous client-server updates cause unstable local training.**
>
> Thanks. We agree that **RL-only optimization may introduce unstable training** due to asynchronous client-server updates (`RL-only in Fig.2`). We alleviate it via: 1) **SFT warm start**: early SFT provides a robust initialization, preventing unstable RL updates. 2) **KL regularization**: RL update is constrained to the reference model (including latest global RL model), stabilizing policy updates. 3) **Global TE training**: The text encoder is updated centrally with uploading embeddings, ensuring that semantic clusters do not shift significantly. They jointly improve training stability (`SFT-RL in Fig.2`).
>
> > **Q2: Ask for FLOPs and memory footprint at client.**
>
> Thanks for the suggestion. The quantitative analysis is shown below. Compared to baselines, our FedDTL has lower FLOPs and tolerable memory footprint. Thus, FedDTL is suitable for limited edge devices.
>
> | |FLOPs|Memory|
> |-|-|-|
> |pFedDC|46.46G|6.92G|
> |FedPGP|286.44G|6.04G|
> |pFedMMA|46.15G|3.65G|
> |PromptFL|110.1G|2.02G|
> |FedMaPLe|110.22G|6.14G|
> |FedLoRA|109.98G|6.47G|
> |FedDTL|22.30G(SFT)/33.45G(RL)|5.65G(SFT)/5.98G(RL)|

---

> > ### Author Rebuttal · Reviewer_bLBC · 2026-04-03
> >
> > I would like to thank the author for the detailed responses regarding computational efficiency, privacy risks, and convergence stability. The author’s replies have essentially addressed all of my concerns. Given that the author has provided comprehensive quantitative analysis (FLOPs, memory usage, and privacy-enhancing experiments) and refined the theoretical explanations, I believe this work demonstrates significant practical value and originality in the field of federated vision-language models, and I have decided to raise my rating to 5.

---

> > > ### Author Response · Authors · 2026-04-07
> > >
> > > Thanks for your positive recommendation and for raising the score to Accept. We also appreciate constructive suggestions and comments on computational efficiency, privacy risks, and convergence stability. Following your suggestion, we will include the additional analysis and privacy enhancement (embedding aggregation / adding noise to embeddings) experiments in our appendix for a clearer and strengthened revision. We are grateful for your engagement in the rebuttal process.

---

### Official Review · Reviewer_DCyn · 2026-03-12

**Soundness:** 2
**Presentation:** 2
**Significance:** 3
**Originality:** 3
**Overall Recommendation:** 3
**Confidence:** 3

**Summary:**

This paper studies federated adaptation of pre-trained vision-language models under heterogeneous client data. It proposes FedDTL, which decouples image and text encoder training across clients and servers, and combines this with a two-stage local fine-tuning strategy consisting of supervised fine-tuning followed by an RL-based stage. The goal is to reduce inter-client inconsistency and intra-client over-specialization while balancing task adaptation and generalization. Experiments across label-skew and feature-shift settings show promising results. Overall, the authors focus on the concept of balancing adaptation and generalization in federated VLMs. The manuscript’s general domain is federated learning for vision-language model adaptation.

**Compliance With Llm Reviewing Policy:**

Affirmed.

**Key Questions For Authors:**

•	What aspects of the method specifically require reinforcement learning, rather than standard stochastic or regularized optimization?
•	What is the privacy threat model when uploading image embeddings, and how does the method protect against potential privacy leakage?
•	Were baseline methods carefully tuned for the experimental settings used in the paper, especially in full-data experiments?
•	Can the authors report variance or confidence intervals for the main experimental results?
•	How sensitive is the method to key hyperparameters, such as noise magnitude, embedding upload ratio, and RL sampling strategy?
•	What is the additional communication cost per round compared with standard federated learning approaches?

**Limitations:**

•	The reinforcement-learning-based stage lacks fully convincing conceptual justification relative to simpler stochastic optimization alternatives.
•	The claimed privacy advantages are not supported by theoretical or empirical analysis.
•	The evaluation focuses mainly on aggregate accuracy, with limited analysis of failure cases or representation behavior.
•	The computational and communication costs of the proposed framework are not fully discussed.

**Strengths And Weaknesses:**

Strengths

•	The paper addresses an important problem: federated adaptation of vision–language models under heterogeneous client distributions, which is increasingly relevant for decentralized deployment of multimodal models.
•	The motivation is clear and well explained, particularly the discussion of optimization inconsistency across clients and local model over-specialization in federated training.
•	The proposed decoupled training strategy that separates image encoder updates on clients and text encoder optimization on the server is conceptually intuitive and aligns with the structure of CLIP-like models.
•	Using a shared text encoder as a global semantic anchor is a reasonable design choice that may help maintain cross-client semantic consistency.
•	The experimental evaluation is relatively broad, covering multiple datasets and both label-skew and feature-shift scenarios, as well as few-shot and full-data settings.

Weaknesses
•	The reinforcement learning stage is interesting, but its conceptual advantage over stochastic or regularized supervised optimization is not fully established. While the paper defines a GRPO-inspired objective, it remains somewhat unclear whether the gains arise specifically from RL or from the added stochastic refinement mechanism.
•	The manuscript claims privacy benefits from uploading embeddings, but no formal privacy analysis or empirical evaluation is provided.
•	The technical novelty is moderate, as the framework mainly combines several existing ideas rather than introducing a fundamentally new training paradigm.
•	The paper reports average accuracy improvements, but limited discussion is provided regarding statistical variance or robustness of the results.
•	The communication and computational overhead introduced by server-side optimization and RL refinement is not analyzed quantitatively.

---

> ### Author Rebuttal · Authors · 2026-03-31
>
> We thank Reviewer DCyn for the valuable comments. Reviewer notes that our FedDTL "​**addresses an important problem in federated VLMs under data heterogeneity**​" with a "**clear and well-explained motivation**", and our decoupling strategy is **a conceptually intuitive innovation** with a "​**broad experiment evaluation**​". We address the concerns below.
>
> > **W1/Q1: Are the gains from RL or from the added stochastic/regularized mechanism?**
>
> Thanks. We would like to clarify the potential misunderstanding regarding our RL stage: ***the stochastic/regularized mechanisms are not added components beyond RL, but part of the components in RL optimization itself***. Our proposed GRPO-inspired RL consists of three components (from "Stage 2: RL-based Generalization Enhancement" to Eq.8 in `Sec. 3.3`): stochastic mechanism for action sampling, reward estimation, and policy optimization via reward-driven policy gradient and KL regularization. These are essential components in RL area, rather than additional modules. Thus, **the gains from the inherent stochastic exploration and KL-regularized optimization in RL stage, rather than the additional stochastic/regularized mechanism in addition to RL**. If we have misunderstood the reviewer's point, we welcome further discussion during the discussion period.
>
> > **W2/Q2: Privacy concern.**
>
> Thanks. There is a potential confusion about "privacy benefits from uploading embeddings". Our design aims to ***reduce privacy risks compared to raw image data or full feature sharing***. Specifically, only the class token embedding (encodes high-level semantic information) is uploaded, while patch token embeddings (contains richer visual details) remain local, mitigating reconstruction risks. Thus, the fine-grained private attributes remain protected. Then, we analyze potential attacks and introduce noise perturbation to further protect privacy (cf. `W1(II) in Reviewer NxRm`). In summary, FedDTL balances global task adaptation and generalization in heterogeneous federated VLMs and achieves a trade-off between performance and privacy. Incorporating a privacy-preserving scheme is a promising future direction.
>
> > **W3: Novelty concern.**
>
> Thanks. We would like to clarify that ***we address an unexplored problem in federated VLMs: balancing global task adaptation and generalization under heterogeneous data in few-shot and full-data FL settings***. Our work is not a simple combination of existing ideas, but a jointly well-designed method with three innovative perspectives:
>
> - As far as we know, we are the first to propose **modality-decoupled training** in federated VLMs to address inter-client optimization inconsistency.
> - We propose **a novel GRPO-inspired RL in federated VLMs** with SFT warm-up to mitigate intra-client over-specialization.
> - We provide a unified evaluation across **few-shot and full-data settings**, which is overlooked in prior works (focus on local strategy optimization and lightweight parameter aggregation).
>
> We believe these aspects establish clear novelty. We welcome reviewer to provide specific references and further discuss the novelty concern if needed.
>
> > **W4/Q4+W5/Q6+Q5: Ask for (I) statistical variance, (II) communication and computation overhead, and (III) sensitivity to hyperparameters.**
>
> Thanks for your suggestion. **(I)** We follow standard practice in baselines and report **average accuracy over three trials** (cf. `L291-293 right`). Due to space limitations, full results are in `Appendix C.1` and variance results (Tab.1-4) at the anonymous link (https://anonymous.4open.science/r/FedDTL-6977/ICML_rebuttal.pdf).
>
> **(II)** The quantitative results on CIFAR100 are below. Compared to baselines and standard federated VLMs (FedLoRA), our FedDTL is **computationally efficient**. While embedding uploading increases communication cost, it can be **significantly reduced by uploading a few embeddings** (cf. `Fig.3`). If we upload 16 embeddings per class, the base and novel accuracy in the full-data setting are 93.85/80.02(IID), 91.83/79.21(Dir(0.1)), and 90.06/77.71(Non-IID). This enables a trade-off between performance and communication.
>
> | Per-round|Comm|Trainable Param|FLOPs|
> |-|-|-|-|
> |pFedDC|49.25K|1.07M|46.46G|
> |FedPGP|16K|0.02M|286.44G|
> |pFedMMA|8.25K|0.64M|46.15G|
> |PromptFL|8K|0.008M|110.1G|
> |FedMaPLe|3.39M|3.39M|110.22G|
> |FedLoRA|0.39M|0.39M|109.98G|
> |FedDTL(ours)|0.96M|0.18M|22.30G(SFT)/33.45G(RL)|
>
> **(III)** We have already conducted hyperparameter ablations in our paper:
>
> - Noise magnitude (cf.`Fig.9`),
> - Embedding upload ratio (cf.`Fig.3`),
> - RL sampling strategy (cf.`Tab.7, Fig.8`).
>
> Results show that **FedDTL is robust across these hyperparameters** (see `Appendix D.5`).
>
> > **Q3: Were experimental settings in baselines tuned?**
>
> Thanks. We follow the original and recommended implementation and setting of all baselines. To ensure fairness, we test the final converged performance and maintain consistent settings across all baselines and our FedDTL.

---

> > ### Author Rebuttal · Reviewer_DCyn · 2026-04-05
> >
> > More broadly, the remaining concerns I raised have not been adequately addressed in the rebuttal. As such, I do not find the responses sufficient to change my evaluation, and I will maintain my current score.

---

> > > ### Author Response · Authors · 2026-04-07
> > >
> > > Thanks for your time in carefully reviewing our paper and rebuttal. We noticed that you selected (c) Partially resolved or unresolved. To better clarify our response, we have carefully revisited all the concerns, including 5 weaknesses and 6 key questions. We summarized them into related points and provided detailed responses in the rebuttal. Below, we briefly restate our clarifications for addressing the reviewer’s concerns:
> > >
> > > 1) **Weakness 1 & Question 1**: These points focus on whether the gains are from RL or from the added stochastic/regularized mechanism. We provided detailed responses in `W1/Q1`. In summary, we aim to clarify that the stochastic/regularized mechanisms are not additionally independent components beyond RL, but are the intrinsic core components in RL optimization itself. Therefore, ***the gains from the inherent stochastic exploration and KL-regularized optimization in RL stage, rather than the additional stochastic/regularized mechanism in addition to RL***.
> > > 2) **Weakness 2 & Question 2**: These points focus on the privacy of uploading embeddings. We have provided a response in `W2/Q2`. In this response, we claimed that detailed analysis and experiment results are included in `W1(II) of Reviewer NxRm` due to space limitations.
> > > The detailed response: ***We only upload the visual class token embedding (that means $\bar{z}_v$ in the paper is the class token embedding instead of full feature embeddings), while keeping patch token embeddings local***. The server only needs the class token embedding (only contains high-level classification knowledge) to update the text encoder. The patch token embeddings that contain fine-grained private features are not shared, mitigating reconstruction attacks. Under the **honest-but-curious server attack assumption**, the server only accesses normalized class token embeddings and partial LoRA parameters, significantly limiting attack feasibility: 1) gradient inversion attacks (e.g., GradInversion[R1]) require full gradients; 2) generative reconstruction attacks (GAN/Diffusion[R2]) require rich conditioning signals (e.g., full embeddings, intermediate features), which are not available in our method. We **further enhance privacy via noise perturbation**.  As shown below (noise $\sigma=0.1$), the average performance of FedDTL with noise is even on par with FedDTL and exceeds the optimal baseline, demonstrating the privacy–utility trade-off. Further privacy protection on FedDTL is a promising future direction (e.g., DP noise design on theoretical perspective $\sigma=\Delta f\sqrt{2ln(1.25/\delta)}/\epsilon$).
> > > |Base/Novel|Few-NonIID|Few-Dir(0.1)|Full-NonIID|Full-Dir(0.1)|
> > > |-|-|-|-|-|
> > > |FedMaPLe|83.63/77.56|84.05/77.69|80.56/69.41|89.27/70.10|
> > > |FedDTL|89.58/83.01|90.95/82.64|91.64/77.72|92.40/76.59|
> > > |FedDTL+noise|88.77/82.23|89.69/82.17|91.44/79.04|91.80/77.86|
> > >
> > > 3) **Weakness 3**: This point focuses on novelty. We gave a detailed analysis in `W3`, and we would like to emphasize again that ***our work addresses a new problem in federated VLMs***: balancing global task adaptation and generalization across various heterogeneous data settings under both few-shot and full-data regimes. Then, we ***propose a novel decoupled framework and a GRPO-inspired RL scheme*** to address this challenge instead of simply combining existing techniques.
> > > 4) **Weakness 4 & Question 4**: These points ask for statistical variance. Due to the space constraint, we provided the statistical variance via the anonymous link (refer to `W4/Q4 + W5/Q6 + Q5(I)`). We would like to emphasize again that we follow standard practice in baselines and report average accuracy over three trials (cf. `L291-293 right`).
> > > 5) **Weakness 5 & Question 6**: These points ask for the analysis of the communication and computational cost. We provided detailed comparisons of communication overhead, FLOPs, and trainable parameters across baselines, standard federated VLMs, and our FedDTL in `W4/Q4 + W5/Q6 + Q5(II)`.
> > > 6) **Question 5**: This point asks for the hyperparameter analysis. The mentioned hyperparameters (i.e., noise magnitude, embedding upload ratio, and RL sampling strategy) are already shown in the original main text and appendix, and we have indicated their positions in `W4/Q4 + W5/Q6 + Q5(III)`.
> > > 7) **Question 3**: This point focuses on the baseline setting, which we addressed in `Q3`.
> > >
> > > Thank you again for your time and consideration. We would like to respectfully note that, ***in our rebuttal, we have provided detailed point-by-point responses to all concerns raised in the original review, together with additional clarifications and supporting evidence. We hope that these clarifications will further help the reviewer to solve the remaining concerns.***
> > >
> > >
> > > [R1] See through gradients: Image batch recovery via gradinversion, CVPR 2021.
> > >
> > > [R2] Diffusion-driven GAN Inversion for Multi-Modal Face Image Generation, CVPR 2024.

---

### Official Review · Reviewer_NxRm · 2026-03-13

**Soundness:** 3
**Presentation:** 2
**Significance:** 2
**Originality:** 2
**Overall Recommendation:** 4
**Confidence:** 3

**Summary:**

The manuscript proposes a method for doing Federated SFT and RL on VLMs. The main idea is that the text encoder can be trained on the server while maintaining privacy.

**Compliance With Llm Reviewing Policy:**

Affirmed.

**Final Justification:**

See rebuttal acknowledgement

**Key Questions For Authors:**

Can you isolate the effect of the RL part, is it possible to use this with the other methods mentioned?

**Strengths And Weaknesses:**

Strengths

— the idea to separate the text encoder seems novel

— The results compared to several baselines are promising

Weakness

— The papers contributions are somewhat incremental and the presentation does not give a strong motivation for the relevance of the setting. Importantly it seems that there are many assumptions added compared to the prior work and a standard FL setting that the server can perform significant computation and that the text in VLM training has minimal privacy leakage. But these assumptions are neither well motivated practically nor proven in the case of the privacy claims.

— The comparisons in light of the above seem a bit contrived since the assumptions of the baseline methods are different. Further an SFT/RL pipeline is used but it is not clear if the compared methods benefit from RL. There doesn’t seem to be a proper non-federated baseline.

---

> ### Author Rebuttal · Authors · 2026-03-31
>
> We thank Reviewer NxRm for the valuable comments. The reviewer notes our "**idea of separating the text encoder is novel**​" and "**experimental results are promising**​". We address the concerns below.
>
> > **W1: The reviewer challenges the assumption (I) not well motivated; (II) has minimal privacy leakage.**
>
> Thanks. **(I)** We clarify that leveraging server-side computation is a ***well-established and promising trend*** in heterogeneous FL. Prior works like [R1], [R2], and split FL demonstrate that server-side training is a practical design. However, **these works always struggle to generalize to unseen classes**. In contrast, VLMs (e.g., CLIP) learn a shared image-text embedding space, allowing zero-shot recognition via text prompts and suited for generalization tasks. But recent federated VLMs rely on purely local updates with simple server-side aggregation, making them difficult to handle inter-client optimization inconsistency in heterogeneous FL. Our FedDTL **makes the first attempt to leverage server-side computation via a decoupling scheme in federated VLMs** to alleviate this issue (cf. `Tab.1`).
>
> **(II)** Good point! ***We only upload the visual class token embedding (that means $\bar{z}_v$ in the paper is the class token embedding instead of full feature embeddings), while keeping patch token embeddings local***. The server only needs the class token embedding (only contains high-level classification knowledge) to update the text encoder. The patch token embeddings that contain fine-grained private features are not shared, mitigating reconstruction attacks. Under the **honest-but-curious server attack assumption**, the server only accesses normalized class token embeddings and partial LoRA parameters, significantly limiting attack feasibility: 1) gradient inversion attacks (e.g., GradInversion[R3]) require full gradients; 2) generative reconstruction attacks (GAN/Diffusion[R4]) require rich conditioning signals (e.g., full embeddings, intermediate features), which are not available in our method. We **further enhance privacy via noise perturbation**.  As shown below (noise $\sigma=0.1$), the average performance of FedDTL with noise is even on par with FedDTL and exceeds the optimal baseline, demonstrating the privacy–utility trade-off.
>
> |Base/Novel|Few-NonIID|Few-Dir(0.1)|Full-NonIID|Full-Dir(0.1)|
> |-|-|-|-|-|
> |FedMaPLe|83.63/77.56|84.05/77.69|80.56/69.41|89.27/70.10|
> |FedDTL|89.58/83.01|90.95/82.64|91.64/77.72|92.40/76.59|
> |FedDTL+noise|88.77/82.23|89.69/82.17|91.44/79.04|91.80/77.86|
>
> Further privacy protection on FedDTL is a promising future direction (e.g., DP noise design on theoretical perspective $\sigma=\Delta f\sqrt{2ln(1.25/\delta)}/\epsilon$).
>
> [R1] Data-Free Knowledge Distillation for Heterogeneous Federated Learning, ICML 2021.
>
> [R2] Efficient Model Personalization in Federated Learning via Client-Specific Prompt Generation, ICCV 2023.
>
> [R3] See through gradients: Image batch recovery via gradinversion, CVPR 2021.
>
> [R4] Diffusion-driven GAN Inversion for Multi-Modal Face Image Generation, CVPR 2024.
>
> > **W2: Misinterpretation of RL gains.**
>
> Thanks. We would like to emphasize that RL is not merely an auxiliary technique but rather a core contribution of our work. ***To the best of our knowledge, this is the first attempt to introduce a GRPO-inspired RL framework into federated VLM to improve generalization under data heterogeneity***. `Fig.2` and `Q1(II)` show the effective generalization enhancement with our SFT-RL scheme. Thus, RL is a core contribution, not an unfair comparison.
>
> > **Q1: Asks for (I) the isolated effect of RL part, (II) whether baselines benefit from RL.**
>
> Thanks. **(I)** We have conducted ablations on our proposed GRPO-inspired RL: `Tab.3` shows that our RL scheme (row 3) improves generalization vs. baseline (row 1). `Fig.2` compares SFT-only, RL-only, and SFT-RL schemes on performance, demonstrating that our RL improves generalization and training stability. Besides, the performance gain is robust on different RL algorithms (cf. `Appendix D.2 & Tab.7`).
>
> **(II)** Good point. Directly porting RL to baselines is non-trivial since RL requires a stochastic policy net, while baselines use deterministic CLIP. Thus, **we apply our proposed GRPO-inspired RL strategy to baselines** and results below:
>
> |Base/Novel|Few-NonIID|Full-NonIID|Few-Dir(0.1)|Full-Dir(0.1)|
> |-|-|-|-|-|
> |pFedDC|60.46/74.27|56.74/62.56|76.06/77.68|69.08/71.55|
> |pFedMMA|79.51/82.39|39.41/53.63|78.10/80.09|68.14/71.39|
> |PromptFL|67.62/80.11|60.23/66.65|70.53/78.77|62.85/67.85|
> |FedMaPLe|82.67/82.39|79.83/73.96|83.03/80.96|82.76/75.80|
> |FedDTL(ours)|89.58/83.01|91.64/77.72|90.95/82.64|92.40/76.59|
>
> Compared to their original results (cf. `Tab.1`), the novel accuracy is improved, demonstrating our effective RL strategy on generalization enhancement. Base accuracy variations suggest that RL needs to be better redesigned into different FL objectives, which is an interesting direction for future work.

---

> > ### Author Rebuttal · Reviewer_NxRm · 2026-04-02
> >
> > My concerns are largely addressed and will raise score but I think the paper does require some significant writing revisions to incorporate the rebuttal answers and concerns of the reviewers in terms of better motivation and proof in particularly the RL aspect. For this reason I am borderline.

---

> > > ### Author Response · Authors · 2026-04-07
> > >
> > > Thanks for your thoughtful feedback and for raising your score. We are glad that our responses have largely addressed your main concerns. We also agree with your suggestion that the paper would benefit from clearer motivation and incorporating additional experimental analysis, particularly for the RL component. We will carefully incorporate the rebuttal clarifications and additional experiments into the revision. Importantly, we would like to emphasize that these modifications are primarily related to presentation and further additional experiments, rather than fundamental issues in our proposed framework or empirical validation. To address your concerns more concretely, we will revise our paper as follows:
> > >
> > > * **Server-side computation assumption**: We agree that this aspect can be better motivated. In the revision, ***we will incorporate a more explicit discussion in the "Related Works" section to clarify the practicality of server-side computation***. Specifically, we will add a subsection titled "Server-Assisted in Federated Learning" and demonstrate that the server-side training is a practical design via prior works, such as [R1], [R2], and split FL. However, these works always struggle to generalize to unseen classes. In contrast, VLMs (e.g., CLIP) learn a shared image-text embedding space, allowing zero-shot recognition via text prompts and suited for generalization tasks. But recent federated VLMs rely on purely local updates with simple server-side aggregation, making them difficult to handle inter-client optimization inconsistency in heterogeneous FL. Then, we will further highlight that our framework makes the first attempt to leverage server-side computation via our proposed novel decoupling scheme in federated VLMs to balance global task adaptation and generalization.
> > > * **RL motivation and validation**: We appreciate this suggestion. The motivation of the RL stage has been discussed in the final paragraph (i.e., L95 left-L64 right) of the "Introduction" section and "Reinforcement Learning for Federated Optimization" in the "Related Work" section. In the revision, ***we will further improve clarity on the RL motivation***: highlight that our proposed GRPO-inspired RL scheme is not merely an auxiliary technique but rather a core contribution of our work. Extending our proposed RL scheme to other baselines can be viewed as an additional ablation study from an orthogonal perspective, further demonstrating the effectiveness of our proposed RL scheme in improving generalization. ***We will include this additional empirical evidence in the appendix*** to further support its effectiveness, without affecting the main presentation.
> > > * **Privacy-related experiments**: The additional privacy enhancement (adding noise to embeddings) builds upon our existing framework (which aims to alleviate inter-client optimization inconsistency and intra-client over-specialization in federated VLMs for ensuring an effective balance between global task adaptation and generalization across various FL data distributions in both few-shot and full-data settings) and can be viewed as an extended privacy protection analysis. Similar to the "Impact of communication cost" in our paper (L433 left) for communication analysis, we view this as a complementary extension, and ***will include an "Impact of embedding uploading" analysis to evaluate the trade-off between privacy and performance in the "Ablation Study" subsection***.
> > >
> > > Overall, we believe these improvements are feasible within the revision. In our view, they mainly concern presentation, clarification, and additional strengthening, rather than the core ideas or the main validity of the proposed method. **We are truly grateful for your thoughtful feedback and for the time you have invested in the rebuttal process. We will carefully incorporate your suggestions into the final version.**
> > >
> > > [R1] Data-Free Knowledge Distillation for Heterogeneous Federated Learning, ICML 2021.
> > >
> > > [R2] Efficient Model Personalization in Federated Learning via Client-Specific Prompt Generation, ICCV 2023.

---

### Decision · Program_Chairs · 2026-04-30

**Decision:**

Accept (regular)

**Comment:**

**Summary as I observed.**
In this paper, the authors propose FedDTL, a federated learning framework for adapting vision-language models under heterogeneous client data, including decoupled encoder training and two-stage local fine-tuning strategies, which are validated through extensive experiments. The paper receives final scores of 5/5/3/4.

**Strengths as I observed.**
The reviewers generally consider the proposed decoupled training strategy to be novel and well-motivated for mitigating inter-client optimization inconsistency, which is consistent with my assessment. Meanwhile, the two-stage SFT+RL local training pipeline is viewed as a technically interesting contribution that improves generalization performance across diverse scenarios. Additionally, the experimental evaluation is comprehensive, covering multiple datasets under both few-shot and full-data settings. The paper is also clearly written and well organized overall, with presentation quality that supports the accessibility and reproducibility of the proposed approach.

**Weaknesses as I observed.**
Some reviewers raised concerns regarding the level of novelty, the motivation and positioning of the RL component, and the lack of formal privacy analysis for embedding sharing. In addition, the communication and computational efficiency analysis could be presented more clearly in the main paper. In the rebuttal, the authors responded to the reviewers' comments point by point and largely addressed these concerns. I suggest that the final version clarify the role of the RL stage with additional ablation evidence in the main text and provide a clearer discussion of privacy assumptions and efficiency trade-offs.

After carefully reviewing the original paper, the reviewers' comments, and the authors' responses, I believe that the strengths of this paper outweigh its weaknesses, and the proposed decoupled training strategy and two-stage fine-tuning pipeline make a meaningful contribution beyond existing methods. Given that, I recommend acceptance.